# Sample and Computationally Efficient Robust Learning of Gaussian Single-Index Models

**Puqian Wang**
Department of Computer Science
University of Wisconsin, Madison
pwang333@wisc.edu

**Nikos Zarifis**
Department of Computer Science
University of Wisconsin, Madison
zarifis@wisc.edu

**Ilias Diakonikolas**
Department of Computer Science
University of Wisconsin, Madison
ilias@cs.wisc.edu

**Jelena Diakonikolas**
Department of Computer Science
University of Wisconsin, Madison
jelena@cs.wisc.edu

## Abstract

A single-index model (SIM) is a function of the form $\sigma(\mathbf{w}^* \cdot \mathbf{x})$, where $\sigma : \mathbb{R} \to \mathbb{R}$ is a known link function and $\mathbf{w}^*$ is a hidden unit vector. We study the task of learning SIMs in the agnostic (a.k.a. adversarial label noise) model with respect to the $L_2^2$-loss under the Gaussian distribution. Our main result is a sample and computationally efficient agnostic proper learner that attains $L_2^2$-error of $O(\mathrm{OPT}) + \epsilon$, where OPT is the optimal loss. The sample complexity of our algorithm is $\tilde{O}(d^{\lceil k^*/2 \rceil} + d/\epsilon)$, where $k^*$ is the information-exponent of $\sigma$ corresponding to the degree of its first non-zero Hermite coefficient. This sample bound nearly matches known CSQ lower bounds, even in the realizable setting. Prior algorithmic work in this setting had focused on learning in the realizable case or in the presence of semi-random noise. Prior computationally efficient robust learners required significantly stronger assumptions on the link function.

## 1 Introduction

Single-index models (SIMs) [Ich93, HJS01, HMS$^+$04, DJS08, KS09, KKSK11, DH18, DGK$^+$20, DKTZ22, WZDD23, DNGL23] are a classical supervised learning model characterized by hidden low-dimensional structure. The term SIM refers to any function $f$ of the form $f(\mathbf{x}) = \sigma(\mathbf{w} \cdot \mathbf{x})$, where $\sigma : \mathbb{R} \to \mathbb{R}$ is the link (or activation) function and $\mathbf{w} \in \mathbb{R}^d$ is the hidden vector. In most settings of interest, the link function is assumed to satisfy certain regularity properties. Indeed, for an arbitrary function, learnability is information-theoretically impossible.

The efficient learnability of SIMs has been the focus of extensive investigation for several decades. For example, the special case where $\sigma$ is the sign function corresponds to Linear Threshold Functions whose study goes back to the Perceptron algorithm [Ros58]. Classical early works [KS09, KKSK11] studied the efficient learnability of SIMs for monotone and Lipschitz link functions. They showed that a gradient method efficiently learns SIMs for any distribution on the unit ball. More recently, there has been a resurgence of research on the topic with a focus on first-order methods. Indeed, the non-convex optimization landscape of SIMs exhibits rich structure and has become a useful testbed for the analysis of such methods. [Sol17] showed that SGD efficiently learns SIMs for the case that $\sigma$ is the ReLU activation and the distribution is Gaussian. [CLS15, CCFM19, SQW18] showed that gradient descent succeeds for the phase retrieval problem, corresponding to $\sigma(t) = t^2$ or $\sigma(t) = |t|$.

38th Conference on Neural Information Processing Systems (NeurIPS 2024).

More recently, a line of work [DH18, BAGJ21, DNGL23, DPVLB24] studied the efficient learnability of SIMs going significantly beyond the monotonicity assumption. Specifically, [BAGJ21, DNGL23] developed efficient gradient-based SIM learners for a general class of — not necessarily monotone — link functions under the Gaussian distribution. Roughly speaking, these works show that the complexity of learning SIMs is intimately related to the Hermite structure of the link function (roughly, the smallest degree non-zero Hermite coefficient). The results of the current paper are most closely related to the aforementioned works.

All the aforementioned algorithmic results succeed in the realizable model (i.e., with clean labels) or in a few cases in the presence of random label noise. The focus of this work is on learning SIMs in the challenging *agnostic* (or adversarial label noise) model [Hau92, KSS94]. In the agnostic model, no assumptions are made on the observed labels and the goal is to compute a hypothesis that is competitive with the *best-fit* function in the class. The algorithmic study of agnostically learning SIMs is not new. A recent line of work [DGK+20, DKTZ22, ATV23, WZDD23, GGKS23, ZWDD24] has given efficient agnostic SIM learners (typically based on first-order methods) with near-optimal error guarantees under natural distributional assumptions. *The key difference between prior work and the results of the current paper is in the assumptions on the link function.* Specifically, prior robust learners succeed for (a subclass of) *monotone and Lipschitz link functions.* In contrast, this work develops robust learners in the more general setting of [BAGJ21, DNGL23].

In order to precisely describe our contributions, we require the definition of the agnostic learning problem for Gaussian SIMs. Let $\mathcal{D}$ be a distribution of labeled examples $(\mathbf{x}, y) \in \mathbb{R}^d \times \mathbb{R}$ whose $\mathbf{x}$-marginal is the standard Gaussian, and let $\mathcal{L}_2^\sigma(\mathbf{w}) := \mathbf{E}_{(\mathbf{x},y)\sim\mathcal{D}}[(\sigma(\mathbf{w} \cdot \mathbf{x}) - y)^2]$ be the $L_2^2$ (or squared) loss of the hypothesis $h(\mathbf{x}) = \sigma(\mathbf{w} \cdot \mathbf{x})$ with respect to $\mathcal{D}$. Given i.i.d. samples from $\mathcal{D}$, the goal is to compute a hypothesis with squared error competitive with OPT, where OPT is the best attainable $L_2^2$-error by any function in the target class.

**Problem 1.1** (Robustly Learning Gaussian SIMs). *Let $\mathcal{D}$ be a distribution of labeled examples $(\mathbf{x}, y) \in \mathbb{R}^d \times \mathbb{R}$ whose $\mathbf{x}$-marginal is $\mathcal{D}_\mathbf{x} = \mathcal{N}(0, \mathbf{I}_d)$ and $y$ is arbitrary. We say that an algorithm is a $C$-approximate proper SIM learner, for some $C \geq 1$, if given $\epsilon > 0$ and i.i.d. samples from $\mathcal{D}$, the algorithm outputs a vector $\widehat{\mathbf{w}} \in \mathbb{S}^{d-1}$ such that with high probability it holds $\mathcal{L}_2^\sigma(\widehat{\mathbf{w}}) \leq C \operatorname{OPT} + \epsilon$, where $\operatorname{OPT} := \mathcal{L}_2^\sigma(\mathbf{w}^*)$ and $\mathbf{w}^* \in \operatorname{argmin}_{\mathbf{w} \in \mathbb{S}^{d-1}} \mathcal{L}_2^\sigma(\mathbf{w})$.*

First, note that Problem 1.1 does not make realizability assumptions on the distribution $\mathcal{D}$. That is, the labels are allowed to be arbitrary and the goal is to be competitive against the best-fit function in the class. Second, our focus is on obtaining efficient learners that achieve a *constant factor approximation* to the optimum loss — independent of the dimension $d$. The reason we require a constant factor approximation, instead of optimal error of $\operatorname{OPT} + \epsilon$, is the existence of computational hardness results ruling out this possibility. Specifically, even if the link function is the ReLU, there is strong evidence that any algorithm achieving error $\operatorname{OPT} + \epsilon$ in the above setting requires $d^{\operatorname{poly}(1/\epsilon)}$ time [DKZ20, GGK20, DKPZ21, DKR23].

Recent work [DGK+20, DKTZ22, ATV23, WZDD23] gave efficient, constant-factor robust learners, for the special case of Problem 1.1 where the link function lies in a proper subclass of monotone and Lipschitz functions. In this work, we obtain a broad generalization of these results to a much more general class of link functions, defined below.

We now proceed to formalize the assumptions on the link function. Let $\sigma : \mathbb{R} \to \mathbb{R}$ be a real-valued function that admits the Hermite decomposition $\sigma(z) = \sum_{k \geq 0} c_k \operatorname{he}_k(z)$, where $c_k = \mathbf{E}_{z \sim \mathcal{N}(0,1)}[\sigma(z)\operatorname{he}_k(z)]$ and $\operatorname{he}_k$ is the normalized probabilist's Hermite polynomial, defined by

$$\operatorname{he}_k(z) = \frac{(-1)^k}{\sqrt{k!}} \exp\left(\frac{z^2}{2}\right) \frac{\mathrm{d}^k}{\mathrm{d}z^k} \exp\left(-\frac{z^2}{2}\right).$$

We make the following assumptions.

**Assumption 1** (Family of Link Functions). *Suppose that $\sigma$ is normalized, namely $\mathbf{E}_{z \sim \mathcal{N}(0,1)}[\sigma^2(z)] = \sum_{k \geq 0} c_k^2 = 1$. We assume the following:*

*(i) The first non-zero Hermite coefficient has degree $k^*$ and is prominent: $c_{k^*}$ is an absolute constant that is bounded away from zero.*

*(ii) The fourth moment of $\sigma(z)$ is bounded: $\mathbf{E}_{z \sim \mathcal{N}(0,1)}[\sigma^4(z)] \leq B_4 < \infty$.*

*(iii) The second moment of the derivative of $\sigma(z)$ is bounded:* $\mathbf{E}_{z \sim \mathcal{N}(0,1)}[(\sigma'(z))^2] = \sum_{k \geq k^*} kc_k^2 \leq C_{k^*}$, *where $C_{k^*}$ is an absolute constant whenever $k^*$ is an absolute constant.*

*The parameter $k^*$ is known as the* information exponent *of $\sigma$.*

The information exponent $k^*$ can be viewed as a complexity measure of the link function. Specifically, ReLU activations correspond to $k^* = 1$. The same holds for the class of bounded activations considered in [DKTZ22, WZDD23]. The link functions used in phase retrieval have $k^* = 2$.

We note that Assumption 1 is (at least) as general as those used in [BAGJ21, DNGL23] — which focused on the realizable setting. Comparing against previous constant-factor agnostic learners, Assumption 1 strongly subsumes the class of "bounded activations" [DKTZ22, WZDD23]. In particular, it is easy to see that there exist functions satisfying Assumption 1 with constant $k^*$ that are far from monotone. See Appendix A for a detailed justification.

In prior work, [DNGL23], building on [BAGJ21], gave a sample-efficient gradient method for learning SIMs under Assumption 1 *in the realizable setting*. The sample complexity of their method is $\tilde{O}(d^{k^*/2} + d/\epsilon)$. This sample upper bound essentially matches known lower bounds in the Correlational Statistical Query (CSQ) model [DLS22, AAM23].

This discussion leads to the following question, which served as the motivation for the current work:

> *Is there an efficient constant-factor agnostic learner*
> *for Gaussian SIMs under Assumption 1?*

As our main contribution, we answer this question in the affirmative. Interestingly, our algorithm also relies on a gradient-method (Riemannian optimization over the sphere) following a non-trivial initialization step. We emphasize that this is the first polynomial-time constant-factor agnostic learner for this task under Assumption 1.

Specifically, we establish the following result (see Theorem 3.5 for a more detailed statement).

**Theorem 1.2** (Main Result, Informal). *There exists an algorithm that draws $n = \tilde{\Theta}_{k^*}(d^{\lceil k^*/2 \rceil} + d/\epsilon)$ labeled samples, runs in $\mathrm{poly}(n, d)$ time, and outputs a weight vector $\widehat{\mathbf{w}} \in \mathbb{S}^{d-1}$ that with high probability satisfies $\mathcal{L}_2^\sigma(\widehat{\mathbf{w}}) \leq C \, \mathrm{OPT} + \epsilon$, where $C = O(C_{k^*})$.*

Theorem 1.2 gives the first sample and computationally efficient robust learner for Gaussian SIMs under Assumption 1. This generalizes the algorithm of [DNGL23] to the agnostic setting and nearly matches the aforementioned CSQ lower bounds (our algorithm fits the CSQ framework). It is worth pointing out that, while more efficient (non-CSQ) algorithms have been developed for the realizable case [CM20], these algorithms provably fail in the agnostic regime. Finally, we remark that very recent work [DPVLB24] developed an efficient SIM learner and a nearly matching SQ lower bound in a model that allows for some forms of label noise. Importantly, their model does not capture the adversarial label noise studied here. More specifically, the algorithms developed in this prior work [DNGL23, DPVLB24] fail in the agnostic setting. See Appendix B for a detailed discussion.

## 1.1 Preliminaries

For $n \in \mathbb{Z}_+$, let $[n] := \{1, \ldots, n\}$. We use lowercase bold characters to denote vectors and uppercase bold characters for matrices and tensors. For $\mathbf{x} \in \mathbb{R}^d$ and $i \in [d]$, $\mathbf{x}_i$ denotes the $i$-th coordinate of $\mathbf{x}$, and $\|\mathbf{x}\|_2 := (\sum_{i=1}^d \mathbf{x}_i^2)^{1/2}$ denotes the $\ell_2$-norm of $\mathbf{x}$. We use $\mathbf{x} \cdot \mathbf{y}$ for the inner product of $\mathbf{x}, \mathbf{y} \in \mathbb{R}^d$ and $\theta(\mathbf{x}, \mathbf{y})$ for the angle between $\mathbf{x}, \mathbf{y}$. We slightly abuse notation and denote by $\mathbf{e}_i$ the $i^{\text{th}}$ standard basis vector in $\mathbb{R}^d$. We use $\mathbb{1}\{\mathcal{E}\}$ to denote the indicator of a statement $\mathcal{E}$. We use $\mathcal{N}_d$ to denote the $d$-dimensional standard Gaussian distribution, i.e., $\mathcal{N}_d = \mathcal{N}(0, \mathbf{I})$. We use $\mathbb{B}_d$ to denote the centered unit ball in $\mathbb{R}^d$ and denote the unit sphere in $\mathbb{R}^d$ by $\mathbb{S}^{d-1}$. We use $\mathrm{proj}_{\mathbb{B}_d}(\cdot)$ to denote the projection operator that projects a vector to the unit ball.

Given $\mathbf{M} \in \mathbb{R}^{d_1 \times d_2}$ and a left singular vector $\mathbf{v} \in \mathbb{R}^{d_1}$ of $\mathbf{M}$, we denote the corresponding singular value of $\mathbf{v}$ by $\rho(\mathbf{v})$. In addition, we use $\rho_1 \geq \rho_2 \geq \cdots \geq \rho_{\min\{d_1, d_2\}}$ to denote the singular values of a matrix $\mathbf{M} \in \mathbb{R}^{d_1 \times d_2}$. We use $\mathcal{S}_k$ to denote the set of all possible permutations of $k$ distinct objects. Given a unit vector $\mathbf{w}$, we define $\mathbf{P}_{\mathbf{w}^\perp} := \mathbf{I} - \mathbf{w}\mathbf{w}^\top$ to be the projection matrix that maps a vector $\mathbf{v}$ to its component that is orthogonal to $\mathbf{w}$, i.e., $\mathbf{P}_{\mathbf{w}^\perp}\mathbf{v} = \mathbf{v}^{\perp \mathbf{w}}$.

Given a vector $\mathbf{x} \in \mathbb{R}^d$, the (normalized) Hermite multivariate tensor is defined by [Rah17]:

$$(\mathbf{He}_k(\mathbf{x}))_{i_1,\dots,i_k} \coloneqq \left( \frac{\alpha_1! \dots \alpha_d!}{k!} \right)^{1/2} \mathrm{he}_{\alpha_1}(\mathbf{x}_1) \dots \mathrm{he}_{\alpha_d}(\mathbf{x}_d), \text{ where } \alpha_j = \sum_{l=1}^{k} \mathbb{1}\{i_l = j\}, \ \forall j \in [d].$$

We use the standard $O(\cdot), \Theta(\cdot), \Omega(\cdot)$ asymptotic notation. We use $\widetilde{O}(\cdot)$ to omit polylogarithmic factors in the argument. We write $E \gtrsim F$ for two non-negative expressions $E$ and $F$ to denote that *there exists* some positive universal constant $c > 0$ such that $E \geq c\,F$.

## 1.2 Technical Overview

Our technical approach consists of two main parts: (1) new results for tensor PCA, which allow us to obtain an initial parameter vector $\mathbf{w}^0$ that is nontrivially aligned with the target $\mathbf{w}^*$ and (2) a structural "alignment sharpness" result, which we use to argue that a variant of Riemannian minibatch stochastic gradient descent on a sphere applied to a "truncated square loss" (defined below) converges geometrically fast. In proving these results, we review elementary tensor algebra and basic properties of Hermite polynomials, and prove several structural results for Hermite polynomials that are instructive and may be useful to non-experts entering this area.

We now highlight some of the key ideas used in our work.

**Initialization via Tensor PCA** For our optimization algorithm to work, a warm start ensuring nontrivial alignment between the initial vector $\mathbf{w}^0$ and the target vector $\mathbf{w}^*$, as measured by $\mathbf{w}^0 \cdot \mathbf{w}^*$, is essential. In particular, a consequence of our results in Claim 3.1 and Lemma 3.2 is that $\mathbf{w}^0 \cdot \mathbf{w}^* = \Omega(1)$ is required to deal with the highly corruptive agnostic noise. Unfortunately, if we were to select $\mathbf{w}_0$ by drawing uniformly random samples from the sphere, exponentially many in $d$ such samples would be needed to ensure that with constant probability at least one of the sampled vectors $\mathbf{w}_0$ is such that $\mathbf{w}^0 \cdot \mathbf{w}^* = \Omega(1)$, due to standard results on concentration of measure on the (unit) sphere.

Perhaps surprisingly, we prove that the tensor PCA method developed in [RM14] when applied to our problem with $O(d^{\lceil k^*/2 \rceil})$ samples[1] ensures that $\mathbf{w}^0 \cdot \mathbf{w}^* \geq 1 - \min\{1/k^*, 1/2\}$. The reason that this result is surprising is that the method in [RM14] was developed to solve the following problem: given a $k$-tensor of the form

$$\mathbf{T} = \tau \mathbf{v}^{\otimes k} + \mathbf{A}, \tag{PCA-S}$$

where $\mathbf{A}$ is a $k$-tensor with i.i.d. standard Gaussian entries and $\tau > 0$ a "signal strength" parameter, recover the planted (signal) vector $\mathbf{v}$. This 'single-observation' model is equivalent (in law) to the following 'multi-observation' model ([BAGJ20]): given $n$ i.i.d. copies $\mathbf{T}^{(i)} = \tau' \mathbf{v}^{\otimes k} + \mathbf{A}^{(i)}$ with $\tau' = \tau/\sqrt{n}$, recover $\mathbf{v}$ using the empirical estimation:

$$\widehat{\mathbf{T}} = \tau' \mathbf{v}^{\otimes k} + (1/n) \sum_{i=1}^{n} \mathbf{A}^{(i)}. \tag{PCA-M}$$

In our setting, we wish to recover a vector $\mathbf{w}^*$ (up to some constant alignment error) for the $k$-Chow tensor $\mathbf{C}_k = \mathbf{E}_{(\mathbf{x},y)\sim\mathcal{D}}[y\mathbf{He}_k(\mathbf{x})]$, which can be decomposed as

$$\mathbf{C}_k = \mathop{\mathbf{E}}_{\mathbf{x}\sim\mathcal{N}_d} \Big[ \sum_{j \geq k^*} c_j \langle \mathbf{He}_j(\mathbf{x}), \mathbf{w}^{*\otimes j} \rangle \mathbf{He}_k(\mathbf{x}) \Big] + \mathop{\mathbf{E}}_{(\mathbf{x},y)\sim\mathcal{D}}[(y - \sigma(\mathbf{w}^* \cdot \mathbf{x}))\mathbf{He}_k(\mathbf{x})]. \tag{1}$$

The first term in this decomposition can be viewed as the "signal" $k$-tensor. The second term represents noise, which, due to $y$ being potentially arbitrary, cannot be assumed to be Gaussian. Thus, previously developed techniques for tensor PCA, which crucially rely on the "Gaussianity" of the noise term, do not apply here.

To obtain our result, we first argue that for any $k \geq k^*$, the top left singular vector $\mathbf{v}^*$ of the $k$-Chow tensor $\mathbf{C}_k$ unfolded into a matrix of roughly equal dimensions carries a "strong signal" about the target vector $\mathbf{w}^*$: its associated singular value scales with $c_k$ whenever $c_k = \Omega(\sqrt{\mathrm{OPT}})$ and it has a nontrivial alignment with the vectorized version of the $l$-tensor $\mathbf{w}^{*\otimes l}$ for $l = \lfloor k/2 \rfloor$ (Lemma 2.2).

To prove the desired alignment result using the empirical estimate of the matrix-unfolded $k$-Chow tensor $\mathbf{C}_k$, we rely on the application of a matrix concentration inequality obtained very recently

---

[1]Ignoring dependence on other problem parameters for simplicity; see Proposition 2.1 for a precise statement.

in [BvH22]. This requires a rather technical argument utilizing Gaussian hypercontractivity of multivariate polynomials of bounded degree, which we show characterizes the different "variance-like" quantities associated with the empirical estimate of (the matrix-unfolded) $\mathbf{C}_k$, for which we apply the aforementioned matrix concentration (see Lemma 2.4 and its proof).

Another intriguing aspect of our initialization result is that it is possible to use it directly to obtain an $O(\sqrt{\text{OPT}} + \epsilon)$-error solution. In particular, in the realizable case studied in [DNGL23], where $\text{OPT} = 0$, this result directly leads to error $O(\epsilon)$ in a sample and computationally efficient manner, with a rather simple algorithm and sample complexity comparable to [DNGL23].

**Optimization on a Sphere** The second key ingredient in our work is a structural result, stated in Lemma 3.3, which ensures that the gradient field (Riemannian gradient of a truncated loss) guiding the steps of our algorithm (which can be interpreted as a Riemmanian minibatch SGD on a sphere) negatively correlates with $\mathbf{w}^*$ to a significant extent. This property can be viewed as the considered gradient field, associated with the $L_2^2$ loss truncated to only contain the first nonzero term in the Hermite expansion of the activation function, containing a strong "signal" that can "pull" the algorithm iterates towards the target $O(\text{OPT}) + \epsilon$ solutions. We rely on this property to argue that as long as our algorithm (initialized using the tensor PCA method described above) does not have as its iterate a vector with $O(\text{OPT}) + \epsilon$ loss, the distance between the iterate vector and the target vector must contract. As a consequence, this algorithm converges in $O(\log(1/\epsilon))$ iterations.

This argument parallels the line of work [MBM18, DGK$^+$20, WZDD23, ZWDD24] on addressing learning of single-index models by proving structural, optimization-theory local error bounds that bound below the growth of a loss function outside the set of target solutions. Conceptually, the local error bounds from this line of work all have an intuitive interpretation as showing existence of a strong "signal" in the problem that can be used to guide algorithm updates towards target solutions. However, the methodology by which our structural result is obtained is entirely different, as it crucially relies on properties of Hermite polynomials, which were not considered in this past work.

## 2 Initialization Procedure

In this section, we show how to get an initial parameter vector $\mathbf{w}^0$ such that $\mathbf{w}^0 \cdot \mathbf{w}^* = 1 - \epsilon_0$ for some small constant $\epsilon_0$. The main technique is a tensor PCA algorithm that finds the principal component of a noisy degree-$k$-Chow tensor for any $k \geq k^*$, as long as $\text{OPT} \lesssim c_k^2$. Such a degree-$k$ Chow tensor is defined by $\mathbf{C}_k = \mathbf{E}_{(\mathbf{x},y)\sim\mathcal{D}}[y\mathbf{He}_k(\mathbf{x})]$, and we denote its noiseless counterpart by

$$\mathbf{C}_k^* = \mathop{\mathbf{E}}_{\mathbf{x}\sim\mathcal{N}_d}[\sigma(\mathbf{w}^* \cdot \mathbf{x})\mathbf{He}_k(\mathbf{x})] = \mathop{\mathbf{E}}_{\mathbf{x}\sim\mathcal{N}_d}\Big[\sum_{j\geq k^*} c_j\langle\mathbf{He}_j(\mathbf{x}), \mathbf{w}^{*\otimes j}\rangle\mathbf{He}_k(\mathbf{x})\Big].$$

Furthermore, let us denote the difference between $\mathbf{C}_k$ and $\mathbf{C}_k^*$ by

$$\mathbf{H}_k := \mathbf{C}_k - \mathbf{C}_k^* = \mathop{\mathbf{E}}_{(\mathbf{x},y)\sim\mathcal{D}}[(y - \sigma(\mathbf{w}^* \cdot \mathbf{x}))\mathbf{He}_k(\mathbf{x})].$$

Note that since $\mathbf{He}_k(\mathbf{x})$ is a symmetric tensor for any $\mathbf{x}$, all $\mathbf{C}_k, \mathbf{C}_k^*$, and $\mathbf{H}_k$ are symmetric tensors.

We use the following matrix unfolding operator that maps a $k$-tensor $\mathbf{T}$ to a matrix in $\mathbb{R}^{d^l \times d^{k-l}}$:[2]

$$\mathsf{Mat}_{(l,k-l)}(\mathbf{T})_{i_1+(i_2-1)d+\cdots+(i_l-1)d^{l-1}, j_1+\cdots+(j_{k-l}-1)d^{k-l-1}} := (\mathbf{T})_{i_1,i_2,\ldots,i_l,j_1,\ldots,j_{k-l}}$$

for all $i_1, \ldots, i_l, j_1, \ldots, j_{k-l} \in [d]$. We also define the 'vectorize' operator and 'tensorize' operators, which map a vector $\mathbf{v} \in \mathbb{R}^{d^l}$ to an $l$-tensor for any integer $l$, and vice versa. In detail,

$$\mathsf{Tensor}(\mathbf{v})_{i_1,\ldots,i_l} := \mathbf{v}_{i_1+(i_2-1)d+\cdots+(i_l-1)d^{l-1}}, \ \forall i_1, \ldots, i_l \in [d],$$

$$\mathsf{Vec}(\mathbf{v}^{\otimes l})_{i_1+(i_2-1)d+\cdots+(i_l-1)d^{l-1}} := \mathbf{v}_{i_1}\mathbf{v}_{i_2}\ldots\mathbf{v}_{i_l}, \ \forall i_1, \ldots, i_l \in [d].$$

Finally, given a vector $\mathbf{v} \in \mathbb{R}^{d^l}$, we can also convert this vector to a matrix of size $\mathbb{R}^{d \times d^{l-1}}$:

$$\mathsf{Mat}_{(1,l-1)}(\mathbf{v})_{i,j_1,\ldots,j_{l-1}} = \mathbf{v}_{i+(j_1-1)d+\cdots+(j_{l-1}-1)d^{l-1}}, \ \forall i, j_1, \ldots, j_{l-1} \in [d].$$

Throughout this section, we take $l := \lfloor k/2 \rfloor$. We leverage the tensor unfolding algorithm proposed in [RM14], which can succinctly be described as follows. First we unfold the degree-$k$ Chow tensor

---
**Algorithm 1** $k$-Chow Tensor PCA
---
1: **Input:** Parameters $\epsilon, k, \epsilon_0, c_k, B_4 > 0$; Sample access to $\mathcal{D}$
2: Let $l = \lfloor k/2 \rfloor$
3: Draw $n = \Theta(e^k \log^k (B_4/\epsilon) d^{k-l}/(\epsilon_0^2) + 1/\epsilon)$ samples $\{(\mathbf{x}^{(i)}, y^{(i)})\}_{i=1}^n$ from $\mathcal{D}$
4: Construct $\widehat{\mathbf{M}} := (1/n) \sum_{i=1}^n \mathsf{Mat}_{(l,k-l)}(y^{(i)} \mathbf{He}_k(\mathbf{x}^{(i)}))$; compute its top left singular vector $\widehat{\mathbf{v}}^*$
5: Compute the top-left singular vector $\widehat{\mathbf{u}}$ of the matrix $\mathsf{Mat}_{1,l-1}(\widehat{\mathbf{v}}^*)$
6: **Return:** $\widehat{\mathbf{u}}$
---

to a matrix in $\mathbb{R}^{d^l \times d^{k-l}}$ and find its top-left singular vector $\mathbf{v} \in \mathbb{R}^{d^l}$. Then, we calculate the matrix $\mathsf{Mat}_{(1,l-1)}(\mathbf{v})$, and output its top left singular vector $\mathbf{u}$.

Our main result for initialization is that the output of Algorithm 1 significantly correlates with $\mathbf{w}^*$.

**Proposition 2.1** (Initialization). *Suppose Assumption 1 holds. Assume that* $\mathrm{OPT} \leq c_{k^*}^2/(64k^*)^2$, *and let* $\epsilon_0 = c_{k^*}/(256k^*)$. *Then, Algorithm 1 applied to Problem 1.1 with* $k = k^*$ *uses* $n = \Theta((k^*)^2 e^{k^*} \log^{k^*}(B_4/\epsilon) d^{\lceil k^*/2 \rceil}/(c_{k^*}^2) + 1/\epsilon)$ *samples, runs in polynomial time, and outputs a vector* $\mathbf{w}^0 \in \mathbb{S}^{d-1}$ *such that* $\mathbf{w}^0 \cdot \mathbf{w}^* \geq 1 - \min\{1/k^*, 1/2\}$.

We remark here that Algorithm 1 can also be used to find an approximate solution of our agnostic learning problem; however the error dependence on OPT is *suboptimal*, scaling with its square-root. For full details of this argument, included for completeness, see Proposition D.3 in Appendix D.

In the remainder of this section, we sketch the proof of Proposition 2.1, which relies on two main ingredients: (1) alignment of the left singular vectors $\mathbf{v}$ of matrix-unfolded $k$-Chow tensor and the target vector $\mathbf{w}^*$, which can be interpreted as the $k$-Chow tensor containing a strong "signal" about the target vector $\mathbf{w}^*$, and (2) matrix concentration for the unfolded tensor, so that we can translate "population" properties of the $k$-Chow tensor to computable empirical quantities.

**Signal in the $k$-Chow Tensor** Our first observation is that for any left singular vector $\mathbf{v}$ of $\mathsf{Mat}_{(l,k-l)}(\mathbf{C}_k)$, the singular value $\rho(\mathbf{v})$ is close to the inner product between $\mathbf{v}$ and $\mathsf{Vec}(\mathbf{w}^{*\otimes l})$, where $l = \lfloor k/2 \rfloor$. Concretely, we have:

**Lemma 2.2.** *Let* $\mathbf{v}$ *be any left singular vector of* $\mathsf{Mat}_{(l,k-l)}(\mathbf{C}_k)$. *Then,* $|\rho(\mathbf{v}) - c_k(\mathsf{Vec}(\mathbf{w}^{*\otimes l}) \cdot \mathbf{v})| \leq \sqrt{\mathrm{OPT}}$.

*Proof Sketch of Lemma 2.2.* Recall that the singular value of the left singular vector $\mathbf{v}$ satisfies

$$\rho(\mathbf{v}) = \max_{\mathbf{r} \in \mathbb{R}^{d^{k-l}}, \|\mathbf{r}\|_2 = 1} \mathbf{v}^\top \mathsf{Mat}_{(l,k-l)}(\mathbf{C}_k)\mathbf{r} \overset{(i)}{=} \max_{\mathbf{r} \in \mathbb{R}^{k-l}, \|\mathbf{r}\|_2 = 1} \langle \mathbf{C}_k, \mathsf{Tensor}(\mathbf{v}) \otimes \mathsf{Tensor}(\mathbf{r}) \rangle,$$

where we used Fact C.1(2) in $(i)$. Since $\mathbf{C}_k = \mathbf{C}_k^* + \mathbf{H}_k$, we further have

$$\langle \mathbf{C}_k, \mathsf{Tensor}(\mathbf{v}) \otimes \mathsf{Tensor}(\mathbf{r}) \rangle = \langle \mathbf{C}_k^*, \mathsf{Tensor}(\mathbf{v}) \otimes \mathsf{Tensor}(\mathbf{r}) \rangle + \langle \mathbf{H}_k, \mathsf{Tensor}(\mathbf{v}) \otimes \mathsf{Tensor}(\mathbf{r}) \rangle.$$

We bound both terms above respectively. For the first term, plugging in the definition of $\mathbf{C}_k^*$ and using the orthonormality property of Hermite tensors (Fact C.3) and basic tensor algebraic calculations,

$$\langle \mathbf{C}_k^*, \mathsf{Tensor}(\mathbf{v}) \otimes \mathsf{Tensor}(\mathbf{r}) \rangle = c_k(\mathsf{Vec}(\mathbf{w}^{*\otimes l}) \cdot \mathbf{v})(\mathsf{Vec}(\mathbf{w}^{*\otimes k-l}) \cdot \mathbf{r}). \tag{2}$$

Next, for the second term, after applying Cauchy-Schwarz inequality, one can show that it holds

$$|\langle \mathbf{H}_k, \mathsf{Tensor}(\mathbf{v}) \otimes \mathsf{Tensor}(\mathbf{r}) \rangle| \leq \sqrt{\mathrm{OPT}} \|\mathrm{Sym}(\mathsf{Tensor}(\mathbf{v}) \otimes \mathsf{Tensor}(\mathbf{r}))\|_F \leq \sqrt{\mathrm{OPT}}. \tag{3}$$

Combining Equation (2) and Equation (3), we get that the singular value of $\mathbf{v}$ must satisfy

$$\rho(\mathbf{v}) \leq \max_{\mathbf{r} \in \mathbb{R}^{d^{k-l}}, \|\mathbf{r}\|_2 = 1} c_k(\mathsf{Vec}(\mathbf{w}^{*\otimes l}) \cdot \mathbf{v})(\mathsf{Vec}(\mathbf{w}^{*\otimes k-l}) \cdot \mathbf{r}) + \sqrt{\mathrm{OPT}}$$

$$= c_k(\mathsf{Vec}(\mathbf{w}^{*\otimes l}) \cdot \mathbf{v}) + \sqrt{\mathrm{OPT}}, \tag{4}$$

where in the equation above, we used the observation that as $\|\mathsf{Vec}(\mathbf{w}^{*\otimes k-l})\|_2 = \|\mathbf{w}^{*\otimes k-l}\|_F = 1$, it holds $\max_{\mathbf{r} \in \mathbb{R}^{d^{k-l}}, \|\mathbf{r}\|_2 = 1}(\mathsf{Vec}(\mathbf{w}^{*\otimes k-l}) \cdot \mathbf{r}) = \|\mathsf{Vec}(\mathbf{w}^{*\otimes k-l})\|_2 = 1$. Since Equation (3) implies $\langle \mathbf{H}_k, \mathsf{Tensor}(\mathbf{v}) \otimes \mathsf{Tensor}(\mathbf{r}) \rangle \geq -\sqrt{\mathrm{OPT}}$, similarly we have $\rho(\mathbf{v}) \geq c_k(\mathsf{Vec}(\mathbf{w}^{*\otimes l}) \cdot \mathbf{v}) - \sqrt{\mathrm{OPT}}$, completing the proof of Lemma 2.2. $\qquad\square$

---

[2] A summary of useful algebraic properties of the unfolded tensor is provided in Fact D.1 in Appendix D.

Lemma 2.2 implies that the top left singular vector $\mathbf{v}^*$ aligns well with $\mathsf{Vec}(\mathbf{w}^{*\otimes l})$. The full version of Corollary 2.3 is deferred to Corollary D.5.

**Corollary 2.3.** *The top left singular vector* $\mathbf{v}^* \in \mathbb{R}^{d^l}$ *of the unfolded tensor* $\mathsf{Mat}_{(l,k-l)}(\mathbf{C}_k)$ *satisfies* $\mathbf{v}^* \cdot \mathsf{Vec}(\mathbf{w}^{*\otimes l}) \geq 1 - (2\sqrt{\mathrm{OPT}})/c_k$.

**Concentration of the Unfolded Tensor Matrix** Let us denote $\mathbf{M}^{(i)} = \mathsf{Mat}_{(l,k-l)}(y^{(i)}\mathbf{He}_k(\mathbf{x}^{(i)}))$, for $i \in [n]$ and $\widehat{\mathbf{M}} = \frac{1}{n}\sum_{i=1}^n \mathbf{M}^{(i)}$, which is the empirical approximation of $\mathbf{M} = \mathsf{Mat}_{(l,k-l)}(\mathbf{C}_k) = \mathsf{Mat}_{(l,k-l)}(\mathbf{E}_{(\mathbf{x},y)\sim\mathcal{D}}[y\mathbf{He}_k(\mathbf{x})])$. Though we showed in Corollary 2.3 that the top left singular vector $\mathbf{v}^*$ of the population $\mathbf{M}$ strongly correlates with the signal $\mathsf{Vec}(\mathbf{w}^{*\otimes l})$, we only have access to the empirical estimate $\widehat{\mathbf{M}}$ and its corresponding top left singular vector $\widehat{\mathbf{v}}^*$. Thus, to guarantee that $\widehat{\mathbf{v}}^*$ correlates significantly with $\mathsf{Vec}(\mathbf{w}^{*\otimes l})$ as well, we need to show that the angle between the $\mathbf{v}^*$ and $\widehat{\mathbf{v}}^*$ is sufficiently small as long as we use sufficiently many samples. To this aim, we apply Wedin's theorem (Fact D.6). Wedin's theorem states that $\sin(\theta(\mathbf{v}^*, \widehat{\mathbf{v}}^*))$ can be bounded above by:

$$\sin(\theta(\mathbf{v}^*, \widehat{\mathbf{v}}^*)) \leq \|\mathbf{M} - \widehat{\mathbf{M}}\|_2/(\rho_1 - \rho_2 - \|\mathbf{M} - \widehat{\mathbf{M}}\|_2),$$

where $\rho_1$ and $\rho_2$ are the top 2 singular values of $\mathbf{M}$. We prove in Claim D.7 that $\rho_1 - \rho_2 \geq (c_k - 8\sqrt{\mathrm{OPT}})/2 \gtrsim c_k$ under the assumption that $\sqrt{\mathrm{OPT}} \lesssim c_k$, hence $\rho_1 - \rho_2$ is bounded below by a constant. Thus, our remaining goal is to bound the operator norm such that $\|\mathbf{M} - \widehat{\mathbf{M}}\|_2 \leq \epsilon_0$ where $\epsilon_0 > 0$ is a small constant. This can be accomplished by applying a recently obtained matrix concentration inequality from [BvH22, DPVLB24] (stated in Fact D.8), with additional technical arguments. Plugging the lower bound on the singular gap $\rho_1 - \rho_2$ and the upper bound on the operator norm $\|\mathbf{M} - \widehat{\mathbf{M}}\|_2$ back into Wedin's theorem (Fact D.6), we obtain the following main technical lemma of this subsection, whose proof can be found in Appendix D:

**Lemma 2.4** (Sample Complexity for Estimating the Unfolded Tensor Matrix). *Let* $\epsilon, \epsilon_0 > 0$. *Consider the unfolded matrix* $\mathbf{M} = \mathsf{Mat}_{(l,k-l)}(\mathbf{E}_{(\mathbf{x},y)\sim\mathcal{D}}[y\mathbf{He}_k(\mathbf{x})])$ *and its empirical estimate* $\widehat{\mathbf{M}} := (1/n)\sum_{i=1}^n \mathsf{Mat}_{(l,k-l)}(y^{(i)}\mathbf{He}_k(\mathbf{x}^{(i)}))$, *where* $\{(\mathbf{x}^{(i)}, y^{(i)})\}_{i=1}^n$ *are* $n = \Theta(e^k \log^k(B_4/\epsilon)d^{k/2}/\epsilon_0^2 + 1/\epsilon)$ *i.i.d. samples from* $\mathcal{D}$. *Then, with probability at least* $1 - \exp(-d^{1/2})$, $\|\widehat{\mathbf{M}} - \mathbf{M}\|_2 \leq \epsilon_0$. *Moreover, if* $\widehat{\mathbf{v}}^*$ *is the top left singular vector of* $\widehat{\mathbf{M}}$, *then with probability at least* $1 - \exp(-d^{1/2})$,

$$\widehat{\mathbf{v}}^* \cdot \mathsf{Vec}(\mathbf{w}^{*\otimes l}) \geq 1 - \frac{2}{c_k}\sqrt{\mathrm{OPT}} - \frac{2\epsilon_0}{(c_k/2 - 4\sqrt{\mathrm{OPT}}) - \epsilon_0}.$$

After getting an approximate top left singular vector $\widehat{\mathbf{v}}^* \in \mathbb{R}^{d^l}$ of $\mathsf{Mat}_{(l,k-l)}(\mathbf{E}_{(\mathbf{x},y)\sim\mathcal{D}}[y\mathbf{He}_k(\mathbf{x})])$, the final piece of the argument is that finding the top left singular vector of the matrix $\mathsf{Mat}_{(1,l-1)}(\widehat{\mathbf{v}}^*)$ completes the task of computing a vector $\mathbf{u}$ that correlates strongly with $\mathbf{w}^*$. Concretely, we have:

**Lemma 2.5.** *Suppose that* $\widehat{\mathbf{v}}^* \cdot \mathsf{Vec}(\mathbf{w}^{*\otimes l}) \geq 1 - \epsilon_1$ *for some* $\epsilon_1 \in (0, 1/16]$. *Then, the top left singular vector* $\mathbf{u} \in \mathbb{R}$ *of* $\mathsf{Mat}_{(1,l-1)}(\widehat{\mathbf{v}}^*)$ *satisfies* $\mathbf{u} \cdot \mathbf{w}^* \geq 1 - 2\epsilon_1$.

**Proof of Proposition 2.1** Since $\sqrt{\mathrm{OPT}} \leq c_{k^*}/(64k^*) \leq c_{k^*}/64$, choosing $\epsilon_0 = c_{k^*}/(256k^*) \leq c_{k^*}/256$ in Lemma 2.4, we obtain that using $n = \Theta((k^*)^2 e^{k^*} \log^{k^*}(B_4/\epsilon)d^{\lceil k^*/2 \rceil}/(c_{k^*}^2) + 1/\epsilon)$, it holds with probability at least $1 - \exp(-d^{1/2})$ that

$$\widehat{\mathbf{v}}^* \cdot \mathsf{Vec}(\mathbf{w}^{*\otimes l}) \geq 1 - \frac{2}{c_k}\sqrt{\mathrm{OPT}} - \frac{2\epsilon_0}{(c_k/2 - 4\sqrt{\mathrm{OPT}}) - \epsilon_0} \geq 1 - \frac{1}{16k^*}.$$

Then applying Lemma 2.5 with $\epsilon_1 \leq 1/(16k^*) \leq 1/16$, we get that the output $\mathbf{u}$ of Algorithm 1 satisfies $\mathbf{u} \cdot \mathbf{w}^* \geq 1 - 2\epsilon_1 \geq 1 - 1/(8k^*) \geq 1 - \min\{1/k^*, 1/2\}$, completing the proof. $\square$

## 3 Optimization via Riemannian Gradient Descent

After obtaining $\mathbf{w}^0$ from Algorithm 1, we run Riemannian minibatch SGD Algorithm 2 on the 'truncated loss' (see definition in Equation (5)). In the rest of the section, we first present the definition of the truncated $L_2^2$ loss $\mathcal{L}_2^\phi$ and its Riemannian gradient and then proceed to proving that Algorithm 2 converges to a constant approximate solution in $O(\log(1/\epsilon))$ iterations. Due to space constraints, omitted proofs are provided in Appendix E.

**Algorithm 2** Riemannian GD with Warm-start
___
1: **Input:** Parameters $\epsilon, k^*, c_{k^*}, B_4 > 0; T, \eta$; Sample access to $\mathcal{D}$.
2: $\mathbf{w}^0 = \textbf{Initialization}[\epsilon, k^*, c_{k^*}, B_4, \epsilon_0 = c_{k^*}/(256k^*)]$ (Algorithm 1).
3: **for** $t = 0, \dots, T-1$ **do**
4:     Draw $n = \Theta(C_{k^*} d e^{k^*} \log^{k^*+1}(B_4/\epsilon)/(\epsilon\delta))$ samples from $\mathcal{D}$
5:     $\widehat{\mathbf{g}}(\mathbf{w}^t) = \frac{1}{n}\sum_{i=1}^n k^* c_{k^*} y^{(i)}(\mathbf{I} - \mathbf{w}^t(\mathbf{w}^t)^\top)\langle\mathbf{He}_{k^*}(\mathbf{x}^{(i)}), (\mathbf{w}^t)^{\otimes k^*-1}\rangle$.
6:     $\mathbf{w}^{t+1} = (\mathbf{w}^t - \eta\widehat{\mathbf{g}}(\mathbf{w}^t))/\|\mathbf{w}^t - \eta\widehat{\mathbf{g}}(\mathbf{w}^t)\|_2$.
7: **Return:** $\mathbf{w}^T$
___

### 3.1 Truncated Loss and the Sharpness property of the Riemannian Gradient

Instead of directly minimizing the $L_2^2$ loss $\mathcal{L}_2^\sigma$, we work with the following truncated loss that drops all the terms higher than $k^*$ in the polynomial expansion of $\sigma$:

$$\mathcal{L}_2^\phi(\mathbf{w}) := 2\big(1 - \mathop{\mathbf{E}}_{(\mathbf{x},y)\sim\mathcal{D}}[y\phi(\mathbf{w}\cdot\mathbf{x})]\big), \text{ where } \phi(\mathbf{w}\cdot\mathbf{x}) = \langle\mathbf{He}_{k^*}(\mathbf{x}), \mathbf{w}^{\otimes k^*}\rangle. \tag{5}$$

Similarly, the noiseless surrogate loss is defined as

$$\mathcal{L}_2^{*\phi}(\mathbf{w}) := 2\big(1 - \mathop{\mathbf{E}}_{(\mathbf{x},y)\sim\mathcal{D}}[\sigma(\mathbf{w}^*\cdot\mathbf{x})\phi(\mathbf{w}\cdot\mathbf{x})]\big) = 2\big(1 - c_{k^*}(\mathbf{w}\cdot\mathbf{w}^*)^{k^*}\big). \tag{6}$$

It can be shown (using Fact C.1(2)) that the gradient of the truncated $L_2^2$ loss equals:

$$\nabla\mathcal{L}_2^\phi(\mathbf{w}) = -2\mathop{\mathbf{E}}_{(\mathbf{x},y)\sim\mathcal{D}}[\nabla\phi(\mathbf{w}\cdot\mathbf{x})y] = -2\mathop{\mathbf{E}}_{(\mathbf{x},y)\sim\mathcal{D}}\big[k^* c_{k^*} y\langle\mathbf{He}_{k^*}(\mathbf{x}), \mathbf{w}^{\otimes k^*-1}\rangle\big], \tag{7}$$

while for the gradient of the noiseless $L_2^2$ loss we have

$$\nabla\mathcal{L}_2^{*\phi}(\mathbf{w}) = -2\mathop{\mathbf{E}}_{(\mathbf{x},y)\sim\mathcal{D}}\big[k^* c_{k^*}\sigma(\mathbf{w}^*\cdot\mathbf{x})\langle\mathbf{He}_{k^*}(\mathbf{x}), \mathbf{w}^{\otimes k^*-1}\rangle\big]. \tag{8}$$

Recall that $\mathbf{P}_{\mathbf{w}^\perp} := \mathbf{I} - \mathbf{w}\mathbf{w}^\top$. Then the Riemannian gradient of the $L_2^2$ loss $\mathcal{L}_2^\phi$, denoted by $\mathbf{g}(\mathbf{w})$ is

$$\mathbf{g}(\mathbf{w}) := \mathbf{P}_{\mathbf{w}^\perp}(\nabla\mathcal{L}_2^\phi(\mathbf{w})) = -2\mathop{\mathbf{E}}_{(\mathbf{x},y)\sim\mathcal{D}}\big[k^* y\mathbf{P}_{\mathbf{w}^\perp}\langle\mathbf{He}_{k^*}(\mathbf{x}), \mathbf{w}^{\otimes k^*-1}\rangle\big]. \tag{9}$$

Similarly, the Riemannian gradient of the noiseless $L_2^2$ loss $\mathcal{L}_2^{*\phi}$ is defined by

$$\mathbf{g}^*(\mathbf{w}) := \mathbf{P}_{\mathbf{w}^\perp}(\nabla\mathcal{L}_2^{*\phi}(\mathbf{w})) = -2\mathop{\mathbf{E}}_{(\mathbf{x},y)\sim\mathcal{D}}\big[k^*\sigma(\mathbf{w}^*\cdot\mathbf{x})\mathbf{P}_{\mathbf{w}^\perp}\langle\mathbf{He}_{k^*}(\mathbf{x}), \mathbf{w}^{\otimes k^*-1}\rangle\big]. \tag{10}$$

We first show that $\mathbf{g}^*(\mathbf{w})$ carries information about the alignment between vectors $\mathbf{w}$ and $\mathbf{w}^*$.

**Claim 3.1.** *For any* $\mathbf{w}\in\mathbb{S}^{d-1}$, *we have* $\mathbf{g}^*(\mathbf{w}) = -2k^* c_{k^*}(\mathbf{w}\cdot\mathbf{w}^*)^{k^*-1}(\mathbf{w}^*)^{\perp\mathbf{w}}$.

Let us denote the difference between the noisy and the noiseless Riemannian gradient by $\xi(\mathbf{w})$, i.e., $\xi(\mathbf{w}) := \mathbf{g}(\mathbf{w}) - \mathbf{g}^*(\mathbf{w}) = -2\mathbf{E}_{(\mathbf{x},y)\sim\mathcal{D}}[(y - \sigma(\mathbf{w}^*\cdot\mathbf{x}))\mathbf{P}_{\mathbf{w}^\perp}\nabla\phi(\mathbf{w}\cdot\mathbf{x})]$. We next show that the norm of $\xi(\mathbf{w})$ and the inner product between $\xi(\mathbf{w})$ and $\mathbf{w}^*$ are both bounded.

**Lemma 3.2.** *Let* $\xi(\mathbf{w}) = \mathbf{g}(\mathbf{w}) - \mathbf{g}^*(\mathbf{w})$ *as defined above. Then,* $\|\xi(\mathbf{w})\|_2 \leq 2k^* c_{k^*}\sqrt{\mathrm{OPT}}$ *and* $|\xi(\mathbf{w})\cdot\mathbf{w}^*| \leq 2k^* c_{k^*}\sqrt{\mathrm{OPT}}\|(\mathbf{w}^*)^{\perp\mathbf{w}}\|_2$.

We are now ready to present the main structural result of this section.

**Lemma 3.3** (Sharpness). *Assume* $\mathrm{OPT} \leq c/(4e)^2$ *for some small absolute constant* $c < 1$. *Let* $\mathbf{w}\in\mathbb{S}^{d-1}$ *and suppose that* $\mathbf{w}\cdot\mathbf{w}^* \geq 1 - 1/k^*$. *Let* $\theta := \theta(\mathbf{w}, \mathbf{w}^*)$. *If* $\sin\theta \geq 4e\sqrt{\mathrm{OPT}}$, *then* $\mathbf{g}(\mathbf{w})\cdot\mathbf{w}^* \leq -(1/2)\|\mathbf{g}^*(\mathbf{w})\|_2\sin\theta$.

*Proof.* We start by noticing that by Claim 3.1, the noiseless gradient satisfies the following property:

$$\mathbf{g}^*(\mathbf{w})\cdot\mathbf{w}^* = -2k^* c_{k^*}(\mathbf{w}\cdot\mathbf{w}^*)^{k^*-1}\|(\mathbf{w}^*)^{\perp\mathbf{w}}\|_2^2 = -\|\mathbf{g}^*(\mathbf{w})\|_2\sin\theta,$$

where we used that since $\|\mathbf{w}\|_2 = \|\mathbf{w}^*\|_2 = 1$, we have $\|(\mathbf{w}^*)^{\perp\mathbf{w}}\|_2 = \sin\theta$. Furthermore, applying Lemma 3.2 we have the following sharpness property with respect to the $L_2^2$ loss:

$$\mathbf{g}(\mathbf{w})\cdot\mathbf{w}^* = \mathbf{g}^*(\mathbf{w})\cdot\mathbf{w}^* + \xi(\mathbf{w})\cdot\mathbf{w}^* \leq -(\|\mathbf{g}^*(\mathbf{w})\|_2 - 2k^* c_{k^*}\sqrt{\mathrm{OPT}})\sin\theta. \tag{11}$$

Observe that $(1 - 1/t)^{t-1} \geq 1/e$ for all $t \geq 1$. Therefore, when $\mathbf{w}\cdot\mathbf{w}^* \geq 1 - 1/k^*$, we have

$$\|\mathbf{g}^*(\mathbf{w})\|_2 = 2k^* c_{k^*}(\mathbf{w}\cdot\mathbf{w}^*)^{k^*-1}\sin\theta \geq 2k^* c_{k^*}(1 - 1/k^*)^{k^*-1}\sin\theta \geq e^{-1}k^* c_{k^*}\sin\theta.$$

Hence, when $\sin\theta \geq 4e\sqrt{\mathrm{OPT}}$ and $\mathbf{w}\cdot\mathbf{w}^* \geq 1 - 1/k^*$, we have $\|\mathbf{g}^*(\mathbf{w})\|_2 \geq 4k^* c_{k^*}\sqrt{\mathrm{OPT}}$. Thus, as long as $\sin\theta \geq 4e\sqrt{\mathrm{OPT}}$, we have that $\mathbf{g}(\mathbf{w})\cdot\mathbf{w}^* \leq -\frac{1}{2}\|\mathbf{g}^*(\mathbf{w})\|_2\sin\theta$. $\square$

## 3.2 Concentration of Gradients

Define the empirical estimate of $\mathbf{g}(\mathbf{w})$ as $\widehat{\mathbf{g}}(\mathbf{w}) \coloneqq (1/n)\sum_{i=1}^{n} k^* c_{k^*} y^{(i)} \mathbf{P}_{\mathbf{w}^\perp} \langle \mathbf{He}_{k^*}(\mathbf{x}^{(i)}), \mathbf{w}^{\otimes k^*-1} \rangle$. The following lemma provides the upper bounds on the number of samples required to approximate the Riemannian gradient $\mathbf{g}(\mathbf{w})$ by $\widehat{\mathbf{g}}(\mathbf{w})$.

**Lemma 3.4** (Concentration of Gradients). *Let $\mathbf{w}^*, \mathbf{w} \in \mathbb{S}^{d-1}$. Let $\widehat{\mathbf{g}}(\mathbf{w})$ be the empirical estimate of the Riemannian gradient. Furthermore, denote the angle between $\mathbf{w}$ and $\mathbf{w}^*$ by $\theta$, and denote $\kappa = (k^* c_{k^*})^2 e^{k^*} \log^{k^*}(B_4/\epsilon)$. Then, with probability at least $1 - \delta$, it holds $\|\widehat{\mathbf{g}}(\mathbf{w}) - \mathbf{g}(\mathbf{w})\|_2 \lesssim \sqrt{d\kappa/(n\delta)}$, and $(\widehat{\mathbf{g}}(\mathbf{w}) - \mathbf{g}(\mathbf{w})) \cdot \mathbf{w}^* \lesssim \sqrt{\kappa/(n\delta)} \sin^2 \theta$.*

## 3.3 Proof of Main Theorem

We proceed to the main theorem of this paper. It shows that using at most $\tilde{\Theta}(d^{\lceil k^*/2 \rceil} + d/\epsilon)$ samples, Algorithm 2 (with initialization from Algorithm 1) generates a vector $\widehat{\mathbf{w}}$ such that $\mathcal{L}_2^\sigma(\widehat{\mathbf{w}}) = O(\text{OPT}) + \epsilon$ within $O(\log(1/\epsilon))$ iterations.

**Theorem 3.5.** *Suppose Assumption 1 holds. Choose the batch size of Algorithm 2 to be $n = \Theta(C_{k^*} d e^{k^*} \log^{k^*+1}(B_4/\epsilon)/(\epsilon\delta))$, and choose the step size $\eta = 9/(40 e k^* c_{k^*})$. Then, after $T = O(\log(C_{k^*}/\epsilon))$ iterations, with probability at least $1 - \delta$, Algorithm 2 outputs $\mathbf{w}^T$ with $\mathcal{L}_2^\sigma(\mathbf{w}^T) = O(C_{k^*} \text{OPT}) + \epsilon$. The total sample complexity of Algorithm 2 is $N = \Theta((k^*/c_{k^*})^2 e^{k^*} \log^{k^*}(B_4/\epsilon) d^{\lceil k^*/2 \rceil} + (C_{k^*} e^{k^*} \log^{k^*+2}(B_4/\epsilon)) \frac{d}{\epsilon\delta})$.*

*Proof Sketch of Theorem 3.5.* Suppose first that $\text{OPT} \geq (c_{k^*}/64k^*)^2$, then by Claim E.7 we know that any unit vector (e.g., $\widehat{\mathbf{w}} = \mathbf{e}_1$) is a constant approximate solution with $\mathcal{L}_2^\sigma(\widehat{\mathbf{w}}) = O(\text{OPT})$. Now suppose that $\text{OPT} \leq (c_{k^*}/64k^*)^2$. Consider the distance between $\mathbf{w}^t$ and $\mathbf{w}^*$ after each update of Algorithm 2. By the non-expansive property of projection operators, we have

$$\|\mathbf{w}^{t+1} - \mathbf{w}^*\|_2^2 = \|\text{proj}_{\mathbb{B}_d}(\mathbf{w}^t - \eta\widehat{\mathbf{g}}(\mathbf{w}^t)) - \mathbf{w}^*\|_2^2 \leq \|\mathbf{w}^t - \eta\widehat{\mathbf{g}}(\mathbf{w}^t) - \mathbf{w}^*\|_2^2$$
$$= \|\mathbf{w}^t - \mathbf{w}^*\|_2^2 + 2\eta\widehat{\mathbf{g}}(\mathbf{w}^t) \cdot (\mathbf{w}^* - \mathbf{w}^t) + \eta^2\|\widehat{\mathbf{g}}(\mathbf{w}^t)\|_2^2. \quad (12)$$

Let $\theta_t = \theta(\mathbf{w}^t, \mathbf{w}^*)$. For the chosen batch size, Lemma 3.4 implies

$$\|\widehat{\mathbf{g}}(\mathbf{w}^t) - \mathbf{g}(\mathbf{w}^t)\|_2^2 \leq (k^* c_{k^*})^2 \epsilon, \quad (\widehat{\mathbf{g}}(\mathbf{w}^t) - \mathbf{g}(\mathbf{w}^t)) \cdot \mathbf{w}^* \leq \sqrt{\epsilon/d} \sin^2 \theta_t.$$

Assume for now that $\sin\theta_t \geq 4e\sqrt{\text{OPT}} + \sqrt{\epsilon}$, hence Lemma 3.3 applies. As $\widehat{\mathbf{g}}(\mathbf{w}^t) \cdot (\mathbf{w}^* - \mathbf{w}^t) = (\widehat{\mathbf{g}}(\mathbf{w}^t) - \mathbf{g}(\mathbf{w}^t)) \cdot \mathbf{w}^* + \mathbf{g}(\mathbf{w}^t) \cdot \mathbf{w}^*$, using Lemma 3.3, we get

$$\widehat{\mathbf{g}}(\mathbf{w}^t) \cdot (\mathbf{w}^* - \mathbf{w}^t) \leq \sqrt{\epsilon/d} \sin\theta_t - (1/2)\|\mathbf{g}^*(\mathbf{w}^*)\|_2 \sin\theta_t. \quad (13)$$

On the other hand, the squared norm term $\|\widehat{\mathbf{g}}(\mathbf{w}^t)\|_2^2$ from Equation (12) can be bounded above by

$$\|\widehat{\mathbf{g}}(\mathbf{w}^t)\|_2^2 \leq 2\|\widehat{\mathbf{g}}(\mathbf{w}^t) - \mathbf{g}(\mathbf{w}^t)\|_2^2 + 2\|\mathbf{g}(\mathbf{w}^t)\|_2^2 \leq (k^* c_{k^*})^2(\text{OPT} + \epsilon) + \|\mathbf{g}^*(\mathbf{w})\|_2^2. \quad (14)$$

Plugging Equation (13) and Equation (14) back into Equation (12), we get that w.p. at least $1 - \delta$,

$$\|\mathbf{w}^{t+1} - \mathbf{w}^*\|_2^2 \leq \|\mathbf{w}^t - \mathbf{w}^*\|_2^2 + 2\eta(\sqrt{\epsilon/d} - \|\mathbf{g}^*(\mathbf{w}^t)\|_2/2)\sin\theta_t$$
$$+ \eta^2((k^* c_{k^*})^2(\text{OPT} + \epsilon) + \|\mathbf{g}^*(\mathbf{w})\|_2^2). \quad (15)$$

Let us assume first that $\theta_t \leq \theta_{t-1} \leq \cdots \leq \theta_0$ and $\sin\theta_t \geq 4e\sqrt{\text{OPT}} + \sqrt{\epsilon}$, then we show that it holds $\theta_{t+1} \leq \theta_t$. Recall in Claim 3.1 it was shown that $\|\mathbf{g}^*(\mathbf{w}^t)\|_2 = 2k^* c_{k^*}(\mathbf{w}^t \cdot \mathbf{w}^*)^{k^*-1} \sin\theta_t$. Since $\mathbf{w}^0$ is the initial parameter vector that satisfies $\mathbf{w}^0 \cdot \mathbf{w}^* \geq 1 - 1/k^*$, by the inductive hypothesis it holds $\mathbf{w}^t \cdot \mathbf{w}^* \geq 1 - 1/k^*$ and hence $1 \geq (\mathbf{w}^t \cdot \mathbf{w}^*)^{k^*-1} \geq 1/e$. Therefore, we further obtain $(2k^* c_{k^*}/e)\sin\theta_t \leq \|\mathbf{g}^*(\mathbf{w}^t)\|_2 \leq 2k^* c_{k^*} \sin\theta_t$. Therefore, using the inductive assumption that $\sin\theta_t \geq 4e\sqrt{\text{OPT}} + \sqrt{\epsilon}$ and further noticing that $\|\mathbf{w}^t - \mathbf{w}^*\|_2 = 2\sin(\theta_t/2)$, with step size $\eta = 9/(40 e k^* c_{k^*})$ we can further bound $\|\mathbf{w}^{t+1} - \mathbf{w}^*\|_2^2$ in Equation (15) as:

$$\|\mathbf{w}^{t+1} - \mathbf{w}^*\|_2^2 \leq (1 - (81/(320e^2)))\|\mathbf{w}^t - \mathbf{w}^*\|_2^2. \quad (16)$$

This shows that $\theta_{t+1} \leq \theta_t$, hence completing the inductive argument. Furthermore, Equation (16) implies that after at most $T = O(\log(1/\epsilon))$ iterations it must hold that $\sin\theta_T \leq 4e\sqrt{\text{OPT}} + \sqrt{\epsilon}$, therefore, we can end the algorithm after at most $O(\log(1/\epsilon))$ iterations. Applying Claim E.7 we know that this final vector $\mathbf{w}^T$ has error bound $\mathcal{L}_2^\sigma(\mathbf{w}^T) = O(C_{k^*}(\text{OPT} + \epsilon))$. Thus, choosing $\epsilon' = C_{k^*}\epsilon$, $\delta' = \delta T$, where $T = O(\log(C_{k^*}/\epsilon'))$, Algorithm 2 outputs a parameter $\mathbf{w}^T$ such that with probability at least $1 - \delta'$, $\mathcal{L}_2^\sigma(\mathbf{w}^T) = O(C_{k^*}\text{OPT}) + \epsilon'$, with batch size $n = \tilde{\Theta}(d/(\epsilon'\delta'))$. In summary, the total number of samples required for Algorithm 2 is $N = \tilde{\Theta}(d^{\lceil k^*/2 \rceil} + d/(\epsilon'\delta'))$. $\square$

## Acknowledgments

PW was supported in part by NSF Award DMS-2023239. NZ was supported in part by NSF Medium Award CCF-2107079. ID was supported in part by NSF Medium Award CCF-2107079 and an H.I. Romnes Faculty Fellowship. JD was supported in part by the Air Force Office of Scientific Research under award number FA9550-24-1-0076 and by the U.S. Office of Naval Research under contract number N00014-22-1-2348. Any opinions, findings and conclusions or recommendations expressed in this material are those of the author(s) and do not necessarily reflect the views of the U.S. Department of Defense.

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

# Supplementary Material

**Organization** The appendix is organized as follows. In Appendix A, we provide a detailed discussion about our assumptions on the activation functions. In Appendix B we compare our results with the most related prior works. In Appendix C, we provide additional preliminaries on basic tensor algebra as well as Hermite polynomials and Hermite tensors. In Appendix D and Appendix E we provide full versions of Section 2 and Section 3 respectively, with omitted lemmas and proofs.

## A Remarks on the Assumptions

Assumption 1($i$) appears in the same form in [BAGJ21, DNGL23]. The remaining two assumptions are implied by $|\sigma'(z)| \le A|z|^q + B$ for constants $A, B, q$, assumed in these works. In detail, Assumption 1($ii$) can be implied from $|\sigma'(z)| \le A|z|^q + B$ using the Gagliardo–Nirenberg inequality [FFRS21]; Assumption 1($iii$) follows from direct calculations as $\mathbf{E}_{z \sim \mathcal{N}_1}[(A|z|^q + B)^2]$ is finite. Hence Assumption 1 is no stronger than the assumptions made in prior work, which considered the less challenging realizable and zero-mean label noise settings.

Some activations satisfying Assumption 1 are:

1. All '$(a, b)$-well-behaved' activations [DGK$^+$20, DKTZ22, WZDD23] that are non-decreasing, zero at the origin, $b$-Lipschitz and $\sigma'(z) \ge a$ when $z \in [0, R]$ satisfy Assumption 1 with $k^* = 1$ (see Claim A.1). This includes ReLU, $\sigma(z) = \max\{0, z\}$ and sigmoid, $\sigma(z) = e^z/(1 + e^z)$.

2. Activations for phase-retrieval [CLS15, CCFM19, SQW18]: $\sigma(z) = z^2$ and $\sigma(z) = |z|$ have information component $k^* = 2$. One can verify that they satisfy Assumption 1 after normalization.

**Claim A.1.** *Let $a, b, R > 0$ be some absolute constants. Let $\sigma : \mathbb{R} \to \mathbb{R}$ be an activation such that it is non-decreasing, $b$-Lipschitz, and satisfies $\sigma(0) = 0$ and $\sigma'(z) \ge a$ for all $z \in [0, R]$. Then $\sigma$ satisfies Assumption 1 with information component $k^* = 1$.*

*Proof.* We calculate the Hermite coefficient $c_1$ of $\sigma$:

$$\mathbf{E}_{z \sim \mathcal{N}_1}[\sigma(z)z] = \mathbf{E}_{z \sim \mathcal{N}_1}[\sigma(z)z \mathbb{1}\{z \ge 0\}] + \mathbf{E}_{z \sim \mathcal{N}_1}[\sigma(z)z \mathbb{1}\{z \le 0\}]$$

$$\overset{(i)}{\ge} \mathbf{E}_{z \sim \mathcal{N}_1}[\sigma(z)z \mathbb{1}\{z \ge 0\}] \ge \mathbf{E}_{z \sim \mathcal{N}_1}[az^2 \mathbb{1}\{z \in [0, R]\}] \gtrsim aR^3 \exp(-R^2),$$

where in $(i)$ we used the monotonicity property of $\sigma$ and that $\sigma(0) = 0$. Thus we get that all well-behaved activations as $k^* = 1$. In addition, the $\mathbf{E}_{z \sim \mathcal{N}_1}[\sigma(z)^2] \le \mathbf{E}_{z \sim \mathcal{N}_1}[b^2 z^2] \le b^2$, hence after normalization, we have $c_1 \gtrsim aR^3 \exp(-R^2)/(2b^2)$, which is an absolute constant bounded away from 0. Finally, we have $\mathbf{E}_{z \sim \mathcal{N}_1}[\sigma(z)^4] \le 3b^4$ and $\mathbf{E}_{z \sim \mathcal{N}_1}[(\sigma'(z))^2] \le b^2$ since $\sigma$ is $b$-Lipschitz. Thus, we have that all well-behaved activations satisfy Assumption 1. $\qquad\square$

## B Comparison with Prior Work

### B.1 Comparison with Prior Works on Agnostically Learning SIMs

A long thread of research has been focusing on agnostic learning of single index models, including [FCG20, DGK$^+$20, DKTZ22, WZDD23, ZWDD24]. A common assumption on the activation function $\sigma$ used in these works is the so-called "well-behaved" property, namely that $\sigma$ is non-decreasing, zero at the origin ($\sigma(0) = 0$), $b$-Lipschitz, and $\sigma'(z) \ge a$ when $z \in [0, R]$. In particular, ReLUs and sigmoids are well-behaved activations. For well-behaved activations, we have shown in Claim A.1 that they have $k^* = 1$ and satisfy Assumption 1. Therefore, we conclude that our assumption Assumption 1 is indeed milder compared to prior works.

We also note that Assumption 1 allows for non-monotonic activations as well, for example $\sigma(z) = z^2$ and $\sigma(z) = |z|$ satisfy Assumption 1 with $k^* = 2$.

At the level of techniques, the algorithmic approaches in the aforementioned prior works on agnostically learning SIMs [FCG20, DGK$^+$20, WZDD23] inherently fail in our more general activation setting. The main reason is that the underlying algorithms only exploit the information in the degree

1-Chow parameters. However, under Assumption 1 with $k^* \geq 2$, the inner product between degree 1-Chow vector and any unit vector $\mathbf{v}$ equals $|\mathbf{E}_{(\mathbf{x},y)\sim\mathcal{D}}[y\mathbf{x}] \cdot \mathbf{v}| \leq \sqrt{\mathrm{OPT}}$ — which provides no information about the hidden vector $\mathbf{w}^*$. That is, in our more general setting, considering Chow vectors of higher degree appears necessary for any optimization method to succeed.

On the other hand, we remark that the algorithms in these works succed under more general distributions (including the Gaussian distribution), such as log-concave distributions.

## B.2 Comparison with [DNGL23]

The work by [DNGL23] applied a smoothing operator to the $L_2^2$ loss inspired by [ADGM17]. For any function $f : \mathbb{R} \to \mathbb{R}$, let the smoothing operator be

$$g_\lambda(f(\mathbf{w} \cdot \mathbf{x})) := \mathop{\mathbf{E}}_{\mu_{\mathbf{w}}} \left[ f\left( \frac{\mathbf{w} + \lambda \mathbf{z}}{\|\mathbf{w} + \lambda \mathbf{z}\|_2} \cdot \mathbf{x} \right) \right],$$

where $\mathbf{z} \sim \mu_{\mathbf{w}}$ is the uniform distribution on the sphere $\mathbb{S}^{d-2}$ that is orthogonal to $\mathbf{w}$, and $\lambda$ is the 'smoothing strength'. Applying the smoothing operator to the loss we get

$$\mathcal{L}_\lambda(\mathbf{w}) := 1 - \mathop{\mathbf{E}}_{\mu_{\mathbf{w}}} \left[ \mathop{\mathbf{E}}_{(\mathbf{x},y)\sim\mathcal{D}} \left[ y\sigma\left( \frac{\mathbf{w} + \lambda \mathbf{z}}{\|\mathbf{w} + \lambda \mathbf{z}\|_2} \cdot \mathbf{x} \right) \right] \right],$$

and

$$\ell_\lambda(\mathbf{w}; \mathbf{x}, y) := 1 - \mathop{\mathbf{E}}_{\mu_{\mathbf{w}}} \left[ y\sigma\left( \frac{\mathbf{w} + \lambda \mathbf{z}}{\|\mathbf{w} + \lambda \mathbf{z}\|_2} \cdot \mathbf{x} \right) \right].$$

The main algorithm of [DNGL23] is an online Riemannian gradient descent algorithm on the smoothed loss $\mathcal{L}_\lambda(\mathbf{w})$.

The dynamics of the online SGD algorithm in [DNGL23] can be split into three stages: In the first stage, starting from a randomly initialized vector $\mathbf{w}^0$ such that $\mathbf{w}^0 \cdot \mathbf{w}^* = \Theta(d^{-1/2})$, the algorithm used a large smoothing operator with $\lambda = d^{1/4}$. Then, after $\tilde{O}(d^{k^*/2-1})$ iterations the algorithm converges to a parameter $\mathbf{w}^1$ such that $\mathbf{w}^1 \cdot \mathbf{w}^* \geq d^{-1/4}$. Then, in the second stage, with zero-smoothing $\lambda = 0$, they run the algorithm for another $\tilde{O}(d^{k^*/2-1})$ iterations and get a parameter $\mathbf{w}^2$ such that $\mathbf{w}^2 \cdot \mathbf{w}^* \geq 1 - d^{-1/4}$. In the third stage, online SGD converges to $\mathbf{w}^3$ with $\mathbf{w}^3 \cdot \mathbf{w}^* \geq 1 - \epsilon$ in $d/\epsilon$ iterations.

However, the smoothing technique does not work in the agnostic setting. The main reason is that the agnostic noise buries the signal of the gradient. To see this, note that in Lemma 11 of [DNGL23], they showed that $g_\lambda(\mathrm{he}_k(\mathbf{w} \cdot \mathbf{x})) = \langle \mathbf{He}_k(\mathbf{x}), \mathbf{T}_k(\mathbf{w}) \rangle$, where

$$\mathbf{T}_k(\mathbf{w}) := \frac{1}{(1+\lambda^2)^{k/2}} \sum_{j=0}^{\lfloor k/2 \rfloor} \binom{k}{2j} \mathrm{Sym}(\mathbf{w}^{\otimes k-2j} \otimes (\mathbf{P}_{\mathbf{w}^\perp}^{\otimes j})) \lambda^{2j} \nu_j^{(d-1)},$$

and $\nu_j^{(d-1)} = \mathbf{E}_{\mathbf{z}\sim\mathbb{S}^{d-2}}[\mathbf{z}_1^{2j}] = \Theta((d-1)^{-j}) = \Theta(d^{-j})$. Note since the smoothing operator is a linear operator, it holds that $\mathcal{L}_\lambda(\mathbf{w}) = 1 - \mathbf{E}_{(\mathbf{x},y)\sim\mathcal{D}}[yg_\lambda(\sigma(\mathbf{w}\cdot\mathbf{x}))] = 1 - \mathbf{E}_{(\mathbf{x},y)\sim\mathcal{D}}[y\sum_{k\geq k^*} c_k g_\lambda(\mathrm{he}_k(\mathbf{w}\cdot\mathbf{x}))] = 1 - \mathbf{E}_{(\mathbf{x},y)\sim\mathcal{D}}[y\sum_{k\geq k^*} c_k\langle \mathbf{He}_k(\mathbf{x}), \mathbf{T}_k(\mathbf{w}) \rangle]$. Let $\mathbf{g}_\lambda(\mathbf{w}) := \mathbf{P}_{\mathbf{w}^\perp}\nabla\mathcal{L}_\lambda(\mathbf{w})$ denote the Riemannian gradient of $\mathcal{L}_\lambda(\mathbf{w})$, we have

$$\mathbf{g}_\lambda(\mathbf{w}) \cdot \mathbf{w}^* = - \mathop{\mathbf{E}}_{(\mathbf{x},y)\sim\mathcal{D}} \left[ y \sum_{k\geq k^*} c_k\langle \mathbf{He}_k(\mathbf{x}), \nabla\mathbf{T}_k(\mathbf{w}) \otimes (\mathbf{w}^*)^{\perp_{\mathbf{w}}} \rangle \right]$$

$$= - \underbrace{\mathop{\mathbf{E}}_{(\mathbf{x},y)\sim\mathcal{D}} \left[ \sigma(\mathbf{w}^* \cdot \mathbf{x}) \sum_{k\geq k^*} c_k\langle \mathbf{He}_k(\mathbf{x}), \nabla\mathbf{T}_k(\mathbf{w}) \otimes (\mathbf{w}^*)^{\perp_{\mathbf{w}}} \rangle \right]}_{\mathcal{I}_1}$$

$$- \underbrace{\mathop{\mathbf{E}}_{(\mathbf{x},y)\sim\mathcal{D}} \left[ (y - \sigma(\mathbf{w}^* \cdot \mathbf{x})) \sum_{k\geq k^*} c_k\langle \mathbf{He}_k(\mathbf{x}), \nabla\mathbf{T}_k(\mathbf{w}) \otimes (\mathbf{w}^*)^{\perp_{\mathbf{w}}} \rangle \right]}_{\mathcal{I}_2} \quad (17)$$

We show that the noise term $\mathcal{I}_2$ kills the signal term $\mathcal{I}_1$ in the beginning stage when $\mathbf{w}\cdot\mathbf{w}^* = \Theta(1/\sqrt{d})$ and $\lambda = d^{1/4}$.

**Claim B.1.** *When* $\mathbf{w} \cdot \mathbf{w}^* = \Theta(1/\sqrt{d})$ *and* $\lambda = d^{1/4}$, *it holds*

$$|\mathcal{I}_1| \lesssim d^{-k^*/2}, \ |\mathcal{I}_2| \lesssim \sqrt{\mathrm{OPT}} d^{-k^*/4}.$$

Thus, in order for the signal to overcome the noise, one requires $\mathrm{OPT} \lesssim d^{-k^*/4}$, which is too strict to hold in reality.

*Proof of Claim B.1.* Following the steps in Lemma 12 in [DNGL23], $\nabla \mathbf{T}_k(\mathbf{w}) \otimes (\mathbf{w}^*)^{\perp \mathbf{w}}$ equals:

$$
\begin{aligned}
&\nabla \mathbf{T}_k(\mathbf{w}) \otimes (\mathbf{w}^*)^{\perp \mathbf{w}} \\
&= \frac{k}{(1+\lambda^2)^{k/2}} \sum_{j=0}^{\lfloor \frac{k-1}{2} \rfloor} \binom{k-1}{2j} \mathrm{Sym}(\mathbf{w}^{\otimes k-2j-1} \otimes \mathbf{P}_{\mathbf{w}^\perp}^{\otimes j}) \otimes (\mathbf{w}^*)^{\perp \mathbf{w}} \lambda^{2j} \nu_j^{(d-1)} \\
&\quad - \frac{\lambda^2 k(k-1)}{(d-1)(1+\lambda^2)^{k/2}} \sum_{j=0}^{\lfloor \frac{k-2}{2} \rfloor} \binom{k-1}{2j} \mathrm{Sym}(\mathbf{w}^{\otimes k-2j-1} \otimes \mathbf{P}_{\mathbf{w}^\perp}^{\otimes j}) \otimes (\mathbf{w}^*)^{\perp \mathbf{w}} \lambda^{2j} \nu_j^{(d+1)}.
\end{aligned}
$$

Plugging the equation above back into Equation (17), we study each term $|\mathcal{I}_1|$ and $|\mathcal{I}_2|$ respectively in the beginning phase when $\mathbf{w} \cdot \mathbf{w}^* = O(d^{-1/2})$.

Using Fact C.3, and recall that in [DNGL23] $\lambda$ is chosen to be $\lambda = d^{1/4}$ and $\nu_j^{(d-1)} = \Theta(d^{-j})$, $\nu_j^{(d+1)} = \Theta(d^{-j})$ by definition, $\mathcal{I}_1$ equals:

$$
\begin{aligned}
|\mathcal{I}_1| &= \sum_{k \geq k^*} c_k^2 \left\langle \mathrm{Sym}(\mathbf{w}^{*\otimes k}), \mathrm{Sym}(\nabla \mathbf{T}_k \otimes (\mathbf{w}^*)^{\perp \mathbf{w}}) \right\rangle \\
&\lesssim \sum_{k \geq k^*} \frac{c_k^2 k}{(1+\lambda^2)^{k/2}} \sum_{j=0}^{\lfloor \frac{k-1}{2} \rfloor} \binom{k-1}{2j} (\mathbf{w} \cdot \mathbf{w}^*)^{k-2j-1} (1 - (\mathbf{w} \cdot \mathbf{w}^*)^2)^j \|(\mathbf{w}^*)^{\perp \mathbf{w}}\|_2 \frac{d^{j/2}}{d^{-j}} \\
&\quad + \sum_{k \geq k^*} \frac{c_k^2 \lambda^2 k(k-1)}{(d-1)(1+\lambda^2)^{k/2}} \sum_{j=0}^{\lfloor \frac{k-2}{2} \rfloor} \binom{k-1}{2j} (\mathbf{w} \cdot \mathbf{w}^*)^{k-2j-1} (1 - (\mathbf{w} \cdot \mathbf{w}^*)^2)^j \|(\mathbf{w}^*)^{\perp \mathbf{w}}\|_2 \frac{d^{j/2}}{d^{-j}} \\
&\overset{(i)}{\lesssim} \sum_{k \geq k^*} \frac{c_k^2 k}{d^{k/4}} \sum_{j=0}^{\lfloor \frac{k-1}{2} \rfloor} \binom{k-1}{2j} d^{-(k-2j-1)/2} \frac{d^{j/2}}{d^{-j}} \\
&\quad + \sum_{k \geq k^*} \frac{c_k^2 k(k-1)}{d^{k/4+1/2}} \sum_{j=0}^{\lfloor \frac{k-2}{2} \rfloor} \binom{k-1}{2j} d^{-(k-2j-1)/2} \frac{d^{j/2}}{d^{-j}} \\
&\overset{(ii)}{\lesssim} \sum_{k \geq k^*} \frac{c_k^2 k}{d^{k/4}} \sum_{j=0}^{\lfloor \frac{k-1}{2} \rfloor} 2^k d^{-k/2+1/2} d^{j/2} + \sum_{k \geq k^*} \frac{c_k^2 k(k-1)}{d^{k/4+1/2}} \sum_{j=0}^{\lfloor \frac{k-2}{2} \rfloor} 2^k d^{-k/2+1/2} d^{j/2} \\
&\overset{(iii)}{\lesssim} \sum_{k \geq k^*} \left( \frac{c_k^2 k}{d^{k/4}} + \frac{c_k^2 k(k-1)}{d^{k/4+1/2}} \right) 2^k d^{-\frac{k}{2}+\frac{1}{2}+\frac{k}{4}} \\
&\lesssim \sum_{k \geq k^*} (k c_k)^2 2^k d^{-k/2} \lesssim \sum_{k \geq k^*} 4^k d^{-k/2} \lesssim 4^{k^*} d^{-k^*/2} ,
\end{aligned}
$$

where in $(i)$ we plugged in the value of $\lambda$ and $\nu_j^{(d-1)}$, $\nu_j^{(d+1)}$; in $(ii)$ we plugged in the value of $\mathbf{w} \cdot \mathbf{w}^* = 1/\sqrt{d}$; and in $(iii)$ we used the fact that $\binom{k}{j} \leq 2^k$. However, the magnitude of the signal from $|\mathcal{I}_1|$ is much smaller compared to the strength of the noise, $|\mathcal{I}_2|$, as by Cauchy-Schwarz, we

have

$$|\mathcal{I}_2| \overset{(iv)}{\leq} \left( \underset{(\mathbf{x},y)\sim\mathcal{D}}{\mathbf{E}}[(y - \sigma(\mathbf{w}^* \cdot \mathbf{x}))^2] \underset{\mathbf{x}\sim\mathcal{N}_d}{\mathbf{E}} \left[ \left( \sum_{k\geq k^*} c_k \langle \mathbf{He}_k(\mathbf{x}), \nabla\mathbf{T}_k(\mathbf{w}) \otimes (\mathbf{w}^*)^{\perp\mathbf{w}} \rangle \right)^2 \right] \right)^{\frac{1}{2}}$$

$$\leq \sqrt{\mathrm{OPT}} \left( \sum_{k\geq k^*} c_k^2 \|\nabla\mathbf{T}_k(\mathbf{w}) \otimes (\mathbf{w}^*)^{\perp\mathbf{w}}\|_F^2 \right)^{\frac{1}{2}}.$$

Note that in Lemma 12 of [DNGL23], it was proved that

$$\|\nabla\mathbf{T}_k(\mathbf{w}) \otimes (\mathbf{w}^*)^{\perp\mathbf{w}}\|_F^2$$

$$\lesssim \frac{k^2}{(1+\lambda^2)^k} \sum_{j=0}^{\lfloor\frac{k-1}{2}\rfloor} \binom{k-1}{2j} \lambda^{4j} \nu_j^{(d-1)} + \frac{\lambda^4 k^4}{d^2(1+\lambda^2)^k} \sum_{j=0}^{\lfloor\frac{k-2}{2}\rfloor} \binom{k-2}{2j} \lambda^{4j} \nu_j^{(d+1)}$$

$$\overset{(v)}{\lesssim} \frac{k^2 2^k}{d^{\frac{k}{2}}} + \frac{k^4 2^k}{d^{\frac{k}{2}+1}} \lesssim k^4 2^k d^{-k/2}.$$

where in $(v)$ we plugged in the value of $\lambda$ and $\nu_j^{(d-1)}$, $\nu_j^{(d+1)}$. Thus, it holds that

$$|\mathcal{I}_2| \lesssim \sqrt{\mathrm{OPT}} \left( \sum_{k\geq k^*} c_k^2 4^k d^{-\frac{k}{2}} \right)^{1/2} \lesssim \sqrt{\mathrm{OPT}} 2^{k^*} d^{-k^*/4}.$$

Finally, we remark that the equality holds at $(iv)$ when

$$y = \sigma(\mathbf{w}^* \cdot \mathbf{x}) + \frac{\sqrt{\mathrm{OPT}}}{\sqrt{\sum_{k\geq k^*} c_k^2 \|\nabla\mathbf{T}_k(\mathbf{w}) \otimes (\mathbf{w}^*)^{\perp\mathbf{w}}\|_F^2}} \left( \sum_{k\geq k^*} c_k \langle \mathbf{He}_k(\mathbf{x}), \nabla\mathbf{T}_k(\mathbf{w}) \otimes (\mathbf{w}^*)^{\perp\mathbf{w}} \rangle \right).$$

$\square$

### B.3 Comparison with [DPVLB24]

In [DPVLB24], the authors studied a milder noise model than the agnostic setting. In their model, the joint distribution on $(\mathbf{x}, y)$ is defined as $\mathbf{E}_y[y \mid \mathbf{x}] = \sigma(\mathbf{w}^* \cdot \mathbf{x}) + \xi(\mathbf{w}^* \cdot \mathbf{x})$, where $\xi : \mathbb{R} \mapsto \mathbb{R}$ is assumed to be known to the learner. Note that in this model the labels $y$ are independent of all the directions orthogonal to $\mathbf{w}^*$, i.e., the random vector $\mathbf{x}^{\perp\mathbf{w}^*}$ is independent of $y$. In comparison, in the agnostic setting, the distribution of the labels is $\mathbf{E}_y[y \mid \mathbf{x}] = \sigma(\mathbf{w}^* \cdot \mathbf{x}) + \xi'(\mathbf{x})$, where $\xi' : \mathbb{R}^d \mapsto \mathbb{R}$ is unknown and can depend arbitrarily on $\mathbf{x}$. In particular, the aforementioned independence with the directions orthogonal to $\mathbf{w}^*$ does not hold. As a result of their milder noise model, the authors can utilize information on the joint distribution to mitigate the corruption of the noise. This assumption is significantly weaker than agnostic noise.

In addition, instead of studying the information component $k^*$ defined as the first non-zero Hermite coefficient of the activation $\sigma$, [DPVLB24] considered the generative component of *the label $y$*, which is defined as $\bar{k}^* := \min_{k\geq 0}\{\lambda_k > 0\}$, where $\lambda_k := \sqrt{\mathbf{E}_y[\zeta_k^2]}$, and $\zeta_k(y) := \mathbf{E}_z[\mathrm{he}_k(z)|y]$. The main results in [DPVLB24] are twofold. First, they proved an SQ lower bound showing that under the setting aforementioned, any polynomial time SQ algorithm requires at least $O(d^{\bar{k}^*/2})$ samples to learn the hidden direction $\mathbf{w}^*$. Then, they provided an SQ algorithm using partial-trace operators that returns a vector $\widehat{\mathbf{w}}$ such that $(\widehat{\mathbf{w}} \cdot \mathbf{w}^*)^2 \geq 1 - \epsilon^2$ with $O(d^{\bar{k}^*/2} + d/\epsilon^2)$ samples, matching the SQ lower bound.

The algorithm proposed in [DPVLB24] can be described in short as follows: given joint distribution P, calculate $\zeta_{\bar{k}^*}(y)$ and transform the label $y$ to $\zeta_{\bar{k}^*}(y)$. Then, they applied tensor PCA on $\zeta_{\bar{k}^*}(y)\mathbf{He}_k(\mathbf{x})$, using the partial trace operator and tensor power iteration.

We note that the partial trace method for tensor PCA fails to find an initialization vector $\mathbf{w}^0$ such that $\mathbf{w}^0 \cdot \mathbf{w}^* \geq c$ for some positive absolute constant $c$ under agnostic noise. We explain this briefly below. For simplicity, let us consider $k$ being even. Note that for a $k$-tensor $\mathbf{T}$ with even $k$, the partial trace operator is defined by $\mathsf{PT}(\mathbf{T}) = \langle \mathbf{T}, \mathbf{I}^{\otimes(k-2)/2} \rangle$, where $\mathbf{I}$ is the identity matrix in $\mathbb{R}^{d\times d}$. The idea

in [DPVLB24] and [ADGM17] is to use the top eigenvector $\mathbf{v}_1 \in \mathbb{R}^d$ of $\mathsf{PT}(\mathbf{E}_{(\mathbf{x},y)\sim\mathcal{D}}[y\mathbf{He}_k(\mathbf{x})])$ as a warm-start. However, we show that under agnostic noise, this eigenvector $\mathbf{v}_1$ does not contain a strong enough signal to provide information of $\mathbf{w}^*$. Note that by definition of the partial trace operator, for any $\mathbf{v} \in \mathbb{B}_d$ we have

$$\mathbf{v}^\top \mathsf{PT}(\mathop{\mathbf{E}}_{(\mathbf{x},y)\sim\mathcal{D}}[y\mathbf{He}_k(\mathbf{x})])\mathbf{v} = \mathop{\mathbf{E}}_{(\mathbf{x},y)\sim\mathcal{D}}[y\mathbf{v}^\top \langle \mathbf{He}_k(\mathbf{x}), \mathbf{I}^{\otimes(k-2)/2}\rangle \mathbf{v}] \tag{18}$$

$$= \mathop{\mathbf{E}}_{(\mathbf{x},y)\sim\mathcal{D}}[y\langle \mathbf{He}_k(\mathbf{x}), \mathbf{I}^{\otimes(k-2)/2} \otimes \mathbf{v}^{\otimes 2}\rangle]$$

$$= \mathop{\mathbf{E}}_{\mathbf{x}\sim\mathcal{N}_d}[\sigma(\mathbf{w}^* \cdot \mathbf{x})\langle \mathbf{He}_k(\mathbf{x}), \mathbf{I}^{\otimes(k-2)/2} \otimes \mathbf{v}^{\otimes 2}\rangle]$$

$$+ \mathop{\mathbf{E}}_{(\mathbf{x},y)\sim\mathcal{D}}[(y - \sigma(\mathbf{w}^* \cdot \mathbf{x}))\langle \mathbf{He}_k(\mathbf{x}), \mathbf{I}^{\otimes(k-2)/2} \otimes \mathbf{v}^{\otimes 2}\rangle]$$

$$= c_k\langle \mathbf{w}^{*\otimes k}, \mathbf{I}^{\otimes(k-2)/2} \otimes \mathbf{v}^{\otimes 2}\rangle + \mathop{\mathbf{E}}_{(\mathbf{x},y)\sim\mathcal{D}}[(y - \sigma(\mathbf{w}^* \cdot \mathbf{x}))\langle \mathbf{He}_k(\mathbf{x}), \mathbf{I}^{\otimes(k-2)/2} \otimes \mathbf{v}^{\otimes 2}\rangle]. \tag{19}$$

The first term in the equality above equals $c_k\langle \mathbf{w}^{*\otimes k}, \mathbf{I}^{\otimes(k-2)/2} \otimes \mathbf{v}^{\otimes 2}\rangle = c_k(\mathbf{w}^* \cdot \mathbf{v})^2$, however, the second term can be as large as (by Cauchy-Schwarz):

$$\mathop{\mathbf{E}}_{(\mathbf{x},y)\sim\mathcal{D}}[(y - \sigma(\mathbf{w}^* \cdot \mathbf{x}))\langle \mathbf{He}_k(\mathbf{x}), \mathbf{I}^{\otimes(k-2)/2} \otimes \mathbf{v}^{\otimes 2}\rangle]$$

$$\overset{(i)}{\leq} \sqrt{\mathop{\mathbf{E}}_{(\mathbf{x},y)\sim\mathcal{D}}[(y - \sigma(\mathbf{w}^* \cdot \mathbf{x}))^2]\mathop{\mathbf{E}}_{\mathbf{x}\sim\mathcal{N}_d}[\langle \mathbf{He}_k(\mathbf{x}), \mathbf{I}^{\otimes(k-2)/2} \otimes \mathbf{v}^{\otimes 2}\rangle^2]}$$

$$\leq \sqrt{\mathrm{OPT}}\|\mathbf{I}^{\otimes(k-2)/2} \otimes \mathbf{v}^{\otimes 2}\|_F = \sqrt{\mathrm{OPT}}d^{(k-2)/4}.$$

As suggested by inequality $(i)$ above, let $\mathbf{v}$ be any unit vector orthogonal to $\mathbf{w}^*$, consider an agnostic noise model

$$y = \sigma(\mathbf{w}^* \cdot \mathbf{x}) + (\sqrt{\mathrm{OPT}}/d^{(k-2)/4})\langle \mathbf{He}_k(\mathbf{x}), \mathbf{I}^{\otimes(k-2)/2} \otimes \mathbf{v}^{\otimes 2}\rangle,$$

then $\mathbf{E}_{(\mathbf{x},y)\sim\mathcal{D}}[(y - \sigma(\mathbf{w}^* \cdot \mathbf{x}))^2] = \mathrm{OPT}$. However, as we can see from Equation (18), it holds that

$$(\mathbf{w}^*)^\top \mathsf{PT}(\mathop{\mathbf{E}}_{(\mathbf{x},y)\sim\mathcal{D}}[y\mathbf{He}_k(\mathbf{x})])\mathbf{w}^*$$

$$= c_k + \frac{\sqrt{\mathrm{OPT}}}{d^{(k-2)/4}} \mathop{\mathbf{E}}_{(\mathbf{x},y)\sim\mathcal{D}}[\langle \mathbf{He}_k(\mathbf{x}), \mathbf{I}^{\otimes(k-2)/2} \otimes \mathbf{v}^{\otimes 2}\rangle\langle \mathbf{He}_k(\mathbf{x}), \mathbf{I}^{\otimes(k-2)/2} \otimes (\mathbf{w}^*)^{\otimes 2}\rangle]$$

$$= c_k + \frac{\sqrt{\mathrm{OPT}}}{d^{(k-2)/4}} \langle \mathrm{Sym}(\mathbf{I}^{\otimes(k-2)/2} \otimes \mathbf{v}^{\otimes 2}), \mathrm{Sym}(\mathbf{I}^{\otimes(k-2)/2} \otimes (\mathbf{w}^*)^{\otimes 2})\rangle$$

$$\overset{(ii)}{\lesssim} c_k + \frac{\sqrt{\mathrm{OPT}}}{d^{(k-2)/4}}d^{(k-4)/2} = c_k + \sqrt{\mathrm{OPT}}d^{k/4-3/2},$$

where $(ii)$ comes from the fact that for any permutation $\pi$ such that $\{\pi(1), \pi(2)\} \cap \{1, 2\} \neq \varnothing$, since $\mathbf{v} \perp \mathbf{w}^*$, it holds

$$\sum_{i_1,\ldots,i_k} \mathbf{v}_{i_1}\mathbf{v}_{i_2}\mathbf{w}^*_{i_{\pi(1)}}\mathbf{w}^*_{i_{\pi(2)}}\mathbf{I}_{i_3,i_4}\ldots\mathbf{I}_{i_{k-1},i_k}\mathbf{I}_{i_{\pi(3)},i_{\pi(4)}}\ldots\mathbf{I}_{i_{\pi(k-1)},i_{\pi(k)}} = 0;$$

and on the other hand, when $\{\pi(1), \pi(2)\} \cap \{1, 2\} = \varnothing$, then

$$\sum_{i_1,\ldots,i_k=1}^d \mathbf{v}_{i_1}\mathbf{v}_{i_2}\mathbf{w}^*_{i_{\pi(1)}}\mathbf{w}^*_{i_{\pi(2)}}\mathbf{I}_{i_3,i_4}\ldots\mathbf{I}_{i_{k-1},i_k}\mathbf{I}_{i_{\pi(3)},i_{\pi(4)}}\ldots\mathbf{I}_{i_{\pi(k-1)},i_{\pi(k)}} \leq d^{(k-4)/2}.$$

To see this, note that there exists a chain of identity matrices

$$\mathbf{I}_{i_{\pi(a_1)},i_{\pi(a_1+1)}}\mathbf{I}_{i_{j_1},i_{j_1+1}}\mathbf{I}_{i_{\pi(a_2)},i_{\pi(a_2+1)}}\mathbf{I}_{i_{j_2},i_{j_2+1}}\cdots\begin{cases}\mathbf{I}_{i_{\pi(a_m)},i_{\pi(a_m+1)}}\\ \mathbf{I}_{i_{j_m},i_{j_m+1}}\end{cases}$$

such that $\pi(a_1) = 1, \pi(a_1 + 1) = j_1, j_1 + 1 = \pi(a_2), \pi(a_2 + 1) = j_2\ldots$ until we have $i_{\pi(a_m+1)} \in \{i_1, i_2\}$ or $i_{j_m+1} \in \{i_{\pi(1)}, i_{\pi(2)}\}$. For the latter case, it implies that

$$\sum \mathbf{v}_{i_1}(\mathbf{w}^*)_{i_{j_m+1}}\mathbf{I}_{i_{\pi(a_1)},i_{\pi(a_1+1)}}\mathbf{I}_{i_{j_1},i_{j_1+1}}\mathbf{I}_{i_{\pi(a_2)},i_{\pi(a_2+1)}}\mathbf{I}_{i_{j_2},i_{j_2+1}}\cdots\mathbf{I}_{i_{j_m},i_{j_m+1}} = \sum_{i_1}^d \mathbf{v}_{i_1}(\mathbf{w}^*)_{i_1} = 0.$$

Therefore, only the first case is interesting. Since $\pi(a_1) = 1$, and $\pi$ is a bijection, hence $\pi(a_m+1) \neq 1$ therefore the only possible case would be $i_{\pi(a_m+1)} = i_2$, then we have

$$\sum \mathbf{v}_{i_1} \mathbf{v}_{i_{\pi(a_m+1)}} \mathbf{I}_{i_{\pi(a_1)}, i_{\pi(a_1+1)}} \mathbf{I}_{i_{j_1}, i_{j_1+1}} \mathbf{I}_{i_{\pi(a_2)}, i_{\pi(a_2+1)}} \mathbf{I}_{i_{j_2}, i_{j_2+1}} \cdots \mathbf{I}_{i_{\pi(a_m)}, i_{\pi(a_m+1)}} = \sum_{i_1} \mathbf{v}_{i_1}^2 = 1.$$

Continue the discussion for $\mathbf{v}_{i_2}$, $\mathbf{w}^*_{i_{\pi(1)}}$ and $\mathbf{w}^*_{i_{\pi(2)}}$, and let $\{j_5, j_6, \ldots, j_{k-1}, j_k\} = \{i_1, \ldots, i_k\} - \{i_1, i_2, i_{\pi(1)}, i_{\pi(2)}\}$, we get that

$$\sum_{i_1, \ldots, i_k = 1}^{d} \mathbf{v}_{i_1} \mathbf{v}_{i_2} \mathbf{w}^*_{i_{\pi(1)}} \mathbf{w}^*_{i_{\pi(2)}} \mathbf{I}_{i_3, i_4} \cdots \mathbf{I}_{i_{k-1}, i_k} \mathbf{I}_{i_{\pi(3)}, i_{\pi(4)}} \cdots \mathbf{I}_{i_{\pi(k-1)}, i_{\pi(k)}} \leq \sum_{j_5, \ldots, j_k = 1}^{d} \mathbf{I}_{j_5, j_6}^2 \cdots \mathbf{I}_{j_{k-1}, j_k}^2$$
$$= d^{(k-4)/2}.$$

Similarly, for any unit vector $\mathbf{u} \perp \mathbf{v}$ and $\mathbf{u} \perp \mathbf{w}^*$, it holds

$$\mathbf{u}^\top \mathsf{PT}(\mathop{\mathbf{E}}_{(\mathbf{x}, y) \sim \mathcal{D}}[y \mathbf{He}_k(\mathbf{x})]) \mathbf{u} \lesssim \sqrt{\mathrm{OPT}} d^{k/4 - 3/2}.$$

But one can also show that

$$\mathbf{v}^\top \mathsf{PT}(\mathop{\mathbf{E}}_{(\mathbf{x}, y) \sim \mathcal{D}}[y \mathbf{He}_k(\mathbf{x})]) \mathbf{v} = \frac{\sqrt{\mathrm{OPT}}}{d^{(k-2)/4}} \mathop{\mathbf{E}}_{(\mathbf{x}, y) \sim \mathcal{D}}[\langle \mathbf{He}_k(\mathbf{x}), \mathbf{I}^{\otimes (k-2)/2} \otimes \mathbf{v}^{\otimes 2} \rangle^2]$$
$$= \frac{\sqrt{\mathrm{OPT}}}{d^{(k-2)/4}} \|\mathrm{Sym}(\mathbf{I}^{\otimes (k-2)/2} \otimes \mathbf{v}^{\otimes 2})\|_F^2$$
$$\approx \sqrt{\mathrm{OPT}} d^{k/4 - 1/2};$$

This implies that to guarantee that $\mathbf{w}^*$ is the unique top eigenvector, it has to be that $\mathrm{OPT} \lesssim c_k^2 d^{-(k-2)/2}$. Therefore, the noise term completely buries the signal $(\mathbf{w}^* \cdot \mathbf{v})^2$ unless $\mathrm{OPT} \lesssim d^{-(k-2)/2}$, which is unrealistic to assume.

### B.4 Remarks on Tensor PCA

We summarize some historical bits of Tensor PCA. [RM14] proposed the following 'spiked' tensor PCA problem: given a $k$-tensor of the form[3]

$$\mathbf{T} = \tau \mathbf{v}^{\otimes k} + \mathbf{A}, \tag{PCA-S}$$

where $\mathbf{A}$ is a $k$-tensor with i.i.d. standard Gaussian entries, recover the planted vector $\mathbf{v}$. The 'single-observation' model is equivalent (in law) to the following 'multi-observation' model([BAGJ20]): given $n$ i.i.d. copies $\mathbf{T}^{(i)} = \tau' \mathbf{v}^{\otimes k} + \mathbf{A}^{(i)}$ with $\tau' = \tau/\sqrt{n}$, recover $\mathbf{v}$ using the empirical estimation:

$$\widehat{\mathbf{T}} = \tau' \mathbf{v}^{\otimes k} + \frac{1}{n} \sum_{i=1}^{n} \mathbf{A}^{(i)}. \tag{PCA-M}$$

In [RM14], it has been shown that for model (PCA-S), it is information-theoretically impossible to recover $\mathbf{v}$ when $\tau < (\beta_k^* - o_d(1))\sqrt{d}$, for some real constant $\beta_k^*$; however, there also exists a constant $\beta_k'$ such that the information-theoretic optimal threshold for $\tau$ is $\tau > (\beta_k' + o_d(1))\sqrt{d}$ (see also [DH21]). However, there is a huge statistical-computational gap for solving tensor PCA problems and it is conjectured impossible to solve (PCA-S) when $\tau \lesssim d^{k/4}$ for $k \geq 3$ [RM14, DH21]. For multi-observation model (PCA-M), this thresholds translates to a sample complexity of $n \gtrsim d^{k/2}$ when $\tau' = O(1)$.

[RM14] proposed the tensor unfolding algorithm that recovers $\mathbf{v}$ in (PCA-S) when $\tau \gtrsim d^{\lceil k/2 \rceil/2}$, i.e., for (PCA-M), the required sample complexity is $\Omega(d^{\lceil k/2 \rceil})$. However, it is conjectured in [RM14] that the tensor unfolding algorithm can actually deal with $\tau \gtrsim d^{k/4}$.

---

[3]Note that [RM14] takes a different normalization and in the notation of [RM14], it holds $\beta \approx \tau/\sqrt{d}$.

Note that the unfolding algorithm requires $\tau \gtrsim d$ when $k = 3$. Many papers are devoted to improving from $\tau \gtrsim d$ to $\tau \gtrsim d^{3/4}$ and reducing the runtime and memory cost. To name a few, [HSS15, HSSS16] used Sum-of-Squares algorithms with partial trace operators to achieve the goal within $O(d^3)$ runtime; [ADGM17] also used partial trace operators, but instead of using SOS algorithms, they injected noise to smooth the landscape of the loss.

Perhaps an interesting aspect of our paper is that the unfolding algorithm can deal with stronger noise (compared to the Gaussian noise $\mathbf{A}$, our noise is very heavy-tailed) and is more robust to the noise, as partial trace operator does not work under our agnostic noise. However, this might be attributed to the special structure of the noisy chow tensor.

## C  Additional Preliminaries

### C.1  Elementary Tensor Algebra

We review basic definitions and elementary tensor algebra used throughout the paper. For any positive integer $k$, a $k$-tensor $\mathbf{T}$ is defined as a multilinear real function that maps $k$ vectors to a real number, i.e., $\mathbf{T} : \mathbb{R}^d \times \cdots \times \mathbb{R}^d \to \mathbb{R}$. A $k$-tensor $\mathbf{T}$ can also be viewed as a multidimensional array, where each entry is associated with $k$ indices $i_1, \ldots, i_k$ in $[d]$ and equals $\mathbf{T}_{i_1,\ldots,i_k} = \mathbf{T}(\mathbf{e}_{i_1}, \ldots, \mathbf{e}_{i_k})$.

Given a vector $\mathbf{w} \in \mathbb{R}^d$, $\mathbf{w} = (\mathbf{w}_1, \ldots, \mathbf{w}_d)^\top$, the $k$-product-tensor $\mathbf{w}^{\otimes k}$ is defined by

$$(\mathbf{w}^{\otimes k})_{i_1,\ldots,i_k} := \mathbf{w}_{i_1} \mathbf{w}_{i_2} \cdots \mathbf{w}_{i_k}, \ \forall i_1, \ldots, i_k \in [d].$$

Given a $k$-tensor $\mathbf{T}$ and an $l$-tensor $\mathbf{T}'$ (with $l \geq k$), the inner product (or contraction) between $\mathbf{T}$ and $\mathbf{T}'$ is defined by $(\langle \mathbf{T}, \mathbf{T}' \rangle)_{i_{k+1},\ldots,i_l} := \sum_{i_1,i_2,\ldots,i_k}^d (\mathbf{T})_{i_1,i_2,\ldots,i_k} (\mathbf{T}')_{i_1,i_2,\ldots,i_k,i_{k+1},\ldots,i_l}$. In other words, the inner product of a $k$-tensor and an $l$-tensor yields an $(l-k)$-tensor. When $k = l$, the inner product between $\mathbf{T}$ and $\mathbf{T}'$ is a real number. In particular, for vectors $\mathbf{w}, \mathbf{v} \in \mathbb{R}^d$, and any $k \geq 1$,

$$\langle \mathbf{w}^{\otimes k}, \mathbf{v}^{\otimes k} \rangle = (\mathbf{w} \cdot \mathbf{v})^k, \tag{20}$$

where $\mathbf{w} \cdot \mathbf{v}$ is the standard inner product between vectors.

A tensor $\mathbf{T}$ is symmetric if $(\mathbf{T})_{\ldots,i,\ldots,j,\ldots} = (\mathbf{T})_{\ldots,j,\ldots,i,\ldots}$. We can symmetrize a $k$-tensor $\mathbf{T}$ by summing up all its copies with permuted indices $i_1, \ldots, i_k$ and dividing the sum by $k!$, which is the total number of permutations. We define the symmetrization operator of a $k$-tensor $\mathbf{T}$ by

$$(\mathrm{Sym}(\mathbf{T}))_{i_1,\ldots,i_k} = \frac{1}{k!} \sum_{\pi \in \mathcal{S}_k} (\mathbf{T})_{i_{\pi(1)},\ldots,i_{\pi(k)}}.$$

When $k = 2$, this reduces to the symmetrization of a square matrix: $\mathrm{Sym}(\mathbf{T}) = (1/2)(\mathbf{T} + \mathbf{T}^\top)$.

Let us provide some useful observations about tensor algebra.

**Fact C.1.** *Let $\mathbf{w} \in \mathbb{R}^d$ and let $\mathbf{T}$ be any $k$-tensor. Then,*

*(1) The inner product between $\mathbf{w}^{\otimes k}$ and $\mathrm{Sym}(\mathbf{T})$ is equal to the inner product between $\mathbf{w}^{\otimes k}$ and $\mathbf{T}$:*

$$\langle \mathbf{w}^{\otimes k}, \mathrm{Sym}(\mathbf{T}) \rangle = \langle \mathbf{w}^{\otimes k}, \mathbf{T} \rangle. \tag{21}$$

*(2) If $\mathbf{T}$ is a symmetric tensor,*

$$\nabla(\langle \mathbf{T}, \mathbf{w}^{\otimes k} \rangle) = k \langle \mathbf{T}, \mathbf{w}^{\otimes k-1} \rangle \tag{22}$$

*Proof.* To prove the first statement in Fact C.1, note that direct calculation yields:

$$
\begin{aligned}
\langle \mathbf{w}^{\otimes k}, \mathrm{Sym}(\mathbf{T}) \rangle &= \sum_{i_1,\ldots,i_k} \mathbf{w}_{i_1} \cdots \mathbf{w}_{i_k} \frac{1}{k!} \sum_{\pi \in \mathcal{S}} (\mathbf{T})_{i_{\pi(1)},\ldots,i_{\pi(k)}} \\
&= \frac{1}{k!} \sum_{\pi \in \mathcal{S}} \sum_{i_1,\ldots,i_k} \mathbf{w}_{i_{\pi(1)}} \cdots \mathbf{w}_{i_{\pi(k)}} (\mathbf{T})_{i_{\pi(1)},\ldots,i_{\pi(k)}} \\
&= \sum_{i_1,\ldots,i_k} \mathbf{w}_{i_1} \cdots \mathbf{w}_{i_k} (\mathbf{T})_{i_1,\ldots,i_k} = \langle \mathbf{w}^{\otimes k}, \mathbf{T} \rangle.
\end{aligned}
$$

Next for the second statement, let $f(\mathbf{w}) = \langle \mathbf{T}, \mathbf{w}^{\otimes k} \rangle$ where $\mathbf{T}$ is a $k$-tensor. Then, the partial derivative of $f$ w.r.t. $\mathbf{w}_j$ is

$$
\frac{\partial f(\mathbf{w})}{\partial \mathbf{w}_j} = \frac{\partial}{\partial \mathbf{w}_j} \langle \mathbf{T}, \mathbf{w}^{\otimes k} \rangle = \frac{\partial}{\partial \mathbf{w}_j} \left( \sum_{i_1, \dots, i_k} (\mathbf{T})_{i_1, \dots, i_k} \mathbf{w}_{i_1} \cdots \mathbf{w}_{i_k} \right)
$$

$$
= \sum_{i_2, \dots, i_k} (\mathbf{T})_{j, i_2, \dots, i_k} \mathbf{w}_{i_2} \cdots \mathbf{w}_{i_k} + \sum_{i_1, i_3 \dots, i_k} (\mathbf{T})_{i_1, j, i_3, \dots, i_k} \mathbf{w}_{i_1} \mathbf{w}_{i_3} \cdots \mathbf{w}_{i_k} +
$$

$$
\cdots + \sum_{i_1, \dots, i_{k-1}} (\mathbf{T})_{i_1, \dots, i_{k-1}, j} \mathbf{w}_{i_1} \cdots \mathbf{w}_{i_{k-1}}.
$$

Thus if $\mathbf{T}$ is symmetric, then $\nabla(\langle \mathbf{T}, \mathbf{w}^{\otimes k} \rangle) = k \langle \mathbf{T}, \mathbf{w}^{\otimes k-1} \rangle$. $\qquad\square$

## C.2 Hermite Polynomials and Hermite Tensors

We make use of the normalized probabilist's Hermite polynomial, defined by

$$
\mathrm{he}_k(z) = \frac{(-1)^k}{\sqrt{k!}} \exp\left( \frac{z^2}{2} \right) \frac{\mathrm{d}^k}{\mathrm{d}z^k} \exp\left( -\frac{z^2}{2} \right).
$$

We will heavily use the following properties of the normalized Hermite polynomials [AS68]:

**Fact C.2.** *Hermite polynomials satisfy the following properties:*

1. *(Orthonormality)* $\mathbf{E}_{z \sim \mathcal{N}(0,1)}[\mathrm{he}_k(z)\mathrm{he}_j(z)] = \mathbb{1}\{k = j\}$.

2. *(Recurrence)* $\mathrm{he}'_k(z) = \sqrt{k}\,\mathrm{he}_{k-1}(z)$.

Given a vector $\mathbf{x} \in \mathbb{R}^d$, we can then define the (normalized) Hermite multivariate tensor by [Rah17]:

$$
(\mathbf{He}_k(\mathbf{x}))_{i_1, \dots, i_k} := \left( \frac{\alpha_1! \dots \alpha_d!}{k!} \right)^{1/2} \mathrm{he}_{\alpha_1}(\mathbf{x}_1) \dots \mathrm{he}_{\alpha_d}(\mathbf{x}_d), \text{ where } \alpha_j = \sum_{l=1}^{k} \mathbb{1}\{i_l = j\}, \; \forall j \in [d].
$$

For Hermite tensors, we have the following facts:

**Fact C.3.** *Let $\mathbf{x}$ be a $d$-dimensional standard Gaussian random vector.*

1. *For any $k$-tensor $\mathbf{A}$ and $j$-tensor $\mathbf{B}$,*

$$
\mathop{\mathbf{E}}_{\mathbf{x} \sim \mathcal{N}_d}[\langle \mathbf{He}_k(\mathbf{x}), \mathbf{A} \rangle \langle \mathbf{He}_j(\mathbf{x}), \mathbf{B} \rangle] = \mathbb{1}\{k = j\} \langle \mathrm{Sym}(\mathbf{A}), \mathrm{Sym}(\mathbf{B}) \rangle.
$$

2. *For any $\mathbf{w} \in \mathbb{R}^d$ such that $\|\mathbf{w}\|_2 = 1$, $\mathrm{he}_k(\mathbf{w} \cdot \mathbf{x}) = \langle \mathbf{He}(\mathbf{x}), \mathbf{w}^{\otimes k} \rangle$.*

## C.3 Loss and Gradients

We consider the $L_2^2$ (or square) loss, defined by

$$
\mathcal{L}_2^\sigma(\mathbf{w}) := \mathop{\mathbf{E}}_{(\mathbf{x},y) \sim \mathcal{D}}[(\sigma(\mathbf{w} \cdot \mathbf{x}) - y)^2].
$$

Let $\mathbf{w}^* \in \mathrm{argmin}_{\mathbf{w} \in \mathbb{S}^{d-1}} \mathcal{L}_2^\sigma(\mathbf{w})$, and denote the minimum value of the $L_2^2$ loss by $\mathrm{OPT} := \min_{\mathbf{w} \in \mathbb{S}^{d-1}} \mathcal{L}_2^\sigma(\mathbf{w})$. Furthermore, let us define the "noiseless" $L_2^2$ loss by

$$
\mathcal{L}_2^{*\sigma}(\mathbf{w}) := \mathop{\mathbf{E}}_{\mathbf{x} \sim \mathcal{N}_d}[(\sigma(\mathbf{w} \cdot \mathbf{x}) - \sigma(\mathbf{w}^* \cdot \mathbf{x}))^2]. \tag{23}
$$

We observe that the $L_2^2$ loss is determined by the inner product between $\mathbf{w}$ and $\mathbf{w}^*$, therefore, to obtain error $O(\mathrm{OPT}) + \epsilon$, it suffices to minimize the angle between $\mathbf{w}$ and $\mathbf{w}^*$. Concretely, we have:

**Claim C.4.** *Let $\mathbf{w} \in \mathbb{R}^d$ be a unit vector. Then, the $L_2^2$ loss $\mathcal{L}_2^\sigma(\mathbf{w})$ satisfies:*

$$
\mathcal{L}_2^\sigma(\mathbf{w}) \leq 2\mathrm{OPT} + 4\left( 1 - \sum_{k \geq k^*} c_k^2 (\mathbf{w} \cdot \mathbf{w}^*)^k \right).
$$

*Proof.* Recalling that the activation $\sigma$ is normalized so that $\mathbf{E}_{\mathbf{x}\sim\mathcal{N}_d}[\sigma^2(\mathbf{w}\cdot\mathbf{x})] = \mathbf{E}_{\mathbf{x}\sim\mathcal{N}_d}[\sigma^2(\mathbf{w}^*\cdot\mathbf{x})] = 1$, we can simplify the $L_2^2$ loss to

$$\mathcal{L}_2^\sigma(\mathbf{w}) = 2\Big(1 - \underset{(\mathbf{x},y)\sim\mathcal{D}}{\mathbf{E}}[y\sigma(\mathbf{w}\cdot\mathbf{x})]\Big).$$

The noiseless $L_2^2$ loss admits the following decomposition:

$$\begin{aligned}
\mathcal{L}_2^{*\sigma}(\mathbf{w}) &= 2\Big(1 - \underset{\mathbf{x}\sim\mathcal{N}_d}{\mathbf{E}}[\sigma(\mathbf{w}\cdot\mathbf{x})\sigma(\mathbf{w}^*\cdot\mathbf{x})]\Big) \\
&= 2\Big(1 - \underset{\mathbf{x}\sim\mathcal{N}_d}{\mathbf{E}}\Big[\sum_{k\geq k^*}c_k\langle\mathbf{He}_k(\mathbf{x}),\mathbf{w}^{\otimes k}\rangle\sum_{k'\geq k^*}c_{k'}\langle\mathbf{He}_{k'}(\mathbf{x}),\mathbf{w}^{*\otimes k'}\rangle\Big]\Big) \\
&= 2\Big(1 - \sum_{k\geq k^*}c_k^2(\mathbf{w}\cdot\mathbf{w}^*)^k\Big),
\end{aligned}$$  (24)

where in the last equality we used the orthonormality property of Hermite tensors (Fact C.3). Further, using Young's inequality and the definitions of OPT and $\mathcal{L}_2^{*\sigma}(\mathbf{w})$, we also have $\mathcal{L}_2^\sigma(\mathbf{w}) \leq 2\text{OPT} + 2\mathcal{L}_2^{*\sigma}(\mathbf{w})$, which combined with Equation (24) leads to

$$\mathcal{L}_2^\sigma(\mathbf{w}) \leq 2\text{OPT} + 4\Big(1 - \sum_{k\geq k^*}c_k^2(\mathbf{w}\cdot\mathbf{w}^*)^k\Big).$$

$\square$

**Remark C.5.** *Equation* (24) *suggests that even in the realizable case, some assumption on boundedness of $\sum_{k\geq k^*}kc_k^2$ (see Assumption 1(iii)) may be necessary to have a nontrivial bound on the $L_2^2$ loss. Consider an algorithm that outputs a vector $\mathbf{w}$ such that $\mathbf{w}\cdot\mathbf{w}^* = 1-\alpha$ for some $\alpha \in (0,1)$ (if $\alpha = 0$, $\mathbf{w} = \mathbf{w}^*$ since both vectors are on the unit sphere). Since $\sum_{k\geq k^*}c_k^2 = 1$, we can also write $\mathcal{L}_2^{*\sigma}(\mathbf{w}) = 2\sum_{k\geq k^*}c_k^2(1-(1-\alpha)^k)$. For $k = \Omega(1/\alpha)$, $1-(1-\alpha)^k \approx k\alpha$. Thus, if we want an algorithm that works generically for any target accuracy, $\sum_{k\geq k^*}kc_k^2$ ought to be bounded.*

**Remark C.6.** *Even though for $\|\mathbf{w}\|_2 = 1$ we have $\text{he}_k(\mathbf{w}\cdot\mathbf{x}) = \langle\mathbf{He}_k(\mathbf{x}),\mathbf{w}^{\otimes k}\rangle$, the gradients with respect to $\mathbf{w}$ of these two functions are different in general. For example, for $k = 2$, we have*

$$\langle\mathbf{He}_2(\mathbf{x}),\mathbf{w}^{\otimes 2}\rangle = (1/\sqrt{2})((\mathbf{w}\cdot\mathbf{x})^2 - \|\mathbf{w}\|_2^2) \quad and \quad \text{he}_2(\mathbf{w}\cdot\mathbf{x}) = (1/\sqrt{2})((\mathbf{w}\cdot\mathbf{x})^2 - 1),$$

*which are equal in function value, but*

$$\begin{aligned}
\nabla\langle\mathbf{He}_2(\mathbf{x}),\mathbf{w}^{\otimes 2}\rangle &= \sqrt{2}((\mathbf{w}\cdot\mathbf{x})\mathbf{x} - \mathbf{w}) = 2\langle\mathbf{He}_2(\mathbf{x}),\mathbf{w}\rangle, \\
\nabla\text{he}_2(\mathbf{x}) &= \sqrt{2}(\mathbf{w}\cdot\mathbf{x})\mathbf{x} = \sqrt{2}\text{he}_1(\mathbf{w}\cdot\mathbf{x})\mathbf{x},
\end{aligned}$$

*are different. In particular, for the derivative of the left-hand side of Equation (24) to be equal to the derivative of its right-hand side, we need to use the tensor form of Hermite polynomials, because to ensure interchangeability of differentiation and summation, the sequence needs to be uniformly convergent. Note that $\mathbf{E}_{\mathbf{x}\sim\mathcal{N}_d}[\sum_{k\geq k^*}c_k\langle\mathbf{He}_k(\mathbf{x}),\mathbf{w}^{\otimes k}\rangle\sum_{k'\geq k^*}c_{k'}\langle\mathbf{He}_{k'}(\mathbf{x}),\mathbf{w}^{*\otimes k'}\rangle]$ converges to $\sum_{k\geq k^*}c_k^2(\mathbf{w}\cdot\mathbf{w})^k$ uniformly for all $\mathbf{w}\in\mathbb{R}^d$, but the sequence $\mathbf{E}_{\mathbf{x}\sim\mathcal{N}_d}[\sum_{k\geq k^*}c_k\text{he}_k(\mathbf{w}\cdot\mathbf{x})\sum_{k'\geq k^*}c_{k'}\text{he}_{k'}(\mathbf{w}^*\cdot\mathbf{x})]$ converges to $\sum_k c_k^2(\mathbf{w}\cdot\mathbf{w})^k$ only when $\|\mathbf{w}\|_2 = 1$, since it requires $\mathbf{w}\cdot\mathbf{x}\sim\mathcal{N}(0,1)$ to ensure that $\mathbf{E}_{\mathbf{x}\sim\mathcal{N}_d}[\text{he}_k(\mathbf{w}\cdot\mathbf{x})\text{he}_j(\mathbf{w}^*\cdot\mathbf{x})] = \mathbb{1}\{k=j\}(\mathbf{w}\cdot\mathbf{w}^*)^k$.*

As observed in Remark C.6, the gradients of $\text{he}_k(\mathbf{w}\cdot\mathbf{x})$ and $\langle\mathbf{He}_k(\mathbf{x}),\mathbf{w}^{\otimes k}\rangle$ are different in general. Throughout the paper, we will be taking the gradient with respect to the tensor form of $\sigma(\mathbf{w}\cdot\mathbf{x})$; in other words, $\nabla\sigma(\mathbf{w}\cdot\mathbf{x}) = \nabla(\sum_{k\geq k^*}c_k\langle\mathbf{He}_k(\mathbf{x}),\mathbf{w}^{\otimes k}\rangle)$.

## D   Full Version of Section 2

In this section, we show how to get an initial parameter vector $\mathbf{w}^0$ such that $\mathbf{w}^0\cdot\mathbf{w}^* = 1-\epsilon_0$ for some small constant $\epsilon_0$. The main technique is a tensor PCA algorithm that finds the principal

component of a noisy degree-$k$-Chow tensor for any $k \geq k^*$, as long as $\mathrm{OPT} \lesssim c_k^2$. Such a degree-$k$ Chow tensor is defined by $\mathbf{C}_k = \mathbf{E}_{(\mathbf{x},y)\sim\mathcal{D}}[y\mathbf{He}_k(\mathbf{x})]$, and we denote its noiseless counterpart by

$$\mathbf{C}_k^* = \mathop{\mathbf{E}}_{\mathbf{x}\sim\mathcal{N}_d}[\sigma(\mathbf{w}^*\cdot\mathbf{x})\mathbf{He}_k(\mathbf{x})] = \mathop{\mathbf{E}}_{\mathbf{x}\sim\mathcal{N}_d}\left[\sum_{j\geq k^*} c_j\langle\mathbf{He}_j(\mathbf{x}),\mathbf{w}^{*\otimes j}\rangle\mathbf{He}_k(\mathbf{x})\right].$$

Furthermore, let us denote the difference between $\mathbf{C}_k$ and $\mathbf{C}_k^*$ by

$$\mathbf{H}_k := \mathbf{C}_k - \mathbf{C}_k^* = \mathop{\mathbf{E}}_{(\mathbf{x},y)\sim\mathcal{D}}[(y-\sigma(\mathbf{w}^*\cdot\mathbf{x}))\mathbf{He}_k(\mathbf{x})].$$

Note that since $\mathbf{He}_k(\mathbf{x})$ is a symmetric tensor for any $\mathbf{x}$, all $\mathbf{C}_k$, $\mathbf{C}_k^*$ and $\mathbf{H}_k$ are symmetric tensors.

We use the following matrix unfolding operator that maps a $k$-tensor to a matrix in $\mathbb{R}^{d^l \times d^{k-l}}$. Concretely, given a $k$-tensor $\mathbf{T}$, we define:

$$\mathsf{Mat}_{(l,k-l)}(\mathbf{T})_{i_1+(i_2-1)d+\cdots+(i_l-1)d^{l-1},\, j_1+\cdots+(j_{k-l}-1)d^{k-l-1}} := (\mathbf{T})_{i_1,i_2,\ldots,i_l,j_1,\ldots,j_{k-l}}$$

for all $i_1,\ldots,i_l,j_1,\ldots,j_{k-l}\in[d]$.

For notational convenience, we also define the 'vectorize' operator and 'tensorize' operator, which map a vector $\mathbf{v}\in\mathbb{R}^{d^l}$ to an $l$-tensor for any integer $l$, and vice versa. In detail,

$$\mathsf{Tensor}(\mathbf{v})_{i_1,\ldots,i_l} := \mathbf{v}_{i_1+(i_2-1)d+\cdots+(i_l-1)d^{l-1}},\ \forall i_1,\ldots,i_l\in[d];$$

and conversely, we define

$$\mathsf{Vec}(\mathbf{v}^{\otimes l})_{i_1+(i_2-1)d+\cdots+(i_l-1)d^{l-1}} := \mathbf{v}_{i_1}\mathbf{v}_{i_2}\ldots\mathbf{v}_{i_l},\ \forall i_1,\ldots,i_l\in[d].$$

Finally, given a vector $\mathbf{v}\in\mathbb{R}^{d^l}$, we can also convert this vector to a matrix of size $\mathbb{R}^{d\times d^{l-1}}$:

$$\mathsf{Mat}_{(1,l-1)}(\mathbf{v})_{i,j_1,\ldots,j_{l-1}} = \mathbf{v}_{i+(j_1-1)d+\cdots+(j_{l-1}-1)d^{l-1}},\ \forall i,j_1,\ldots,j_{l-1}\in[d].$$

Some simple facts on the algebra of the unfolded matrix are in order.

**Fact D.1.** *Let $\mathbf{T}$ be a symmetric $k$-tensor, and let $\mathbf{r}\in\mathbb{R}^{d^{k-l}}$, $\mathbf{v}\in\mathbb{R}^{d^l}$. Then*

1. *For any index $i\in[d^l]$,*

$$(\mathsf{Mat}_{(l,k-l)}(\mathbf{T})\mathbf{r})_i = \left\langle \mathbf{T}, \mathbf{e}_{i_1'}\otimes\ldots\otimes\mathbf{e}_{i_l'}\otimes\mathsf{Tensor}(\mathbf{r})\right\rangle, \tag{25}$$

   *where $i_1',\ldots,i_l'\in[d]$ satisfies $i = i_1'+(i_2'-1)d+\cdots+(i_l'-1)d^{l-1}$.*

2. *For any index $j\in[d^{k-l}]$,*

$$(\mathsf{Mat}_{(l,k-l)}(\mathbf{T})^\top\mathbf{v})_j = \left\langle \mathbf{T}, \mathsf{Tensor}(\mathbf{v})\otimes\mathbf{e}_{j_1'}\otimes\ldots\otimes\mathbf{e}_{j_{k-l}'}\right\rangle, \tag{26}$$

   *where $j_1',\ldots,j_{k-l}'\in[d]$ satisfies $j_1'+\cdots+(j_{k-l}'-1)d^{k-l-1} = j$.*

3. *Finally,*

$$\mathbf{v}^\top\mathsf{Mat}_{(l,k-l)}(\mathbf{T})\mathbf{r} = \langle\mathbf{T},\mathsf{Tensor}(\mathbf{v})\otimes\mathsf{Tensor}(\mathbf{r})\rangle. \tag{27}$$

*Proof.* First we show that for a symmetric tensor $\mathbf{T}$, the linear transformation $\mathsf{Mat}_{(l,k-l)}(\mathbf{T})\mathbf{r}$ of vector $\mathbf{r}\in\mathbb{R}^{d^{k-l}}$ is equal to the tensor inner product $\langle\mathbf{T},\mathsf{Tensor}(\mathbf{r})\rangle$. This can be proved by direct calculations that for any $i\in[d^l]$:

$$\begin{aligned}
(\mathsf{Mat}_{(l,k-l)}(\mathbf{T})\mathbf{r})_i &= \sum_{j=1}^{d^{(k-l)}}\mathsf{Mat}_{(l,k-l)}(\mathbf{T})_{i,j}\mathbf{r}_j\\
&= \sum_{j_1,\ldots,j_{k-l}\in[d]}\mathsf{Mat}_{(l,k-l)}(\mathbf{T})_{i,\,j_1+\cdots+(j_{k-l}-1)d^{k-l-1}}\mathbf{r}_{j_1+\cdots+(j_{k-l}-1)d^{k-l-1}}\\
&= \sum_{j_1,\ldots,j_{k-l}\in[d]}(\mathbf{T})_{i_1',\ldots,i_l',j_1,\ldots,j_{k-l}}\mathsf{Tensor}(\mathbf{r})_{j_1,\ldots,j_{k-l}},
\end{aligned}$$

where $i'_1, \ldots, i'_j \in [d]$ satisfies $i = i'_1 + (i'_2 - 1)d + \cdots + (i'_l - 1)d^{l-1}$. Observe that the summation above further equals

$$
\begin{aligned}
&(\mathsf{Mat}_{(l,k-l)}(\mathbf{T})\mathbf{r})_i \\
&= \sum_{i_1,\ldots,i_l \in [d]} \sum_{j_1,\ldots,j_{k-l} \in [d]} (\mathbf{T})_{i_1,\ldots,i_l,j_1,\ldots,j_{k-l}} \mathsf{Tensor}(\mathbf{r})_{j_1,\ldots,j_{k-l}} \mathbb{1}\{i_1 = i'_1\} \ldots \mathbb{1}\{i_l = i'_l\} \\
&= \sum_{i_1,\ldots,i_l,j_1,\ldots,j_{k-l} \in [d]} (\mathbf{T})_{i_1,\ldots,i_l,j_1,\ldots,j_{k-l}} (\mathbf{e}_{i'_1} \otimes \ldots \otimes \mathbf{e}_{i'_l} \otimes \mathsf{Tensor}(\mathbf{r}))_{i_1,\ldots,i_l,j_1,\ldots,j_{k-l}} \\
&= \left\langle \mathbf{T}, \mathbf{e}_{i'_1} \otimes \ldots \otimes \mathbf{e}_{i'_l} \otimes \mathsf{Tensor}(\mathbf{r}) \right\rangle.
\end{aligned}
$$

Similarly, for a symmetric tensor $\mathbf{T}$ and any vector $\mathbf{v} \in \mathbb{R}^{d^l}$, and any index $j \in [d^{k-l}]$, it holds

$$
(\mathsf{Mat}_{(l,k-l)}(\mathbf{T})^\top \mathbf{v})_j = \left\langle \mathbf{T}, \mathsf{Tensor}(\mathbf{v}) \otimes \mathbf{e}_{j'_1} \otimes \ldots \otimes \mathbf{e}_{j'_{k-l}} \right\rangle,
$$

where $j'_1 + \cdots + (j'_{k-l} - 1)d^{k-l-1} = j$.

Finally, combining Equation (26) and Equation (25) we get that for any $\mathbf{v} \in \mathbb{R}^{d^l}, \mathbf{r} \in \mathbb{R}^{d^{k-l}}$, the quadratic form $\mathbf{v}^\top \mathsf{Mat}_{(l,k-l)}(\mathbf{T})\mathbf{r}$ equals $\mathbf{v}^\top \mathsf{Mat}_{(l,k-l)}(\mathbf{T})\mathbf{r} = \langle \mathbf{T}, \mathsf{Tensor}(\mathbf{v}) \otimes \mathsf{Tensor}(\mathbf{r}) \rangle$. $\quad\square$

Throughout this section, we define $l = k/2$ when $k$ is even, and $l = (k-1)/2$ when $k$ is odd. In other words, $l = \lfloor k/2 \rfloor$. We leverage the tensor unfolding algorithm proposed in [RM14], which can be described in short as follows. First we unfold the degree-$k$ Chow tensor to a matrix in $\mathbb{R}^{d^l \times d^{k-l}}$, and find its top-left singular vector $\mathbf{v} \in \mathbb{R}^{d^l}$. Then, we calculate the matrix $\mathsf{Mat}_{(1,l-1)}(\mathbf{v})$, and find its top left singular vector $\mathbf{u}$. One can show that this eigenvector $\mathbf{u}$ correlates with $\mathbf{w}^*$ significantly.

---

**Algorithm 3** $k$-Chow Tensor PCA

---

1: **Input:** Parameters $\epsilon, k, \epsilon_0, c_k, B_4 > 0$; Sample access to $\mathcal{D}$
2: Let $l = \lfloor k/2 \rfloor$
3: Draw $n = \Theta(e^k \log^k(B_4/\epsilon)d^{k-l}/(\epsilon_0^2) + 1/\epsilon)$ samples $\{(\mathbf{x}^{(i)}, y^{(i)})\}_{i=1}^n$ from $\mathcal{D}$
4: Construct $\widehat{\mathbf{M}} := (1/n)\sum_{i=1}^n \mathsf{Mat}_{(l,k-l)}(y^{(i)}\mathbf{He}_k(\mathbf{x}^{(i)}))$; compute its top left singular vector $\widehat{\mathbf{v}}^*$

5: Compute the top-left singular vector $\widehat{\mathbf{u}}$ of the matrix $\mathsf{Mat}_{1,l-1}(\widehat{\mathbf{v}}^*)$
6: **Return:** $\widehat{\mathbf{u}}$

---

Our main result for initialization is the following:

**Proposition D.2** (Initialization). *Suppose Assumption 1 holds. Assume that* $\mathrm{OPT} \leq c_{k^*}^2/(64k^*)^2$, *and let* $\epsilon_0 = c_{k^*}/(256k^*)$. *Then, Algorithm 1 applied to Problem 1.1 with* $k = k^*$ *uses*

$$
n = \Theta((k^*)^2 e^{k^*} \log^{k^*}(B_4/\epsilon)d^{\lceil k^*/2 \rceil}/(c_{k^*}^2) + 1/\epsilon)
$$

*samples, runs in polynomial time, and outputs a vector* $\mathbf{w}^0 \in \mathbb{S}^{d-1}$ *such that* $\mathbf{w}^0 \cdot \mathbf{w}^* \geq 1 - \min\{1/k^*, 1/2\}$.

We remark here that Algorithm 3 can also be used to find an approximate solution of our agnostic learning problem; however the dependence on the value of OPT is **suboptimal**, scaling with its square-root. In particular, we have the following proposition:

**Proposition D.3** (Solving the Agnostic Learning Problem Using Tensor PCA). *Suppose Assumption 1 holds. Assume that* $\mathrm{OPT} \leq c_{k^*}^2/(64k^*)^2$ *and* $\epsilon \leq 1/64$. *Let* $\epsilon_0 = c_{k^*}\epsilon/16$. *Then, Algorithm 1 applied to Problem 1.1 with* $k = k^*$ *uses* $n = \Theta(e^{k^*} \log^{k^*}(B_4/\epsilon)d^{\lceil k^*/2 \rceil}/(c_{k^*}^2\epsilon^2) + 1/\epsilon)$ *samples, runs in polynomial time, and outputs a vector* $\mathbf{w}^0 \in \mathbb{S}^{d-1}$ *such that*

$$
\mathbf{w}^0 \cdot \mathbf{w}^* \geq 1 - \frac{4}{c_{k^*}}\sqrt{\mathrm{OPT}} - 2\epsilon/3.
$$

*Furthermore, the* $L_2^2$ *error of* $\mathbf{w}^0$ *is at most*

$$
\mathcal{L}_2^\sigma(\mathbf{w}^0) = O\left(C_{k^*}\left(\frac{1}{c_{k^*}}\sqrt{\mathrm{OPT}} + \epsilon\right)\right).
$$

Thus, when $\mathrm{OPT} = 0$ (i.e., in the realizable cases), applying Algorithm 3 with $O(d^{\lceil k^*/2\rceil}/\epsilon^2)$ samples recovers the hidden vector $\mathbf{w}^*$.

**Roadmap** To prove Proposition D.2 and Proposition D.3 we need three main ingredients. First, we will show (in Lemma D.4 and its corollary Corollary D.5) that the top-left singular vector $\mathbf{v}^*$ of the unfolded matrix $\mathbf{M} := \mathsf{Mat}_{(l,k-l)}(\mathbf{E}_{(\mathbf{x},y)\sim\mathcal{D}}[y\mathbf{He}_k(\mathbf{x})])$ correlates significantly with the vectorized $l$-product tensor, $\mathsf{Vec}(\mathbf{w}^{*\otimes l})$. This indicates that $\mathbf{v}^*$ contains rich information about the direction of $\mathbf{w}^*$. However, since we only have access to $\widehat{\mathbf{M}}$, the empirical estimation of $\mathbf{M}$, we need to ensure that the top-left singular vector of $\widehat{\mathbf{M}}$, denoted by $\widehat{\mathbf{v}}^*$, is close to $\mathbf{v}^*$. This is proved in Lemma D.13 using sophisticated matrix concentration bounds. In particular, in Equation (35) we guarantee that the angle between $\widehat{\mathbf{v}}^*$ and $\mathbf{v}^*$ is bounded by $O(\epsilon_0/c_k)$ for any small constant $\epsilon_0 > 0$, provided that we take $\tilde{\Theta}(d^{\lceil k/2\rceil}/\epsilon_0^2)$ samples and assume that $\mathrm{OPT} \lesssim c_k^2$. The inner product between $\widehat{\mathbf{v}}^*$ and $\mathsf{Vec}(\mathbf{w}^{*\otimes l})$ can then be bounded below by $1 - O((\sqrt{\mathrm{OPT}}+\epsilon_0)/c_k)$. Combining with Corollary D.5, this implies that $\widehat{\mathbf{v}}^*$ correlates with $\mathsf{Vec}(\mathbf{w}^{*\otimes l})$ significantly. Finally, in Lemma D.17 we show that after unfolding the $\mathbb{R}^{d^l}$ vector $\widehat{\mathbf{v}}^*$ to an $\mathbb{R}^{d\times d^{l-1}}$ matrix, its top-left singular vector $\mathbf{u}$ correlates with $\mathbf{w}^*$ significantly; it particular, we have $\mathbf{w}^* \cdot \mathbf{u} \gtrsim 1 - c\epsilon_0$ for some absolute constant $c > 0$. Combining these results and choosing $\epsilon_0 \approx c_{k^*}/k^*$, we get Proposition D.2, and choosing $\epsilon_0 \approx c_{k^*}\epsilon$ yields Proposition D.3.

### D.1 Signal in the $k$-Chow Tensor

Our first observation is that for any left singular vector $\mathbf{v}$ of $\mathsf{Mat}_{(l,k-l)}(\mathbf{C}_k)$, the singular value $\rho(\mathbf{v})$ is close to the inner product between $\mathbf{v}$ and $\mathsf{Vec}(\mathbf{w}^{*\otimes l})$, where $l = \lceil k/2\rceil$. Concretely, we have:

**Lemma D.4.** *Let $\mathbf{v}$ be any left singular vector of $\mathsf{Mat}_{(l,k-l)}(\mathbf{C}_k)$. Then,*

$$|\rho(\mathbf{v}) - c_k(\mathsf{Vec}(\mathbf{w}^{*\otimes l}) \cdot \mathbf{v})| \leq \sqrt{\mathrm{OPT}}.$$

*Proof.* Recall that the singular value of the left singular vector $\mathbf{v}$ satisfies

$$\rho(\mathbf{v}) = \max_{\mathbf{r}\in\mathbb{R}^{d^{k-l}},\|\mathbf{r}\|_2=1} \mathbf{v}^\top \mathsf{Mat}_{(l,k-l)}(\mathbf{C}_k)\mathbf{r} \overset{(i)}{=} \max_{\mathbf{r}\in\mathbb{R}^{k-l},\|\mathbf{r}\|_2=1} \langle\mathbf{C}_k, \mathsf{Tensor}(\mathbf{v}) \otimes \mathsf{Tensor}(\mathbf{r})\rangle,$$

where we used Equation (27) in $(i)$. Since $\mathbf{C}_k = \mathbf{C}_k^* + \mathbf{H}_k$, we further have

$$\langle\mathbf{C}_k, \mathsf{Tensor}(\mathbf{v}) \otimes \mathsf{Tensor}(\mathbf{r})\rangle = \langle\mathbf{C}_k^*, \mathsf{Tensor}(\mathbf{v}) \otimes \mathsf{Tensor}(\mathbf{r})\rangle + \langle\mathbf{H}_k, \mathsf{Tensor}(\mathbf{v}) \otimes \mathsf{Tensor}(\mathbf{r})\rangle.$$

We bound both terms above respectively. For the first term, plugging in the definition of $\mathbf{C}_k^*$ and using Fact C.3, we have

$$\langle\mathbf{C}_k^*, \mathsf{Tensor}(\mathbf{v}) \otimes \mathsf{Tensor}(\mathbf{r})\rangle \tag{28}$$

$$= \mathop{\mathbf{E}}_{\mathbf{x}\sim\mathcal{N}_d}\left[\sum_{j\geq k^*} c_j\langle\mathbf{He}_j(\mathbf{x}), \mathbf{w}^{*\otimes j}\rangle\langle\mathbf{He}_k(\mathbf{x}), \mathsf{Tensor}(\mathbf{v}) \otimes \mathsf{Tensor}(\mathbf{r})\rangle\right]$$

$$\overset{(i)}{=} c_k\left\langle\mathbf{w}^{*\otimes k}, \mathsf{Sym}(\mathsf{Tensor}(\mathbf{v}) \otimes \mathsf{Tensor}(\mathbf{r}))\right\rangle$$

$$\overset{(ii)}{=} c_k\sum_{i_1,\ldots,i_k} \mathbf{w}_{i_1}^* \cdots \mathbf{w}_{i_l}^* \mathsf{Tensor}(\mathbf{v})_{i_1,\ldots,i_l}\mathbf{w}_{i_{l+1}}^* \cdots \mathbf{w}_{i_k}^* \mathsf{Tensor}(\mathbf{r})_{i_{l+1},\ldots,i_k}$$

$$= c_k(\mathsf{Vec}(\mathbf{w}^{*\otimes l}) \cdot \mathbf{v})(\mathsf{Vec}(\mathbf{w}^{*\otimes k-l}) \cdot \mathbf{r}), \tag{29}$$

note that we applied Fact C.3 in equation $(i)$ and Fact C.1(1) in $(ii)$. Next, for the second term, after applying Cauchy-Schwarz inequality, it holds

$$|\langle\mathbf{H}_k, \mathsf{Tensor}(\mathbf{v}) \otimes \mathsf{Tensor}(\mathbf{r})\rangle|$$

$$= \left|\mathop{\mathbf{E}}_{(\mathbf{x},y)\sim\mathcal{D}}\left[(y - \sigma(\mathbf{w}^* \cdot \mathbf{x}))\langle\mathbf{He}_k(\mathbf{x}), \mathsf{Tensor}(\mathbf{v}) \otimes \mathsf{Tensor}(\mathbf{r})\rangle\right]\right|$$

$$\leq \sqrt{\mathop{\mathbf{E}}_{(\mathbf{x},y)\sim\mathcal{D}}[(y - \sigma(\mathbf{w}^* \cdot \mathbf{x}))^2]}\sqrt{\mathop{\mathbf{E}}_{\mathbf{x}\sim\mathcal{N}_d}\left[(\langle\mathbf{He}_k(\mathbf{x}), \mathsf{Tensor}(\mathbf{v}) \otimes \mathsf{Tensor}(\mathbf{r})\rangle)^2\right]}$$

$$= \sqrt{\mathrm{OPT}}\|\mathsf{Sym}(\mathsf{Tensor}(\mathbf{v}) \otimes \mathsf{Tensor}(\mathbf{r}))\|_F.$$

Since for any $k$-tensor $A$ we have $\|\mathrm{Sym}(A)\|_F \le \|A\|_F$, and in addition, observe that as $\|\mathbf{v}\|_2 = \|\mathbf{r}\|_2 = 1$ it holds

$$\|\mathsf{Tensor}(\mathbf{v}) \otimes \mathsf{Tensor}(\mathbf{r})\|_F^2 = \sum_{\substack{i_1,\dots,i_l \\ i_{l+1},\dots,i_k}} (\mathsf{Tensor}(\mathbf{v}))^2_{i_1,\dots,i_l}(\mathsf{Tensor}(\mathbf{r}))^2_{i_{l+1},\dots,i_k} = \sum_{i=1}^{d^l}\sum_{j=1}^{d^{k-l}} \mathbf{v}_i^2 \mathbf{r}_j^2 = 1,$$

we finally have

$$|\langle \mathbf{H}_k, \mathsf{Tensor}(\mathbf{v}) \otimes \mathsf{Tensor}(\mathbf{r})\rangle| \le \sqrt{\mathrm{OPT}}\|\mathsf{Tensor}(\mathbf{v}) \otimes \mathsf{Tensor}(\mathbf{r})\|_F = \sqrt{\mathrm{OPT}}. \qquad (30)$$

Combining Equation (28) and Equation (30), we get that for any $\mathbf{v} \in \mathbb{R}^{d^l}, \mathbf{r} \in \mathbb{R}^{d^{k-l}}$ such that $\|\mathbf{v}\|_2 = \|\mathbf{r}\|_2 = 1$, it holds

$$\mathbf{v}^\top \mathsf{Mat}_{(l,k-l)}(\mathbf{C}_k)\mathbf{r} \le c_k(\mathsf{Vec}(\mathbf{w}^{*\otimes l}) \cdot \mathbf{v})(\mathsf{Vec}(\mathbf{w}^{*\otimes k-l}) \cdot \mathbf{r}) + \sqrt{\mathrm{OPT}}.$$

Therefore, the singular value of $\mathbf{v}$ must satisfy

$$\rho(\mathbf{v}) \le \max_{\mathbf{r} \in \mathbb{R}^{d^{k-l}}, \|\mathbf{r}\|_2=1} c_k(\mathsf{Vec}(\mathbf{w}^{*\otimes l}) \cdot \mathbf{v})(\mathsf{Vec}(\mathbf{w}^{*\otimes k-l}) \cdot \mathbf{r}) + \sqrt{\mathrm{OPT}}$$

$$= c_k(\mathsf{Vec}(\mathbf{w}^{*\otimes l}) \cdot \mathbf{v}) + \sqrt{\mathrm{OPT}}, \qquad (31)$$

where in the equation above, we used the observation that as $\|\mathsf{Vec}(\mathbf{w}^{*\otimes k-l})\|_2 = \|\mathbf{w}^{*\otimes k-l}\|_F = 1$, it holds $\max_{\mathbf{r} \in \mathbb{R}^{d^{k-l}}, \|\mathbf{r}\|_2=1}(\mathsf{Vec}(\mathbf{w}^{*\otimes k-l}) \cdot \mathbf{r}) = \|\mathsf{Vec}(\mathbf{w}^{*\otimes k-l})\|_2 = 1$.

Similarly, we have

$$\rho(\mathbf{v}) = \max_{\mathbf{r} \in \mathbb{R}^{d^{k-l}}, \|\mathbf{r}\|_2=1} \mathbf{v}^\top \mathsf{Mat}_{(l,k-l)}(\mathbf{C}_k)\mathbf{r} = \max_{\mathbf{r} \in \mathbb{R}^{d^{k-l}}, \|\mathbf{r}\|_2=1} \langle \mathbf{C}_k, \mathsf{Tensor}(\mathbf{v}) \otimes \mathsf{Tensor}(\mathbf{r})\rangle$$

$$= \max_{\mathbf{r} \in \mathbb{R}^{d^{k-l}}, \|\mathbf{r}\|_2=1} c_k(\mathsf{Vec}(\mathbf{w}^{*\otimes l}) \cdot \mathbf{v})(\mathsf{Vec}(\mathbf{w}^{*\otimes k-l}) \cdot \mathbf{r}) + \langle \mathbf{H}_k, \mathsf{Tensor}(\mathbf{v}) \otimes \mathsf{Tensor}(\mathbf{r})\rangle$$

$$\ge \max_{\mathbf{r} \in \mathbb{R}^{d^{k-l}}, \|\mathbf{r}\|_2=1} c_k(\mathsf{Vec}(\mathbf{w}^{*\otimes l}) \cdot \mathbf{v})(\mathsf{Vec}(\mathbf{w}^{*\otimes k-l}) \cdot \mathbf{r}) - \sqrt{\mathrm{OPT}}$$

$$= c_k(\mathsf{Vec}(\mathbf{w}^{*\otimes l}) \cdot \mathbf{v}) - \sqrt{\mathrm{OPT}},$$

completing the proof of Lemma D.4. $\qquad\square$

A direct application of Lemma D.4 is that the top-left singular vector $\mathbf{v}^* \in \mathbb{R}^{d^l}$ of $\mathsf{Mat}_{(l,k-l)}(\mathbf{C}_k)$ has singular value at least $c_k - \sqrt{\mathrm{OPT}}$, and in addition, $\mathbf{v}^*$ aligns well with $\mathsf{Vec}(\mathbf{w}^{*\otimes l})$.

**Corollary D.5.** *The top-left singular vector $\mathbf{v}^* \in \mathbb{R}^{d^l}$ of the unfolded tensor $\mathsf{Mat}_{(l,k-l)}(\mathbf{C}_k)$ has corresponding singular value $\rho(\mathbf{v}^*) \ge c_k - \sqrt{\mathrm{OPT}}$. In addition, it holds that $\mathbf{v}^* \cdot \mathsf{Vec}(\mathbf{w}^{*\otimes l}) \ge 1 - (2\sqrt{\mathrm{OPT}})/c_k$.*

*Proof.* Plugging in $\mathbf{v} = \mathsf{Vec}(\mathbf{w}^{*\otimes l})$ to Lemma 2.2, we get that $\rho(\mathsf{Vec}(\mathbf{w}^{*\otimes l})) \ge c_k - \sqrt{\mathrm{OPT}}$. Thus, the top singular value must satisfy $\rho_1 \ge c_k - \sqrt{\mathrm{OPT}}$. Recall again that as proved in Lemma 2.2, it holds $\rho(\mathbf{v}^*) \le c_k \mathbf{v}^* \cdot \mathsf{Vec}(\mathbf{w}^{*\otimes l}) + \sqrt{\mathrm{OPT}}$. Thus, since $\rho(\mathsf{Vec}(\mathbf{w}^{*\otimes l})) \ge c_k - \sqrt{\mathrm{OPT}}$ we have $\mathbf{v}^* \cdot \mathsf{Vec}(\mathbf{w}^{*\otimes l}) \ge 1 - (2\sqrt{\mathrm{OPT}})/c_k$. $\qquad\square$

### D.2 Concentration of the Unfolded Tensor Matrix

We start with some notations. Let us denote $\mathbf{M}^{(i)} = \mathsf{Mat}_{(l,k-l)}(y^{(i)}\mathbf{He}_k(\mathbf{x}^{(i)}))$ for $i \in [n]$ and $\widehat{\mathbf{M}} = \frac{1}{n}\sum_{i=1}^n \mathbf{M}^{(i)}$, which is the empirical approximation of $\mathbf{M} = \mathsf{Mat}_{(l,k-l)}(\mathbf{C}_k) = \mathsf{Mat}_{(l,k-l)}(\mathbb{E}_{(\mathbf{x},y)\sim\mathcal{D}}[y\mathbf{He}_k(\mathbf{x})])$. We will use Wedin's theorem to bound the distance between the top left singular vector $\mathbf{v}^*$ of $\mathbf{M}$ and the top singular vector $\widehat{\mathbf{v}}^*$ of the empirical $\widehat{\mathbf{M}}$.

**Fact D.6** (Wedin's theorem). *Let $\theta(\mathbf{v}^*, \widehat{\mathbf{v}}^*)$ be the angle between the top left singular vectors $\mathbf{v}^* \in \mathbb{R}^{d^l}$ and $\widehat{\mathbf{v}}^* \in \mathbb{R}^{d^l}$ of $\mathbf{M}$ and $\widehat{\mathbf{M}}$ respectively. Let $\rho_1$ and $\rho_2$ be the first 2 singular values of $\mathbf{M}$. Then, it holds that:*

$$\sin(\theta(\mathbf{v}^*, \widehat{\mathbf{v}}^*)) \le \frac{\|\mathbf{M} - \widehat{\mathbf{M}}\|_2}{\rho_1 - \rho_2 - \|\mathbf{M} - \widehat{\mathbf{M}}\|_2}.$$

We first observe that $\mathbf{M}$ admits a large gap between the first and second singular values.

**Claim D.7** (Singular Gap of Unfolded Tensor Matrix). *Let $\rho_1, \rho_2$ be the top two singular values of* $\mathbf{M} = \mathsf{Mat}_{(l,k-l)}(\mathbf{C}_k)$. *Then $\rho_1 - \rho_2 \geq (c_k - 8\sqrt{\mathrm{OPT}})/2$.*

*Proof.* Recall that in Corollary D.5 we showed $\rho_1 = \rho(\mathbf{v}^*) \geq c_k - \sqrt{\mathrm{OPT}}$ and $\mathbf{v}^* \cdot \mathsf{Vec}(\mathbf{w}^{*\otimes l}) \geq 1 - (2\sqrt{\mathrm{OPT}})/c_k$. Now let $\mathbf{v} \in \mathbb{R}^{d^l}$ be any left singular vector of $\mathbf{M}$ that is orthogonal to $\mathbf{v}^*$. We can decompose $\mathsf{Vec}(\mathbf{w}^{*\otimes l})$ into $\mathsf{Vec}(\mathbf{w}^{*\otimes l}) = a\mathbf{v}^* + b\mathbf{v} + \mathbf{v}'$ where $\mathbf{v}'$ is orthogonal to both $\mathbf{v}^*$ and $\mathbf{v}$, and $a^2 + b^2 \leq 1$. Then, since $\mathsf{Vec}(\mathbf{w}^{*\otimes l}) \cdot \mathbf{v}^* = a \geq 1 - (2\sqrt{\mathrm{OPT}})/c_k$, we thus have $\mathsf{Vec}(\mathbf{w}^{*\otimes l}) \cdot \mathbf{v} = b \leq \sqrt{1-a^2} \leq 1 - a^2/2$. This implies that

$$\rho_1 - \rho(\mathbf{v}) \geq c_k - \sqrt{\mathrm{OPT}} - (c_k(\mathsf{Vec}(\mathbf{w}^{*\otimes l}) \cdot \mathbf{v}) + \sqrt{\mathrm{OPT}})$$
$$\geq c_k(1-b) - 2\sqrt{\mathrm{OPT}} \geq c_k(1 - (1 - a^2/2)) - 2\sqrt{\mathrm{OPT}} \geq c_k/2 - 4\sqrt{\mathrm{OPT}},$$

and hence we get $\rho_1 - \rho_2 \geq (c_k - 8\sqrt{\mathrm{OPT}})/2$, completing the proof of Claim D.7. $\qquad\square$

Thus, our remaining goal is to bound the operator norm of $\mathbf{M} - \widehat{\mathbf{M}}$. For this purpose, we use the following matrix concentration inequality from [DPVLB24] (also Theorem 2.7 in [BvH22]).

**Fact D.8** (Lemma I.5 [DPVLB24]). *Let $\mathbf{Z}^{(i)}, i \in [n]$, be independent, mean-zero, self-adjoint matrices. Define:*

$$\gamma^2 := \left\| \mathbf{E}\left[ \left( \sum_{i=1}^n \mathbf{Z}^{(i)} \right)^2 \right] \right\|_2, \quad \gamma_*^2 := \sup_{\|\mathbf{v}\|_2 = \|\mathbf{r}\|_2 = 1} \mathbf{E}\left[ \left( \sum_{i=1}^n \mathbf{v}^\top \mathbf{Z}^{(i)} \mathbf{r} \right)^2 \right], \quad \bar{R}^2 := \mathbf{E}\left[ \max_{i \in [n]} \|\mathbf{Z}^{(i)}\|_2^2 \right].$$

*Then, for any $R \geq \bar{R}^{1/2}\gamma^{1/2} + \sqrt{2}\bar{R}$, and any $t \geq 0$, if $\delta = \mathbf{Pr}[\max_{i \in [n]} \|\mathbf{Z}^{(i)}\|_2 \geq R]$, then with probability at least $1 - \delta - de^{-t}$,*

$$\left\| \sum_{i=1}^n \mathbf{Z}^{(i)} \right\|_2 - 2\gamma \lesssim \gamma_* t^{1/2} + R^{1/3}\gamma^{2/3}t^{2/3} + Rt. \tag{32}$$

However, note that $\widehat{\mathbf{M}}$ and $\mathbf{M}$ are not symmetric matrices, hence to apply matrix concentration inequalities we will be working on the symmetrization of $\widehat{\mathbf{M}}$ and $\mathbf{M}$ for simplicity, which we will denote by $\widehat{\mathbf{P}}$ and $\mathbf{P}$:

$$\widehat{\mathbf{P}} = \frac{1}{n}\sum_{i=1}^n \mathbf{P}^{(i)} = \frac{1}{n}\sum_{i=1}^n \begin{bmatrix} \mathbf{0} & \mathbf{M}^{(i)} \\ \mathbf{M}^{(i)\top} & \mathbf{0} \end{bmatrix} = \begin{bmatrix} \mathbf{0} & \widehat{\mathbf{M}} \\ \widehat{\mathbf{M}}^\top & \mathbf{0} \end{bmatrix}; \quad \mathbf{P} = \begin{bmatrix} \mathbf{0} & \mathbf{M} \\ \mathbf{M}^\top & \mathbf{0} \end{bmatrix}.$$

Before we prove the main theorem of this subsection, we introduce two final pieces of tools that will be used later in the proof. The first one is Gaussian hypercontractivity.

**Fact D.9** (Gaussian Hypercontractivity). *Let $f(\mathbf{x}) : \mathbb{R}^d \to \mathbb{R}$ be a multivariate polynomial of degree at most $k$. Let $\mathbf{x}$ be a standard Gaussian random variable of $\mathbb{R}^d$. Then, for any $p \geq 1$ it holds*

$$\|f(\mathbf{x})\|_{L^p} \leq (p-1)^{k/2}\|f(\mathbf{x})\|_{L^2}.$$

Gaussian hypercontractivity controls the moments of a polynomial $f(\mathbf{x})$. To utilize the bound on these moments, we make use of the following inequality from [DNGL23].

**Fact D.10** (Lemma 23 [DNGL23]). *Let $A, B$ be random variables such that $\|B\|_{L^p} \leq \sigma_B p^C$ for all $p \geq 1$ and some positive real numbers $\sigma_B, C$. Then,*

$$\mathbf{E}[AB] \leq \mathbf{E}[|A|]\sigma_B(2e)^C\left( \max\left\{ 1, \frac{1}{C}\log\left( \frac{(\mathbf{E}[A^2])^{1/2}}{\mathbf{E}[|A|]} \right) \right\} \right)^C.$$

We also make use of the following lemma that bounds the magnitude of label $y$ without loss of generality.

**Lemma D.11** (Bound on Labels). *Let $P_{B_y}(z) : \mathbb{R} \to \mathbb{R}$ be a function that truncates the value of $z$ to the threshold $B_y$: $P_{B_y}(z) = z\mathbb{1}\{|z| \leq B_y\} + B_y\mathbb{1}\{|z| \geq B_y\}$. Assume that Assumption 1 holds. Then choosing $B_y := \sqrt{4B_4/\epsilon}$, it holds that*

$$\mathop{\mathbf{E}}_{(\mathbf{x},y)\sim\mathcal{D}}[(P_{B_y}(y) - \sigma(\mathbf{w}^* \cdot \mathbf{x}))^2] \leq \mathrm{OPT} + \epsilon.$$

*Therefore, it is without loss of generality to assume that $|y| \leq B_y$.*

*Proof.* After truncating the label $y$, we have

$$\mathop{\mathbf{E}}_{(\mathbf{x},y)\sim\mathcal{D}}[(P_t(y) - \sigma(\mathbf{w}^* \cdot \mathbf{x}))^2]$$

$$= \mathop{\mathbf{E}}_{(\mathbf{x},y)\sim\mathcal{D}}[(P_t(y) - \sigma(\mathbf{w}^* \cdot \mathbf{x}))^2\mathbb{1}\{\sigma(\mathbf{w}^* \cdot \mathbf{x}) \leq t\}] + \mathop{\mathbf{E}}_{(\mathbf{x},y)\sim\mathcal{D}}[(P_t(y) - \sigma(\mathbf{w}^* \cdot \mathbf{x}))^2\mathbb{1}\{\sigma(\mathbf{w}^* \cdot \mathbf{x}) \geq t\}]$$

$$\leq \mathop{\mathbf{E}}_{(\mathbf{x},y)\sim\mathcal{D}}[(y - \sigma(\mathbf{w}^* \cdot \mathbf{x}))^2] + \mathop{\mathbf{E}}_{(\mathbf{x},y)\sim\mathcal{D}}[(P_t(y) - \sigma(\mathbf{w}^* \cdot \mathbf{x}))^2\mathbb{1}\{\sigma(\mathbf{w}^* \cdot \mathbf{x}) \geq t\}]$$

$$\leq \mathrm{OPT} + 2\mathop{\mathbf{E}}_{(\mathbf{x},y)\sim\mathcal{D}}[(t^2 + \sigma^2(\mathbf{w}^* \cdot \mathbf{x}))\mathbb{1}\{\sigma(\mathbf{w}^* \cdot \mathbf{x}) \geq t\}].$$

Since $\mathbf{E}_{\mathbf{x}\sim\mathcal{N}_d}[\sigma^4(\mathbf{w}^*\cdot\mathbf{x})] \leq B_4$ by assumption, we have by Markov's inequality that $\mathbf{Pr}[\sigma(\mathbf{w}^*\cdot\mathbf{x}) \geq t] \leq B_4/t^4$. Therefore, we can further bound $\mathbf{E}_{(\mathbf{x},y)\sim\mathcal{D}}[(P_t(y) - \sigma(\mathbf{w}^* \cdot \mathbf{x}))^2]$ from above by

$$\mathop{\mathbf{E}}_{(\mathbf{x},y)\sim\mathcal{D}}[(P_t(y) - \sigma(\mathbf{w}^* \cdot \mathbf{x}))^2] \leq \mathrm{OPT} + \frac{2B_4}{t^2} + 2\sqrt{\mathop{\mathbf{E}}_{\mathbf{x}\sim\mathcal{N}_d}[\sigma^4(\mathbf{w}^* \cdot \mathbf{x})]\,\mathbf{Pr}[\sigma(\mathbf{w}^* \cdot \mathbf{x}) \geq t]}$$

$$\leq \mathrm{OPT} + \frac{4B_4}{t^2}.$$

Thus, choosing $t = \sqrt{4B_4/\epsilon}$ we have

$$\mathop{\mathbf{E}}_{(\mathbf{x},y)\sim\mathcal{D}}[(P_t(y) - \sigma(\mathbf{w}^* \cdot \mathbf{x}))^2] \leq \mathrm{OPT} + \epsilon,$$

indicating that we can assume without loss of generality that $|y| \leq B_y := \sqrt{4B_4/\epsilon}$, completing the proof of Lemma D.11. $\qquad\square$

After assuming that $y$ is bounded by $B_y$ without loss of generality, we can then bound the $2^{\mathrm{nd}}$ and $4^{\mathrm{th}}$ moments of $y$. These bounds on the moments of the label $y$ will be used when we implement Fact D.10 to get finer bounds compared to what we would get from a simple application of Cauchy-Schwarz. In particular, we use Fact D.10 to derive upper bounds on expectations like $\mathbf{E}_{(\mathbf{x},y)\sim\mathcal{D}}[y^2 f^2(\mathbf{x})]$, where $f(\mathbf{x})$ is a polynomial of $\mathbf{x}$, as we have control on the $p^{\mathrm{th}}$ moments of $f(\mathbf{x})$ using Gaussian hypercontractivity Fact D.9.

**Lemma D.12** (Moments of Labels). *If $\mathrm{OPT} \leq 1/16$, then $1/2 \leq \mathbf{E}_y[y^2] \leq 2$ and $\mathbf{E}_y[y^4] \leq 8B_4/\epsilon$.*

*Proof.* We first bound the $2^{\mathrm{nd}}$ moment of the label $y$. Note that since $\sigma(\mathbf{w}^* \cdot \mathbf{x})$ is normalized such that $\mathbf{E}_{\mathbf{x}\sim\mathcal{N}_d}[(\sigma(\mathbf{w}^* \cdot \mathbf{x}))^2] = 1$, we have

$$\mathop{\mathbf{E}}_y[y^2] = \mathop{\mathbf{E}}_{(\mathbf{x},y)\sim\mathcal{D}}[(y - \sigma(\mathbf{w}^* \cdot \mathbf{x}) + \sigma(\mathbf{w}^* \cdot \mathbf{x}))^2]$$

$$\overset{(i)}{\leq} (1 + 1/a)\mathop{\mathbf{E}}_{(\mathbf{x},y)\sim\mathcal{D}}[(y - \sigma(\mathbf{w}^* \cdot \mathbf{x}))^2] + (1 + a)\mathop{\mathbf{E}}_{\mathbf{x}\sim\mathcal{N}_d}[(\sigma(\mathbf{w}^* \cdot \mathbf{x}))^2].$$

We used Young's inequality in $(i)$. Choosing $a = 1/8$ and since we assumed $\mathrm{OPT} \leq 1/16$, it holds $\mathbf{E}_y[y^2] \leq 9/16 + 9/8 \leq 2$. In addition, using Cauchy-Schwarz inequality, we have

$$\mathop{\mathbf{E}}_y[y^2] = \mathop{\mathbf{E}}_{(\mathbf{x},y)\sim\mathcal{D}}[(y - \sigma(\mathbf{w}^* \cdot \mathbf{x}) + \sigma(\mathbf{w}^* \cdot \mathbf{x}))^2]$$

$$= \mathop{\mathbf{E}}_{(\mathbf{x},y)\sim\mathcal{D}}[(y - \sigma(\mathbf{w}^* \cdot \mathbf{x}))^2] + \mathop{\mathbf{E}}_{\mathbf{x}\sim\mathcal{N}_d}[(\sigma(\mathbf{w}^* \cdot \mathbf{x}))^2] + 2\mathop{\mathbf{E}}_{(\mathbf{x},y)\sim\mathcal{D}}[(y - \sigma(\mathbf{w}^* \cdot \mathbf{x}))\sigma(\mathbf{w}^* \cdot \mathbf{x})]$$

$$\geq 1 - 2\sqrt{\mathop{\mathbf{E}}_{(\mathbf{x},y)\sim\mathcal{D}}[(y - \sigma(\mathbf{w}^* \cdot \mathbf{x}))^2]\mathop{\mathbf{E}}_{\mathbf{x}\sim\mathcal{N}_d}[(\sigma(\mathbf{w}^* \cdot \mathbf{x}))^2]} \geq 1/2.$$

This yields the first statement of the lemma. For the remaining statement, notice that since $y \leq B_y$, we have $\mathbf{E}_y[y^4] \leq B_y^2\,\mathbf{E}_y[y^2] \leq 2B_y^2 = 8B_4/\epsilon$. $\qquad\square$

We now proceed to bound the sample complexity of Algorithm 3, the argument for which relies on applying Fact D.8 to $\mathbf{Z}^{(i)} = \frac{1}{n}(\mathbf{P}^{(i)} - \mathbf{P})$ and is summarized in the following lemma.

**Lemma D.13** (Sample Complexity for Estimating the Unfolded Tensor Matrix). *Let $\epsilon, \epsilon_0 > 0$. Consider the unfolded matrix $\mathbf{M} = \mathsf{Mat}_{(l,k-l)}(\mathbf{E}_{(\mathbf{x},y)\sim\mathcal{D}}[y\mathbf{He}_k(\mathbf{x})])$ and its empirical estimate $\widehat{\mathbf{M}} := (1/n)\sum_{i=1}^n \mathsf{Mat}_{(l,k-l)}(y^{(i)}\mathbf{He}_k(\mathbf{x}^{(i)}))$, where $\{(\mathbf{x}^{(i)}, y^{(i)})\}_{i=1}^n$ are $n = \Theta(e^k\log^k(B_4/\epsilon)d^{k/2}/\epsilon_0^2 + 1/\epsilon)$ i.i.d. samples from $\mathcal{D}$. Then, with probability at least $1 - \exp(-d^{1/2})$,*

$$\|\widehat{\mathbf{M}} - \mathbf{M}\|_2 \leq \epsilon_0.$$

*Moreover, if $\widehat{\mathbf{v}}^*$ is the top left-singular vector of $\widehat{\mathbf{M}}$, then with probability at least $1 - \exp(-d^{1/2})$,*

$$\widehat{\mathbf{v}}^* \cdot \mathsf{Vec}(\mathbf{w}^{*\otimes l}) \geq 1 - \frac{2}{c_k}\sqrt{\mathrm{OPT}} - \frac{2\epsilon_0}{(c_k/2 - 4\sqrt{\mathrm{OPT}}) - \epsilon_0}.$$

*Proof.* The proof hinges on applying Fact D.8, for which we need to bound above the parameters $\gamma$, $\gamma_*$, and $\bar{R}$ defined in the same fact. We do so in three separate claims, as follows.

**Claim D.14.** $\gamma \lesssim \dfrac{d^{(k-l)/2}e^k\log^{k/2}(B_4/\epsilon)}{\sqrt{n}} = \sqrt{\dfrac{d^{k-l}e^k\log^k(B_4/\epsilon)}{n}}.$

*Proof.* By the definition of $\gamma$, we have:

$$\gamma^2 = \left\| \mathbf{E}\left[\left(\frac{1}{n}\sum_{i=1}^n \mathbf{P}^{(i)} - \mathbf{P}\right)^2\right] \right\|_2 \leq \frac{1}{n}\|\mathbf{E}[(\mathbf{P}^{(i)} - \mathbf{P})^2]\|_2 \leq \frac{1}{n}\|\mathbf{E}[(\mathbf{P}^{(i)})^2]\|_2$$

$$= \frac{1}{n}\max_{\substack{\mathbf{v}\in\mathbb{R}^{d^l}, \mathbf{r}\in\mathbb{R}^{d^{k-l}} \\ \|\mathbf{v}\|_2^2 + \|\mathbf{r}\|_2^2 = 1}} \mathbf{E}[\mathbf{v}^\top\mathbf{M}^{(i)}(\mathbf{M}^{(i)})^\top\mathbf{v} + \mathbf{r}^\top(\mathbf{M}^{(i)})^\top\mathbf{M}^{(i)}\mathbf{r}],$$

Observe that $\mathbf{v}^\top\mathbf{M}^{(i)}(\mathbf{M}^{(i)})^\top\mathbf{v} = \|\mathbf{v}^\top\mathbf{M}^{(i)}\|_2^2 = \sum_{j=1}^{d^{k-l}}(\mathbf{v}^\top\mathbf{M}^{(i)})_j^2$, and notice that by definition $\mathbf{M}^{(i)} = \mathsf{Mat}_{(l,k-l)}(y^{(i)}\mathbf{He}_k(\mathbf{x}^{(i)}))$ where $y^{(i)}\mathbf{He}_k(\mathbf{x}^{(i)})$ is a symmetric tensor, hence using Equation (26) we get

$$\mathbf{v}^\top\mathbf{M}^{(i)}(\mathbf{M}^{(i)})^\top\mathbf{v} = \sum_{(j_1,j_2,\ldots,j_{l-k})\in[d]^{l-k}} \left\langle y^{(i)}\mathbf{He}_k(\mathbf{x}^{(i)}), \mathsf{Tensor}(\mathbf{v}) \otimes \mathbf{e}_{j_1} \otimes \ldots \otimes \mathbf{e}_{j_{k-l}} \right\rangle^2.$$

As $(\mathbf{x}^{(i)}, y^{(i)})$ are i.i.d. copies of $(\mathbf{x}, y)$, using the linearity of expectation, we have

$$\mathbf{E}[\mathbf{v}^\top\mathbf{M}^{(i)}(\mathbf{M}^{(i)})^\top\mathbf{v}]$$

$$= \sum_{j_1,\ldots,j_{k-l}} \mathbf{E}_{(\mathbf{x},y)\sim\mathcal{D}}\left[y^2\left\langle\mathbf{He}_k(\mathbf{x}), \mathsf{Tensor}(\mathbf{v}) \otimes \mathbf{e}_{j_1} \otimes \ldots \otimes \mathbf{e}_{j_{k-l}}\right\rangle^2\right]. \quad (33)$$

Now given any indices $j_1, \ldots, j_{k-l} \in [d]$, observe that $f_{j_1,\ldots,j_{k-l}}(\mathbf{x}) := \langle\mathbf{He}_k(\mathbf{x}), \mathsf{Tensor}(\mathbf{v}) \otimes \mathbf{e}_{j_1} \otimes \ldots \otimes \mathbf{e}_{j_{k-l}}\rangle$ is a polynomial of $\mathbf{x}$ of degree at most $k$, and note that

$$\mathbf{E}_{\mathbf{x}\sim\mathcal{N}_d}[f_{j_1,\ldots,j_{k-l}}(\mathbf{x})^2] = \mathbf{E}_{\mathbf{x}\sim\mathcal{N}_d}\left[\left\langle\mathbf{He}_k(\mathbf{x}), \mathsf{Tensor}(\mathbf{v}) \otimes \mathbf{e}_{j_1} \otimes \ldots \otimes \mathbf{e}_{j_{k-l}}\right\rangle^2\right]$$

$$= \left\|\mathrm{Sym}(\mathsf{Tensor}(\mathbf{v}) \otimes \mathbf{e}_{j_1} \otimes \ldots \otimes \mathbf{e}_{j_{k-l}})\right\|_F^2$$

$$\leq \left\|\mathsf{Tensor}(\mathbf{v}) \otimes \mathbf{e}_{j_1} \otimes \ldots \otimes \mathbf{e}_{j_{k-l}}\right\|_F^2 \leq 1,$$

where the second line is by Fact C.3. Our goal is to apply Fact D.10 with $A = y^2$ and $B = f_{j_1,\ldots,j_{k-l}}(\mathbf{x})^2$. To this aim, we need to bound above the $L_p$-norm of $f_{j_1,\ldots,j_{k-l}}(\mathbf{x})^2$, i.e., $\mathbf{E}_{\mathbf{x}\sim\mathcal{N}_d}[(f_{j_1,\ldots,j_{k-l}}(\mathbf{x})^2)^p]^{1/p}$, which can be done using Fact D.9:

$$\left(\mathbf{E}_{\mathbf{x}\sim\mathcal{N}_d}[(f_{j_1,\ldots,j_{k-l}}(\mathbf{x})^2)^p]\right)^{1/(2p)} \leq (2p-1)^{k/2}\mathbf{E}_{\mathbf{x}\sim\mathcal{N}_d}[f_{j_1,\ldots,j_{k-l}}(\mathbf{x}^{(i)})^2] \leq (2p)^{k/2}.$$

This implies that $\|f_{j_1,\ldots,j_{k-l}}(\mathbf{x})^2\|_{L^p} \leq 2^k p^k$. Thus, using Fact D.10 with $A = y^2$ and $B = f_{j_1,\ldots,j_{k-l}}(\mathbf{x})^2$, we get

$$\mathbf{E}_{(\mathbf{x},y)\sim\mathcal{D}}\left[y^2\left\langle \mathbf{He}_k(\mathbf{x}), \mathsf{Tensor}(\mathbf{v}) \otimes \mathbf{e}_{j_1} \otimes \ldots \otimes \mathbf{e}_{j_{k-l}}\right\rangle^2\right] \leq \mathbf{E}_y[y^2](4e)^k\left\{1, \frac{1}{k}\log\left(\frac{\mathbf{E}_y[y^4]^{1/2}}{\mathbf{E}_y[y^2]}\right)\right\}^k.$$

Finally, using the bound on the moments of the labels as we proved in Lemma D.12, it holds that

$$\mathbf{E}_{(\mathbf{x},y)\sim\mathcal{D}}\left[y^2\left\langle \mathbf{He}_k(\mathbf{x}), \mathsf{Tensor}(\mathbf{v}) \otimes \mathbf{e}_{j_1} \otimes \ldots \otimes \mathbf{e}_{j_{k-l}}\right\rangle^2\right] \lesssim e^k \log^k(B_4/\epsilon).$$

Plugging the bound above back into Equation (33), we obtain:

$$\mathbf{E}_{(\mathbf{x},y)\sim\mathcal{D}}[\mathbf{v}^\top \mathbf{M}^{(i)}(\mathbf{M}^{(i)})^\top \mathbf{v}] = \sum_{j_1,\ldots,j_{k-l}\in[d]} \mathbf{E}_{(\mathbf{x},y)\sim\mathcal{D}}\left[y^2\left\langle \mathbf{He}_k(\mathbf{x}), \mathsf{Tensor}(\mathbf{v}) \otimes \mathbf{e}_{j_1} \otimes \ldots \otimes \mathbf{e}_{j_{k-l}}\right\rangle^2\right]$$

$$\leq e^k d^{k-l} \log^k(B_4/\epsilon).$$

We now proceed to bound above the second term $\mathbf{E}[\mathbf{r}^\top(\mathbf{M}^{(i)})^\top \mathbf{M}^{(i)}\mathbf{r}]$. Using Equation (25), similar calculations yield that

$$\mathbf{E}[\mathbf{r}^\top(\mathbf{M}^{(i)})^\top \mathbf{M}^{(i)}\mathbf{r}] = \mathbf{E}[\|\mathbf{M}^{(i)}\mathbf{r}\|_2^2]$$

$$= \sum_{i_1,\ldots,i_l} \mathbf{E}_{\mathbf{x}\sim\mathcal{N}_d}\left[(y)^2\left\langle \mathbf{He}_k(\mathbf{x}), \mathbf{e}_{i_1} \otimes \ldots \otimes \mathbf{e}_{i_l} \otimes \mathsf{Tensor}(\mathbf{r})\right\rangle^2\right]$$

$$\leq e^k d^l \log^k(B_4/\epsilon) \leq e^k d^{k-l} \log^k(B_4/\epsilon).$$

Thus, plugging in the value of $B_y = \sqrt{B_4/\epsilon}$ from Lemma D.11, the variance $\gamma$ can be bounded by

$$\gamma \lesssim \frac{d^{(k-l)/2} e^k \log^{k/2}(B_4/\epsilon)}{\sqrt{n}} = \sqrt{\frac{d^{k-l} e^k \log^k(B_4/\epsilon)}{n}}. \qquad \square$$

Next, we bound the operator norm $\gamma_*$ from above.

**Claim D.15.** $\gamma_* \lesssim \frac{e^{k/2} \log^{k/2}(B_4/\epsilon)}{\sqrt{n}}$.

*Proof.* By the definition of $\gamma_*$,

$$\gamma_*^2 = \sup_{\|\tilde{\mathbf{v}}\|_2 = \|\tilde{\mathbf{r}}\|_2 = 1} \mathbf{E}\left[\left(\frac{1}{n}\sum_{i=1}^n \tilde{\mathbf{v}}^\top(\mathbf{P}^{(i)} - \mathbf{P})\tilde{\mathbf{r}}\right)^2\right]$$

$$\leq \sup_{\|\tilde{\mathbf{v}}\|_2 = \|\tilde{\mathbf{r}}\|_2 = 1} \frac{1}{n}\mathbf{E}\left[\left(\tilde{\mathbf{v}}^\top(\mathbf{P}^{(i)} - \mathbf{P})\tilde{\mathbf{r}}\right)^2\right]$$

$$\leq \sup_{\|\tilde{\mathbf{v}}\|_2 = \|\tilde{\mathbf{r}}\|_2 = 1} \frac{1}{n}\mathbf{E}\left[\left(\tilde{\mathbf{v}}^\top\mathbf{P}^{(i)}\tilde{\mathbf{r}}\right)^2\right].$$

Decompose $\tilde{\mathbf{v}}$ into $\tilde{\mathbf{v}}^\top = [(\tilde{\mathbf{v}}^{(1)})^\top, (\tilde{\mathbf{v}}^{(2)})^\top]$, where $\tilde{\mathbf{v}}^{(1)} \in \mathbb{R}^{d^l}$ and $\tilde{\mathbf{v}}^{(2)} \in \mathbb{R}^{d^{k-l}}$. Similarly, we can decompose $\tilde{\mathbf{r}}$ into $\tilde{\mathbf{r}}^\top = [(\tilde{\mathbf{r}}^{(1)})^\top, (\tilde{\mathbf{r}}^{(2)})^\top]$ with the same structure. Then, $\tilde{\mathbf{v}}^\top\mathbf{P}^{(i)}\tilde{\mathbf{r}} = y^{(i)}(\tilde{\mathbf{v}}^{(1)\top}\mathbf{M}^{(i)}\tilde{\mathbf{r}}^{(2)} + \tilde{\mathbf{r}}^{(1)\top}\mathbf{M}^{(i)}\tilde{\mathbf{v}}^{(2)})$. Thus, as $(\mathbf{x}^{(i)}, y^{(i)})$ are i.i.d. samples, we can further bound $\gamma_*^2$ by

$$\gamma_*^2 \lesssim \sup_{\substack{\mathbf{v}\in\mathbb{R}^{d^l},\|\mathbf{v}\|_2=1 \\ \mathbf{r}\in\mathbb{R}^{d^{k-l}},\|\mathbf{r}\|_2=1}} \frac{1}{n}\mathbf{E}_{(\mathbf{x},y)\sim\mathcal{D}}\left[y^2(\mathbf{v}^\top\mathsf{Mat}_{(l,k-l)}(\mathbf{He}_k(\mathbf{x}))\mathbf{r})^2\right]$$

$$= \frac{1}{n}\sup_{\substack{\mathbf{v}\in\mathbb{R}^{d^l},\|\mathbf{v}\|_2=1 \\ \mathbf{r}\in\mathbb{R}^{d^{k-l}},\|\mathbf{r}\|_2=1}} \mathbf{E}_{(\mathbf{x},y)\sim\mathcal{D}}\left[y^2\left\langle \mathbf{He}_k(\mathbf{x}), \mathsf{Tensor}(\mathbf{v}) \otimes \mathsf{Tensor}(\mathbf{r})\right\rangle^2\right],$$

where in the last equality we used Equation (27).

Now for any $\|\mathbf{u}\|_2 = \|\mathbf{v}\|_2 = 1$, define

$$f_{(\mathbf{v},\mathbf{u})}(\mathbf{x}) := \Big\langle \mathbf{He}_k(\mathbf{x}), \mathsf{Tensor}(\mathbf{v}) \otimes \mathsf{Tensor}(\mathbf{r}) \Big\rangle,$$

where $\mathbf{v} \in \mathbb{R}^{d^l}, \mathbf{r} \in \mathbb{R}^{d^{k-l}}$, and $f_{(\mathbf{v},\mathbf{u})}$ is a polynomial of $(\mathbf{x}_1, \ldots, \mathbf{x}_d)$ of degree at most $k$. Note that the polynomial $f_{(\mathbf{v},\mathbf{u})}(\mathbf{x})$ satisfies $f_{(\mathbf{v},\mathbf{u})}(\mathbf{x}) \geq 0$ and $\mathbf{E}_{\mathbf{x} \sim \mathcal{N}_d}[(f_{(\mathbf{v},\mathbf{u})}(\mathbf{x}))^2] = \|\mathsf{Tensor}(\mathbf{v}) \otimes \mathsf{Tensor}(\mathbf{r})\|_F^2 = 1$. Similarly to the upper bound on $\gamma$, we apply Fact D.10 with $A$ being $y^2$ and $B$ being $f_{(\mathbf{v},\mathbf{u})}(\mathbf{x})^2$, which yields that for any $\|\mathbf{v}\|_2 = 1, \|\mathbf{u}\|_2 = 1$,

$$\mathop{\mathbf{E}}_{(\mathbf{x},y) \sim \mathcal{D}} \left[ y^2 \Big\langle \mathbf{He}_k(\mathbf{x}), \mathsf{Tensor}(\mathbf{v}) \otimes \mathsf{Tensor}(\mathbf{r}) \Big\rangle^2 \right] \leq e^k \log^k(B_4/\epsilon).$$

Thus, plugging this inequality back into the upper bound on $\gamma_*$ above, we obtain

$$\gamma_*^2 \lesssim \frac{e^k \log^k(B_4/\epsilon)}{n}.$$

Taking the square root on both sides completes the proof of Claim D.15. □

To apply Fact D.8, it remains to bound $\bar{R}$, which we do in the following claim.

**Claim D.16.** $\bar{R} \leq \frac{e^{k/2} \log^{k/2}(B_4/\epsilon)}{\sqrt{n}}$.

*Proof.* By the definition of $\ell_2$ norm, we have

$$\bar{R}^2 = \mathbf{E} \left[ \max_{i \in [n]} \sup_{\|\tilde{\mathbf{v}}\|_2 = \|\tilde{\mathbf{r}}\|_2 = 1} \frac{1}{n^2} (\tilde{\mathbf{v}}^\top (\mathbf{P}^{(i)} - \mathbf{P}) \tilde{\mathbf{r}})^2 \right] \lesssim \frac{1}{n^2} \mathbf{E} \left[ \max_{i \in [n]} \sup_{\|\tilde{\mathbf{v}}\|_2 = \|\tilde{\mathbf{r}}\|_2 = 1} (\tilde{\mathbf{v}}^\top \mathbf{P}^{(i)} \tilde{\mathbf{r}})^2 \right].$$

Let us define

$$f_i(\mathbf{x}^{(i)}) := \|\mathsf{Mat}_{(l,k-l)}(\mathbf{He}_k(\mathbf{x}^{(i)}))\|_2 = \sup_{\|\mathbf{v}\|_2 = \|\mathbf{r}\|_2 = 1} \Big\langle \mathbf{He}_k(\mathbf{x}^{(i)}), \mathsf{Tensor}(\mathbf{v}) \otimes \mathsf{Tensor}(\mathbf{r}) \Big\rangle,$$

where $\mathbf{v} \in \mathbb{R}^{d^l}, \mathbf{r} \in \mathbb{R}^{d^{k-l}}$, and $f_i(\mathbf{x}^{(i)})$ is a polynomial of $(\mathbf{x}_1^{(i)}, \ldots, \mathbf{x}_d^{(i)})$ of degree at most $k$. Using the decomposition of $\tilde{\mathbf{v}}^\top = [(\tilde{\mathbf{v}}^{(1)})^\top, (\tilde{\mathbf{v}}^{(2)})^\top]$ and $\tilde{\mathbf{r}}^\top = [(\tilde{\mathbf{r}}^{(1)})^\top, (\tilde{\mathbf{r}}^{(2)})^\top]$ again, we get

$$\bar{R}^2 \lesssim \frac{1}{n^2} \mathbf{E} \left[ \max_{i \in [n]} \sup_{\mathbf{v} \in \mathbb{B}_{d^l}, \mathbf{r} \in \mathbb{B}_{d^{k-l}}} (y^{(i)})^2 \langle \mathbf{He}_k(\mathbf{x}^{(i)}), \mathsf{Tensor}(\mathbf{v}) \otimes \mathsf{Tensor}(\mathbf{r}) \rangle^2 \right]$$

$$\leq \frac{1}{n^2} \mathbf{E} \left[ \max_{i \in [n]} (y^{(i)})^2 (f_i(\mathbf{x}^{(i)}))^2 \right].$$

Note that the polynomial $f_i(\mathbf{x}^{(i)})$ satisfies $f_i(\mathbf{x}^{(i)}) \geq 0$ and $\mathbf{E}_{\mathbf{x}^{(i)} \sim \mathcal{N}_d}[(f_i(\mathbf{x}^{(i)}))^2] = \|\mathsf{Tensor}(\mathbf{v}) \otimes \mathsf{Tensor}(\mathbf{r})\|_F^2 \leq 1$. Note that $\mathbf{E}[\max_{i \in [n]} Z_i] \leq \sum_{i=1}^n \mathbf{E}[Z_i]$, thus using Fact D.10 we get:

$$\bar{R}^2 \leq \frac{1}{n^2} \sum_{i=1}^n \mathop{\mathbf{E}}_{(\mathbf{x}^{(i)}, y^{(i)}) \sim \mathcal{D}} [(y^{(i)})^2 (f_i(\mathbf{x}^{(i)}))^2] \leq \frac{e^k \log^k(B_4/\epsilon)}{n}.$$

Taking the square root on both sides completes the proof. □

To apply Fact D.8, we need to choose the parameter $R$ such that $\delta = \mathbf{Pr}[\max_{i \in [n]} (1/n) \|\mathbf{P}^{(i)} - \mathbf{P}\|_2 \geq R]$ is sufficiently small. Consider choosing $R$ such that

$$R \gtrsim \frac{e^k \log^k(B_4/\epsilon) d^{(k-l)/4}}{\sqrt{n}} \geq \bar{R}^{1/2} \gamma^{1/2} + \sqrt{2}\bar{R}.$$

To determine $\delta$, recall that from Fact D.9, we have (using Markov's inequality):

$$\mathbf{Pr}[|f_i(\mathbf{x}^{(i)})| \geq t] \leq \frac{\mathbf{E}_{\mathbf{x} \sim \mathcal{N}_d}[|f_i(\mathbf{x}^{(i)})|^p]}{t^p} \leq \frac{p^{kp/2}}{t^p}. \tag{34}$$

Note that $\|\mathbf{P}^{(i)} - \mathbf{P}\|_2 = |y^{(i)}| f_i(\mathbf{x}^{(i)})$, hence

$$\delta = \mathbf{Pr}\left[\max_{i \in [n]} \frac{1}{n}\|\mathbf{P}^{(i)} - \mathbf{P}\|_2 \geq R\right] \leq \mathbf{Pr}\left[\max_{i \in [n]} B_y f_i(\mathbf{x}^{(i)}) \geq nR\right]$$

$$\overset{(i)}{\leq} n\,\mathbf{Pr}\left[B_y f_i(\mathbf{x}^{(i)}) \geq nR\right] \overset{(ii)}{\leq} n\left(\frac{p^{k/2}}{nR/B_y}\right)^p,$$

where in $(i)$ we used a union bound and in $(ii)$ we used Equation (34) with $t = nR/B_y$. Now setting $p^{k/2} = nR/(eB_y)$, we get

$$\delta \leq \exp(-(nR/(B_y e))^{2/k} + \log(n)) \lesssim \exp(-\log^2(B_4/\epsilon)(\epsilon n)^{1/k} d^{1/4}).$$

In summary, applying Fact D.8 with the bound on $\delta$ and $\gamma$ (Claim D.14), $\gamma_*$ (Claim D.15), $\bar{R}$ (Claim D.16), and choosing $t = d^{k/4}$ in Equation (32), we finally get that with probability at least $1 - \exp(-\log^2(1/\epsilon)(\epsilon n)^{1/k} d^{1/4}) - d\exp(-d^{k/4})$, it holds

$$\|\mathbf{M} - \widehat{\mathbf{M}}\|_2 = \|\widehat{\mathbf{P}} - \mathbf{P}\|_2 \lesssim 2\gamma + \gamma_* t^{1/2} + R^{1/3}\gamma^{2/3}t^{2/3} + Rt \lesssim \frac{\log^{k/2}(B_4/\epsilon)d^{(k-l)/2}}{\sqrt{n}}.$$

Therefore, choosing

$$n = \Theta\left(\frac{e^k \log^k(B_4/\epsilon)d^{k-l}}{\epsilon_0^2} + \frac{1}{\epsilon}\right),$$

we have $\|\mathbf{M} - \widehat{\mathbf{M}}\|_2 \leq \epsilon_0$, with probability at least $1 - \exp(-d^{1/2})$.

To complete the proof of Lemma D.13, we apply Wedin's theorem (Fact D.6) and Claim D.7, which together imply that

$$\sin(\theta(\mathbf{v}^*, \widehat{\mathbf{v}}^*)) \leq \frac{\epsilon_0}{(c_k/2 - 4\sqrt{\mathrm{OPT}}) - \epsilon_0}. \tag{35}$$

We then decompose $\widehat{\mathbf{v}}^*$ into $\widehat{\mathbf{v}}^* = a\mathbf{v}^* + b\mathbf{r}$, where $\mathbf{r} \in \mathbb{R}^{d^l}$ such that $\mathbf{r} \perp \mathbf{v}^*$ and $\|\mathbf{r}\|_2 = 1$, and $a^2 + b^2 = 1$. Since $b = \sin(\theta(\mathbf{v}^*, \widehat{\mathbf{v}}^*))$, applying Corollary D.5 we have

$$\begin{aligned}
\widehat{\mathbf{v}}^* \cdot \mathsf{Vec}(\mathbf{w}^{*\otimes l}) &= a\mathbf{v}^* \cdot \mathsf{Vec}(\mathbf{w}^{*\otimes l}) + b\mathbf{r} \cdot \mathsf{Vec}(\mathbf{w}^{*\otimes l}) \\
&\geq \sqrt{1-b^2}(1 - 2\sqrt{\mathrm{OPT}}/c_k) - b \geq (1 - 2\sqrt{\mathrm{OPT}}/c_k) - (2 - 2\sqrt{\mathrm{OPT}}/c_k)b \\
&\geq 1 - \frac{2}{c_k}\sqrt{\mathrm{OPT}} - \frac{2\epsilon_0}{(c_k/2 - 4\sqrt{\mathrm{OPT}}) - \epsilon_0}.
\end{aligned}$$

This completes the proof of Lemma D.13. $\qquad\square$

After getting an approximate top-left singular vector of $\mathsf{Mat}_{(l,k-l)}(\mathbf{E}_{(\mathbf{x},y)\sim\mathcal{D}}[y\mathbf{He}_k(\mathbf{x})])$, $\widehat{\mathbf{v}}^* \in \mathbb{R}^{d^l}$, we show that finding the top-left singular vector of the matrix $\mathsf{Mat}_{(1,l-1)}(\widehat{\mathbf{v}}^*)$ completes the task of computing a vector $\mathbf{u}$ that correlates strongly with $\mathbf{w}^*$.

**Lemma D.17.** *Suppose that $\widehat{\mathbf{v}}^* \cdot \mathsf{Vec}(\mathbf{w}^{*\otimes l}) \geq 1 - \epsilon_1$ for some $\epsilon_1 \in (0, 1/16]$. Then, the top-left singular vector $\mathbf{u} \in \mathbb{R}$ of $\mathsf{Mat}_{(1,l-1)}(\widehat{\mathbf{v}}^*)$ satisfies $\mathbf{u} \cdot \mathbf{w}^* \geq 1 - 2\epsilon_1$.*

*Proof.* Consider the SVD of $\mathsf{Mat}_{(1,l-1)}(\widehat{\mathbf{v}}^*) \in \mathbb{R}^{d \times d^{l-1}}$:

$$\mathsf{Mat}_{(1,l-1)}(\widehat{\mathbf{v}}^*) = \sum_{i=1}^{d} \rho_i \mathbf{u}^{(i)} (\mathbf{r}^{(i)})^\top,$$

where $\mathbf{u}^{(i)} \in \mathbb{R}^d$, $i \in [d]$, and $\mathbf{r}^{(i)} \in \mathbb{R}^{d^{l-1}}$, $i \in [d]$, are two sets of orthonormal vectors. Note that $\{\mathbf{r}^{(1)}, \ldots, \mathbf{r}^{(d)}\}$ is a subset of $\{\mathbf{r}^{(1)}, \ldots, \mathbf{r}^{(d^l)}\}$, which is an orthonormal basis of $\mathbb{R}^{d^l}$. Since $(\mathbf{w}^*)^\top \mathsf{Mat}_{(1,l-1)}(\widehat{\mathbf{v}}^*)\mathsf{Vec}(\mathbf{w}^{*\otimes l-1}) = \widehat{\mathbf{v}}^* \cdot \mathsf{Vec}(\mathbf{w}^{*\otimes l}) \geq 1 - \epsilon_1$, we have $\rho_1 \geq 1 - \epsilon_1$, and thus

$$(\mathbf{w}^*)^\top \mathsf{Mat}_{(1,l-1)}(\widehat{\mathbf{v}}^*)\mathsf{Vec}(\mathbf{w}^{*\otimes l-1}) = \sum_{i=1}^{d} \rho_i (\mathbf{w}^* \cdot \mathbf{u}^{(i)})(\mathsf{Vec}(\mathbf{w}^{*\otimes l-1}) \cdot \mathbf{r}^{(i)}) \geq 1 - \epsilon_1. \tag{36}$$

Since $\mathbf{u}^{(i)}$, $i \in [d]$ and $\mathbf{r}^{(i)}$, $i \in [d^{l-1}]$ form orthonormal bases of $\mathbb{R}^d$ and $\mathbb{R}^{d^l}$ respectively, we can decompose $\mathbf{w}^*$ and $\mathsf{Vec}(\mathbf{w}^{*\otimes l-1})$ in these bases respectively:

$$\mathbf{w}^* = \sum_{i=1}^{d} a_i \mathbf{u}^{(i)}, \quad \mathsf{Vec}(\mathbf{w}^{*\otimes l-1}) = \sum_{i=1}^{d^{l-1}} b_i \mathbf{r}^{(i)}.$$

Note that since $\|\mathbf{w}^*\|_2 = 1$ and $\|\mathsf{Vec}(\mathbf{w}^{*\otimes l-1})\|_2^2 = \|\mathbf{w}^{*\otimes l-1}\|_F^2 = 1$, we have $\sum_i a_i^2 = 1$ and $\sum_i b_i^2 = 1$. In addition, since $\|\widehat{\mathbf{v}}^*\|_2 = 1$, we have $\|\mathsf{Mat}_{(1,l-1)}(\widehat{\mathbf{v}}^*)\|_F^2 = \sum_{i=1}^d \rho_i^2 = 1$. Therefore, plugging the decomposition above back into Equation (36), we get

$$1 - \epsilon_1 \le \sum_{i=1}^{d} \rho_i a_i b_i \le \rho_1 a_1 b_1 + \rho_2 \sqrt{\sum_{i=2}^{d} a_i^2} \sqrt{\sum_{i=2}^{d} b_i^2}$$

$$\le \rho_1 a_1 b_1 + \sqrt{1 - \rho_1^2}\sqrt{1 - a_1^2}\sqrt{1 - b_1^2} \le \rho_1 a_1 b_1 + \sqrt{1 - \rho_1^2}(1 - a_1 b_1).$$

When $\rho_1 \ge 1 - \epsilon_1 \ge \sqrt{2}/2$, we have $\rho_1 - \sqrt{1 - \rho_1^2} \ge 0$ and then it holds that

$$a_1 b_1 \ge \frac{1 - \sqrt{1 - \rho_1^2} - \epsilon_1}{\rho_1 - \sqrt{1 - \rho_1^2}} := g(\rho_1).$$

We show that when $0 \le \epsilon_1 \le 1/16$ (which implies that $15/16 \le \rho_1 \le 1$), it holds that $g(\rho_1) \ge 1 - 2\epsilon_1$. By the definition of $g(\rho_1)$, it suffices to argue that

$$1 - \rho_1 \ge \epsilon_1(1 - 2\rho_1 + 2\sqrt{1 - \rho_1^2}).$$

This follows by direct calculations as when $\rho_1 \in [15/16, 1]$, $1 - 2\rho_1 + 2\sqrt{1 - \rho_1^2} \le 0$.

Therefore, when $\epsilon_1 \le 1/16$, it holds that $a_1 b_1 \ge 1 - 2\epsilon_1$. Since $0 \le a_1, b_1 \le 1$, it must be that $a_1 \ge 1 - 2\epsilon_1$; this further implies that $\mathbf{w}^* \cdot \mathbf{u} \ge 1 - 2\epsilon_1$. $\qquad\square$

### D.3 Proof of Proposition D.2

*Proof of Proposition D.2.* Since $\sqrt{\mathrm{OPT}} \le c_{k^*}/(64k^*) \le c_{k^*}/64$, choosing $\epsilon_0 = c_{k^*}/(256k^*) \le c_{k^*}/256$ in Lemma D.13, we obtain that using $n = \Theta((k^*)^2 e^{k^*} \log^{k^*}(B_4/\epsilon)d^{\lceil k^*/2\rceil}/(c_{k^*}^2) + 1/\epsilon)$, it holds with probability at least $1 - \exp(-d^{1/2})$ that

$$\widehat{\mathbf{v}}^* \cdot \mathsf{Vec}(\mathbf{w}^{*\otimes l}) \ge 1 - \frac{2}{c_k}\sqrt{\mathrm{OPT}} - \frac{2\epsilon_0}{(c_k/2 - 4\sqrt{\mathrm{OPT}}) - \epsilon_0}$$

$$\ge 1 - \frac{1}{32k^*} - \frac{c_{k^*}/(128k^*)}{c_{k^*}/2 - c_{k^*}/16 - c_{k^*}/256} \ge 1 - \frac{1}{16k^*}.$$

Then applying Lemma D.17 with $\epsilon_1 \le 1/(16k^*) \le 1/16$ we get that the output $\mathbf{u}$ of Algorithm 3 satisfies $\mathbf{u} \cdot \mathbf{w}^* \ge 1 - 2\epsilon_1 \ge 1 - 1/(8k^*) \ge 1 - \min\{1/k^*, 1/2\}$, completing the proof. $\qquad\square$

### D.4 Proof of Proposition D.3

*Proof of Proposition D.3.* Since $\sqrt{\mathrm{OPT}} \le c_{k^*}/64$ and $\epsilon \le 1/64$, choosing $\epsilon_0 = c_{k^*}\epsilon/16$ in Lemma D.13, we obtain that using $n = \Theta(e^{k^*}\log^{k^*}(B_4/\epsilon)d^{\lceil k^*/2\rceil}/(c_{k^*}^2\epsilon^2) + 1/\epsilon)$ samples, it holds with probability at least $1 - \exp(-d^{1/2})$ that $\widehat{\mathbf{v}}^* \cdot \mathsf{Vec}(\mathbf{w}^{*\otimes l}) \ge 1 - (2/c_{k^*})\sqrt{\mathrm{OPT}} + \epsilon/3 (\ge 15/16)$. Then applying Lemma D.17 with $\epsilon_1 = (2/c_{k^*})\sqrt{\mathrm{OPT}} - \epsilon/3$, we get that the output $\mathbf{w}^0$ of Algorithm 3 satisfies $\mathbf{w}^0 \cdot \mathbf{w}^* \ge 1 - 2((2/c_{k^*})\sqrt{\mathrm{OPT}} + \epsilon/3)$.

Finally, to show the upper bound on the $L_2^2$ loss of $\mathbf{w}^0$, we bring in the definition of the $L_2^2$ loss $\mathcal{L}_2^\sigma(\mathbf{w}^0)$, which yields

$$\mathcal{L}_2^\sigma(\mathbf{w}^0) \leq 2\mathrm{OPT} + 2\mathcal{L}_2^{*\sigma}(\mathbf{w}^0) = 2\mathrm{OPT} + 2\left(1 - \sum_{k \geq k^*} c_k^2(\mathbf{w}^0 \cdot \mathbf{w}^*)^k\right)$$

$$= 2\mathrm{OPT} + 2\left(\sum_{k \geq k^*} c_k^2(1 - (\mathbf{w}^0 \cdot \mathbf{w}^*)^k)\right)$$

$$= 2\mathrm{OPT} + 2\left(\sum_{k \geq k^*} c_k^2(1 - (\mathbf{w}^0 \cdot \mathbf{w}^*))(1 + (\mathbf{w}^0 \cdot \mathbf{w}^*) + \cdots + (\mathbf{w}^0 \cdot \mathbf{w}^*)^{k-1})\right)$$

$$\leq 2\mathrm{OPT} + 2\left(\sum_{k \geq k^*} kc_k^2(1 - (\mathbf{w}^0 \cdot \mathbf{w}^*))\right) \lesssim \left(\sum_{k \geq k^*} kc_k^2\right)\left(\frac{4}{c_{k^*}}\sqrt{\mathrm{OPT}} + \epsilon/3\right).$$

Since $\sum_{k \geq k^*} kc_k^2 \leq C_{k^*}$ by Assumption 1$(iii)$, this completes the proof of Proposition D.3. $\qquad\square$

# E    Full Version of Section 3

After getting an initialized vector $\mathbf{w}^0$ using Algorithm 1, we run Riemannian minibatch SGD Algorithm 2 on the 'truncated loss'. In the following sections, we will first present the definition of the truncated $L_2^2$ loss $\mathcal{L}_2^\phi$ and its Riemannian gradient, then we will proceed to show that Algorithm 2 converges to a constant approximate solution in $O(\log(1/\epsilon))$ iterations.

---

**Algorithm 4** Riemannian GD with Warm-start

---

1: **Input:** Parameters $\epsilon, k^*, c_{k^*}, B_4 > 0; T, \eta$; Sample access to $\mathcal{D}$.
2: $\mathbf{w}^0 = \textbf{Initialization}[\epsilon, k^*, c_{k^*}, B_4, \epsilon_0 = c_{k^*}/(256k^*)]$.
3: **for** $t = 0, \ldots, T - 1$ **do**
4:    Draw $n = \Theta(C_{k^*}de^{k^*}\log^{k^*+1}(B_4/\epsilon)/(\epsilon\delta))$ samples from $\mathcal{D}$ and compute

$$\widehat{\mathbf{g}}(\mathbf{w}^t) = \frac{1}{n}\sum_{i=1}^{n} k^* c_{k^*} y^{(i)}(\mathbf{I} - \mathbf{w}^t(\mathbf{w}^t)^\top)\langle\mathbf{He}_{k^*}(\mathbf{x}^{(i)}), (\mathbf{w}^t)^{\otimes k^*-1}\rangle.$$

5:    $\mathbf{w}^{t+1} = (\mathbf{w}^t - \eta\widehat{\mathbf{g}}(\mathbf{w}^t))/\|\mathbf{w}^t - \eta\widehat{\mathbf{g}}(\mathbf{w}^t)\|_2$.
6: **Return:** $\mathbf{w}^T$.

---

## E.1    Truncated Loss and the Sharpness property of the Riemannian Gradient

Instead of directly minimizing the $L_2^2$ loss $\mathcal{L}_2^\sigma$, we work with the following truncated loss that drops all the terms higher than $k^*$ in the polynomial expansion of $\sigma$:

$$\mathcal{L}_2^\phi(\mathbf{w}) := 2\big(1 - \mathop{\mathbf{E}}_{(\mathbf{x},y)\sim\mathcal{D}}[y\phi(\mathbf{w} \cdot \mathbf{x})]\big), \text{ where } \phi(\mathbf{w} \cdot \mathbf{x}) = \langle\mathbf{He}_{k^*}(\mathbf{x}), \mathbf{w}^{\otimes k^*}\rangle. \tag{37}$$

Similarly, the noiseless surrogate loss is defined as

$$\mathcal{L}_2^{*\phi}(\mathbf{w}) := 2\big(1 - \mathop{\mathbf{E}}_{(\mathbf{x},y)\sim\mathcal{D}}[\sigma(\mathbf{w}^* \cdot \mathbf{x})\phi(\mathbf{w} \cdot \mathbf{x})]\big) = 2\big(1 - c_{k^*}(\mathbf{w} \cdot \mathbf{w}^*)^{k^*}\big). \tag{38}$$

Using Fact C.1$(2)$, the gradient of the truncated $L_2^2$ loss equals:

$$\nabla\mathcal{L}_2^\phi(\mathbf{w}) = -2\mathop{\mathbf{E}}_{(\mathbf{x},y)\sim\mathcal{D}}[\nabla\phi(\mathbf{w} \cdot \mathbf{x})y] = -2\mathop{\mathbf{E}}_{(\mathbf{x},y)\sim\mathcal{D}}\big[k^* c_{k^*} y\langle\mathbf{He}_{k^*}(\mathbf{x}), \mathbf{w}^{\otimes k^*-1}\rangle\big], \tag{39}$$

while for the gradient of the noiseless $L_2^2$ loss we have

$$\nabla\mathcal{L}_2^{*\phi}(\mathbf{w}) = -2\mathop{\mathbf{E}}_{(\mathbf{x},y)\sim\mathcal{D}}\big[k^* c_{k^*}\sigma(\mathbf{w}^* \cdot \mathbf{x})\langle\mathbf{He}_{k^*}(\mathbf{x}), \mathbf{w}^{\otimes k^*-1}\rangle\big]. \tag{40}$$

Recall that $\mathbf{P}_{\mathbf{w}^\perp} := \mathbf{I} - \mathbf{w}\mathbf{w}^\top$. Then the Riemannian gradient of the $L_2^2$ loss $\mathcal{L}_2^\phi$, denoted by $\mathbf{g}(\mathbf{w})$ is

$$\mathbf{g}(\mathbf{w}) := \mathbf{P}_{\mathbf{w}^\perp}(\nabla\mathcal{L}_2^\phi(\mathbf{w})) = -2 \mathop{\mathbf{E}}_{(\mathbf{x},y)\sim\mathcal{D}} \left[ k^* y \mathbf{P}_{\mathbf{w}^\perp} \langle \mathbf{He}_{k^*}(\mathbf{x}), \mathbf{w}^{\otimes k^*-1} \rangle \right]. \tag{41}$$

Similarly, the Riemannian gradient of the noiseless $L_2^2$ loss $\mathcal{L}_2^{*\phi}$ is defined by

$$\mathbf{g}^*(\mathbf{w}) := \mathbf{P}_{\mathbf{w}^\perp}(\nabla\mathcal{L}_2^{*\phi}(\mathbf{w})) = -2 \mathop{\mathbf{E}}_{(\mathbf{x},y)\sim\mathcal{D}} \left[ k^* \sigma(\mathbf{w}^* \cdot \mathbf{x}) \mathbf{P}_{\mathbf{w}^\perp} \langle \mathbf{He}_{k^*}(\mathbf{x}), \mathbf{w}^{\otimes k^*-1} \rangle \right]. \tag{42}$$

The following claim establishes that $\mathbf{g}^*(\mathbf{w})$ carries information about the alignment between vectors $\mathbf{w}$ and $\mathbf{w}^*$.

**Claim E.1.** *For any $\mathbf{w} \in \mathbb{S}^{d-1}$, we have $\mathbf{g}^*(\mathbf{w}) = -2k^* c_{k^*}(\mathbf{w} \cdot \mathbf{w}^*)^{k^*-1}(\mathbf{w}^*)^{\perp\mathbf{w}}$.*

*Proof.* Using the definition of $\mathbf{g}^*(\mathbf{w})$ from Equation (42), a direct calculation shows that

$$\mathbf{g}^*(\mathbf{w}) \cdot \frac{(\mathbf{w}^*)^{\perp\mathbf{w}}}{\|(\mathbf{w}^*)^{\perp\mathbf{w}}\|_2}$$

$$\overset{(i)}{=} -2k^* \mathop{\mathbf{E}}_{\mathbf{x}\sim\mathcal{N}_d} \left[ \sum_{k\geq k^*} c_k \left\langle \mathbf{He}_k(\mathbf{x}), \mathbf{w}^{*\otimes k} \right\rangle \cdot \left\langle \mathbf{He}_{k^*}(\mathbf{x}), \mathbf{w}^{\otimes k^*-1} \otimes \frac{(\mathbf{w}^*)^{\perp\mathbf{w}}}{\|(\mathbf{w}^*)^{\perp\mathbf{w}}\|_2} \right\rangle \right]$$

$$\overset{(ii)}{=} -2k^* c_{k^*} \left\langle \mathrm{Sym}(\mathbf{w}^{*\otimes k^*}), \mathrm{Sym}\left( \mathbf{w}^{\otimes k^*-1} \otimes \frac{(\mathbf{w}^*)^{\perp\mathbf{w}}}{\|(\mathbf{w}^*)^{\perp\mathbf{w}}\|_2} \right) \right\rangle$$

$$\overset{(iii)}{=} -2k^* c_{k^*}(\mathbf{w} \cdot \mathbf{w}^*)^{k^*-1}\|(\mathbf{w}^*)^{\perp\mathbf{w}}\|_2,$$

where $(i)$ is by the definition of $\mathbf{g}^*(\mathbf{w})$ and $\sigma(\mathbf{w}^* \cdot \mathbf{x})$, $(ii)$ is by Fact C.3, and $(iii)$ is by Fact C.1(1), Equation (20), and $\|\mathbf{w}^*\|_2 = 1$. Let $\mathbf{v} \in \mathbb{R}^d$ be any unit vector that is orthogonal to $(\mathbf{w}^*)^{\perp\mathbf{w}}$. Observe that $\mathbf{v}^{\perp\mathbf{w}} \cdot \mathbf{w}^* = \mathbf{v} \cdot (\mathbf{w}^*)^{\perp\mathbf{w}} = 0$. Thus, we have

$$\mathbf{g}^*(\mathbf{w}) \cdot \mathbf{v} = -2k^* c_{k^*} \left\langle \mathrm{Sym}(\mathbf{v}^{\perp\mathbf{w}} \otimes \mathbf{w}^{\otimes k^*-1}), \mathrm{Sym}(\mathbf{w}^{*\otimes k^*}) \right\rangle$$

$$= -2k^* c_{k^*}(\mathbf{w} \cdot \mathbf{w}^*)^{k^*-1}(\mathbf{v}^{\perp\mathbf{w}} \cdot \mathbf{w}^*) = 0.$$

This implies that $\mathbf{g}^*(\mathbf{w})$ is parallel to $(\mathbf{w}^*)^{\perp\mathbf{w}}$ and thus $\mathbf{g}^*(\mathbf{w}) = -2k^* c_{k^*}(\mathbf{w}\cdot\mathbf{w}^*)^{k^*-1}(\mathbf{w}^*)^{\perp\mathbf{w}}$. $\quad\square$

Let us denote the difference between the noisy and the noiseless Riemannian gradient by $\xi(\mathbf{w})$:

$$\xi(\mathbf{w}) := \mathbf{g}(\mathbf{w}) - \mathbf{g}^*(\mathbf{w}) = -2 \mathop{\mathbf{E}}_{(\mathbf{x},y)\sim\mathcal{D}}[(y - \sigma(\mathbf{w}^* \cdot \mathbf{x}))\mathbf{P}_{\mathbf{w}^\perp}\nabla\phi(\mathbf{w} \cdot \mathbf{x})] .$$

We next show that the norm of $\xi(\mathbf{w})$ and the inner product between $\xi(\mathbf{w})$ and $\mathbf{w}^*$ are both bounded:

**Lemma E.2.** *Let $\xi(\mathbf{w}) = \mathbf{g}(\mathbf{w}) - \mathbf{g}^*(\mathbf{w})$ as defined above. Then,*

$$\|\xi(\mathbf{w})\|_2 \leq 2k^* c_{k^*}\sqrt{\mathrm{OPT}} \quad and \quad |\xi(\mathbf{w}) \cdot \mathbf{w}^*| \leq 2k^* c_{k^*}\sqrt{\mathrm{OPT}}\|(\mathbf{w}^*)^{\perp\mathbf{w}}\|_2.$$

*Proof.* Using the definition of $\xi(\mathbf{w})$ and the definition of the 2-norm,

$$\|\xi(\mathbf{w})\|_2 = 2k^* c_{k^*} \max_{\mathbf{v}\in\mathbb{S}^{d-1}} \mathop{\mathbf{E}}_{(\mathbf{x},y)\sim\mathcal{D}} \left[ (y - \sigma(\mathbf{w}^* \cdot \mathbf{x}))\mathbf{P}_{\mathbf{w}^\perp} \langle \mathbf{He}_{k^*}(\mathbf{x}), \mathbf{w}^{\otimes k^*-1} \rangle \cdot \mathbf{v} \right]$$

$$= 2k^* c_{k^*} \max_{\mathbf{v}\in\mathbb{S}^{d-1}} \mathop{\mathbf{E}}_{(\mathbf{x},y)\sim\mathcal{D}} \left[ (y - \sigma(\mathbf{w}^* \cdot \mathbf{x}))\langle \mathbf{He}_{k^*}(\mathbf{x}), \mathbf{v}^{\perp\mathbf{w}} \otimes \mathbf{w}^{\otimes k^*-1} \rangle \right]$$

$$\leq 2k^* c_{k^*} \max_{\mathbf{v}\in\mathbb{S}^{d-1}} \sqrt{\mathop{\mathbf{E}}_{(\mathbf{x},y)\sim\mathcal{D}}[(y - \sigma(\mathbf{w}^* \cdot \mathbf{x}))^2] \mathop{\mathbf{E}}_{\mathbf{x}\sim\mathcal{N}_d} \left[ \left( \langle \mathbf{He}_{k^*}(\mathbf{x}), \mathbf{v}^{\perp\mathbf{w}} \otimes \mathbf{w}^{\otimes k^*-1} \rangle \right)^2 \right]}$$

$$= 2k^* c_{k^*} \max_{\mathbf{v}\in\mathbb{S}^{d-1}} \sqrt{\mathrm{OPT}}\|\mathrm{Sym}(\mathbf{v}^{\perp\mathbf{w}} \otimes \mathbf{w}^{\otimes k^*-1})\|_F,$$

where the (only) inequality is by Cauchy-Schwarz, and the last equality is by Fact C.3. As a tensor $\mathbf{A}$, it holds $\|\mathrm{Sym}(\mathbf{A})\|_F \leq \|\mathbf{A}\|_F$, we have

$$\|\mathrm{Sym}(\mathbf{v}^{\perp\mathbf{w}} \otimes \mathbf{w}^{\otimes k^*-1})\|_F^2 \leq \|\mathbf{v}^{\perp\mathbf{w}} \otimes \mathbf{w}^{\otimes k^*-1}\|_F^2 = \|\mathbf{v}^{\perp\mathbf{w}}\|_2^2 \leq 1,$$

which then implies that[4]

$$\|\xi(\mathbf{w})\|_2 \le 2k^* c_{k^*} \sqrt{\mathrm{OPT}}.$$

Following the same line of argument as above, we also get

$$|\xi(\mathbf{w}) \cdot \mathbf{w}^*| \le 2\sqrt{\mathrm{OPT}}\sqrt{(k^* c_{k^*})^2 \|(\mathbf{w}^*)^{\perp \mathbf{w}} \otimes \mathbf{w}^{\otimes k^*-1}\|_F^2} = 2k^* c_{k^*} \sqrt{\mathrm{OPT}}\|(\mathbf{w}^*)^{\perp \mathbf{w}}\|_2.$$

This completes the proof of Lemma E.2. $\qquad\square$

As a direct corollary of Lemma E.2, we now show that the norm of the noisy gradient $\|\mathbf{g}(\mathbf{x})\|_2$ is close to the norm of the noiseless gradient $\|\mathbf{g}^*(\mathbf{w})\|_2$.

**Corollary E.3.** *For any* $\mathbf{w} \in \mathbb{S}^{d-1}$, $\|\mathbf{g}(\mathbf{w})\|_2 \le 2k^* c_{k^*}\sqrt{\mathrm{OPT}} + \|\mathbf{g}^*(\mathbf{w})\|_2$.

*Proof.* Follows by the triangle inequality, as $\|\mathbf{g}(\mathbf{w})\|_2 \le \|\xi(\mathbf{w})\|_2 + \|\mathbf{g}^*(\mathbf{w})\|_2$. $\qquad\square$

We are now ready to present the main structural result of this section.

**Lemma E.4** (Sharpness). *Assume* $\mathrm{OPT} \le c/(4e)^2$ *for some small absolute constant* $c < 1$. *Let* $\mathbf{w} \in \mathbb{R}^d$ *such that* $\|\mathbf{w}\|_2 = 1$ *and suppose that* $\mathbf{w} \cdot \mathbf{w}^* \ge 1 - 1/k^*$. *Let* $\theta := \theta(\mathbf{w}, \mathbf{w}^*)$. *If* $\sin\theta \ge 4e\sqrt{\mathrm{OPT}}$, *then*

$$\mathbf{g}(\mathbf{w}) \cdot \mathbf{w}^* \le -\frac{1}{2}\|\mathbf{g}^*(\mathbf{w})\|_2 \sin\theta.$$

*Proof.* We start by noticing that by Claim E.1, the noiseless gradient satisfies the following property:

$$\mathbf{g}^*(\mathbf{w}) \cdot \mathbf{w}^* = -2k^* c_{k^*}(\mathbf{w} \cdot \mathbf{w}^*)^{k^*-1}\|(\mathbf{w}^*)^{\perp \mathbf{w}}\|_2^2 = -\|\mathbf{g}^*(\mathbf{w})\|_2 \sin\theta,$$

where we used that since $\|\mathbf{w}\|_2 = \|\mathbf{w}^*\|_2 = 1$, we have $\|(\mathbf{w}^*)^{\perp \mathbf{w}}\|_2 = \sin\theta$. Furthermore, applying Lemma E.2 we have the following sharpness property with respect to the $L_2^2$ loss:

$$\mathbf{g}(\mathbf{w}) \cdot \mathbf{w}^* = \mathbf{g}^*(\mathbf{w}) \cdot \mathbf{w}^* + \xi(\mathbf{w}) \cdot \mathbf{w}^* \le -(\|\mathbf{g}^*(\mathbf{w})\|_2 - 2k^* c_{k^*}\sqrt{\mathrm{OPT}})\sin\theta. \qquad (43)$$

Observe that $(1 - 1/t)^{t-1} \ge 1/e$ for all $t \ge 1$. Therefore, when $\mathbf{w} \cdot \mathbf{w}^* \ge 1 - 1/k^*$, the norm of the gradient vector $\mathbf{g}^*$ satisfies

$$\|\mathbf{g}^*(\mathbf{w})\|_2 = 2k^* c_{k^*}(\mathbf{w} \cdot \mathbf{w}^*)^{k^*-1}\sin\theta \ge 2k^* c_{k^*}(1 - 1/k^*)^{k^*-1}\sin\theta \ge e^{-1}k^* c_{k^*}\sin\theta.$$

therefore, when $\sin\theta \ge 4e\sqrt{\mathrm{OPT}}$ and $\mathbf{w} \cdot \mathbf{w}^* \ge 1 - 1/k^*$, we have

$$\|\mathbf{g}^*(\mathbf{w})\|_2 \ge 4k^* c_{k^*}\sqrt{\mathrm{OPT}}.$$

Thus, as long as $\sin\theta \ge 4e\sqrt{\mathrm{OPT}}$, we have that $\mathbf{g}(\mathbf{w}) \cdot \mathbf{w}^* \le -\frac{1}{2}\|\mathbf{g}^*(\mathbf{w})\|_2 \sin\theta$. $\qquad\square$

## E.2 Concentration of Gradients

For notational simplicity, define:

$$\mathbf{g}(\mathbf{w}; \mathbf{x}^{(i)}, y^{(i)}) := k^* c_{k^*} y^{(i)} \mathbf{P}_{\mathbf{w}^\perp}\langle \mathbf{He}_{k^*}(\mathbf{x}^{(i)}), \mathbf{w}^{\otimes k^*-1}\rangle. \qquad (44)$$

Then the empirical estimate of $\mathbf{g}(\mathbf{w})$ is

$$\widehat{\mathbf{g}}(\mathbf{w}) := \frac{1}{n}\sum_{i=1}^n \mathbf{g}(\mathbf{w}; \mathbf{x}^{(i)}, y^{(i)}) = \frac{1}{n}\sum_{i=1}^n k^* c_{k^*} y^{(i)} \mathbf{P}_{\mathbf{w}^\perp}\langle \mathbf{He}_{k^*}(\mathbf{x}^{(i)}), \mathbf{w}^{\otimes k^*-1}\rangle. \qquad (45)$$

The following lemma provides the upper bounds on the number of samples required to approximate the Riemannian gradient $\mathbf{g}(\mathbf{w})$ by $\widehat{\mathbf{g}}(\mathbf{w})$.

---

[4]Note here that if we had not used the truncation of the activation, then the bound on the error term $\xi(\mathbf{w})$ we could get would be $\|\xi(\mathbf{w})\|_2 \le (\sum_{k \ge k^*} c_k^2 k^2)^{1/2}\sqrt{\mathrm{OPT}}$.

**Lemma E.5** (Concentration of Gradients). *Let $\mathbf{w}^*, \mathbf{w} \in \mathbb{S}^{d-1}$. Let $\widehat{\mathbf{g}}(\mathbf{w})$ be the empirical estimate of the Riemannian gradient. Furthermore, denote the angle between $\mathbf{w}$ and $\mathbf{w}^*$ by $\theta$. Then, with probability at least $1 - \delta$ it holds*

$$\|\widehat{\mathbf{g}}(\mathbf{w}) - \mathbf{g}(\mathbf{w})\|_2 \lesssim \sqrt{\frac{d(k^* c_{k^*})^2 e^{k^*} \log^{k^*}(B_4/\epsilon)}{n\delta}};$$

$$(\widehat{\mathbf{g}}(\mathbf{w}) - \mathbf{g}(\mathbf{w})) \cdot \mathbf{w}^* \lesssim \sqrt{\frac{(k^* c_{k^*})^2 e^{k^*} \log^{k^*}(B_4/\epsilon) \sin^4(\theta)}{n\delta}}.$$

*Proof.* By Chebyshev's inequality, we have

$$\mathbf{Pr}\left[\left\|\frac{1}{n}\sum_{i=1}^n \mathbf{g}(\mathbf{w}; \mathbf{x}^{(i)}, y^{(i)}) - \mathbf{g}(\mathbf{w})\right\|_2 \geq t\right] \leq \frac{1}{t^2} \underset{(\mathbf{x},y)\sim\mathcal{D}}{\mathbf{E}}\left[\left\|\frac{1}{n}\sum_{i=1}^n \mathbf{g}(\mathbf{w}; \mathbf{x}^{(i)}, y^{(i)}) - \mathbf{g}(\mathbf{w})\right\|_2^2\right]$$

$$\leq \frac{1}{nt^2} \underset{(\mathbf{x},y)\sim\mathcal{D}}{\mathbf{E}}\left[\left\|\mathbf{g}(\mathbf{w}; \mathbf{x}, y) - \mathbf{g}(\mathbf{w})\right\|_2^2\right]. \tag{46}$$

Let $\mathbf{e}_j$ be the $j^{\text{th}}$ basis of $\mathbb{R}^d$, we have $\|\mathbf{g}(\mathbf{w}; \mathbf{x}, y) - \mathbf{g}(\mathbf{w})\|_2^2 = \sum_{j=1}^d (\mathbf{g}(\mathbf{w}; \mathbf{x}, y) \cdot \mathbf{e}_j - \mathbf{g}(\mathbf{w}) \cdot \mathbf{e}_j)^2$. Thus, it suffices to bound the expectation of each summand $(\mathbf{g}(\mathbf{w}; \mathbf{x}, y) \cdot \mathbf{e}_j - \mathbf{g}(\mathbf{w}) \cdot \mathbf{e}_j)^2$, for $j \in [d]$. Note first that since $\mathbf{g}(\mathbf{w}) = \mathbf{E}_{(\mathbf{x},y)\sim\mathcal{D}}[\mathbf{g}(\mathbf{w}; \mathbf{x}, y)]$, we have

$$\underset{(\mathbf{x},y)\sim\mathcal{D}}{\mathbf{E}}[(\mathbf{g}(\mathbf{w}; \mathbf{x}, y) \cdot \mathbf{e}_j - \mathbf{g}(\mathbf{w}) \cdot \mathbf{e}_j)^2] \leq \underset{(\mathbf{x},y)\sim\mathcal{D}}{\mathbf{E}}[(\mathbf{g}(\mathbf{w}; \mathbf{x}, y) \cdot \mathbf{e}_j)^2]$$

$$= 4(k^* c_{k^*})^2 \underset{(\mathbf{x},y)\sim\mathcal{D}}{\mathbf{E}}\left[y^2 \left\langle \mathbf{He}_{k^*}(\mathbf{x}), \mathbf{e}_j^{\perp\mathbf{w}} \otimes \mathbf{w}^{\otimes k^*-1} \right\rangle^2\right].$$

Denote $f_j(\mathbf{x}) := \langle \mathbf{He}_{k^*}(\mathbf{x}), \mathbf{e}_j^{\perp\mathbf{w}} \otimes \mathbf{w}^{\otimes k^*-1}\rangle$, which is a polynomial of $\mathbf{x}$ of degree $k^*$. In addition, note that $\mathbf{E}_{\mathbf{x}\sim\mathcal{N}_d}[f_j(\mathbf{x})^2] = \|\mathbf{e}_j^{\perp\mathbf{w}} \otimes \mathbf{w}^{\otimes k^*-1}\|_F^2 \leq \|\mathbf{e}_j^{\perp\mathbf{w}}\|_2^2 \leq 1$. Therefore, applying Fact D.9 and Fact D.10 with $A = y^2$ and $B = f_j(\mathbf{x})^2$, we get

$$\underset{(\mathbf{x},y)\sim\mathcal{D}}{\mathbf{E}}\left[y^2 \left\langle \mathbf{He}_{k^*}(\mathbf{x}), \mathbf{e}_j^{\perp\mathbf{w}} \otimes \mathbf{w}^{\otimes k^*-1} \right\rangle^2\right] \leq \underset{(\mathbf{x},y)\sim\mathcal{D}}{\mathbf{E}}[y^2](4e)^{k^*} \max\left\{1, \frac{1}{k^*}\log(B_4/\epsilon)\right\}^{k^*}$$

$$\lesssim e^{k^*} \log^{k^*}(B_4/\epsilon).$$

Thus, it holds that $\mathbf{E}_{(\mathbf{x},y)\sim\mathcal{D}}[(\mathbf{g}(\mathbf{w}; \mathbf{x}, y) \cdot \mathbf{e}_j - \mathbf{g}(\mathbf{w}) \cdot \mathbf{e}_j)^2] \lesssim (k^* c_{k^*})^2 e^{k^*} \log^{k^*}(B_4/\epsilon)$, which further implies that the expectation of $\|\mathbf{g}(\mathbf{w}; \mathbf{x}, y) - \mathbf{g}(\mathbf{w})\|_2^2$ is bounded above by

$$\underset{(\mathbf{x},y)\sim\mathcal{D}}{\mathbf{E}}[\|\mathbf{g}(\mathbf{w}; \mathbf{x}, y) - \mathbf{g}(\mathbf{w})\|_2^2] \lesssim d(k^* c_{k^*})^2 e^{k^*} \log^{k^*}(B_4/\epsilon).$$

Plugging this back into the Chebyshev's bound Equation (46), we obtain

$$\mathbf{Pr}[\|\widehat{\mathbf{g}}(\mathbf{w}) - \mathbf{g}(\mathbf{w})\|_2 \geq t] \leq \frac{d(k^* c_{k^*})^2 e^{k^*} \log^{k^*}(B_4/\epsilon)}{nt^2}.$$

Therefore, with probability at least $1 - \delta$, it holds

$$\|\widehat{\mathbf{g}}(\mathbf{w}) - \mathbf{g}(\mathbf{w})\|_2 \leq \sqrt{\frac{d(k^* c_{k^*})^2 e^{k^*} \log^{k^*}(B_4/\epsilon)}{n\delta}}.$$

Now for the second statement of Lemma E.5, we similarly apply Chebyshev's inequality, which yields

$$\mathbf{Pr}\left[\left|\frac{1}{n}\sum_{i=1}^{n}\mathbf{g}(\mathbf{w};\mathbf{x}^{(i)},y^{(i)})\cdot\mathbf{w}^*-\mathbf{g}(\mathbf{w})\cdot\mathbf{w}^*\right|\geq t\right]$$

$$\leq\frac{1}{t^2}\mathop{\mathbf{E}}_{(\mathbf{x},y)\sim\mathcal{D}}\left[\left|\frac{1}{n}\sum_{i=1}^{n}\mathbf{g}(\mathbf{w};\mathbf{x}^{(i)},y^{(i)})\cdot\mathbf{w}^*-\mathbf{g}(\mathbf{w})\cdot\mathbf{w}^*\right|^2\right]$$

$$\leq\frac{1}{nt^2}\mathop{\mathbf{E}}_{(\mathbf{x},y)\sim\mathcal{D}}[(\mathbf{g}(\mathbf{w};\mathbf{x},y)\cdot\mathbf{w}^*)^2]$$

$$=\frac{4(k^*c_{k^*})^2}{nt^2}\mathop{\mathbf{E}}_{(\mathbf{x},y)\sim\mathcal{D}}[(y\langle\mathbf{He}_{k^*}(\mathbf{x}),\mathbf{w}^{\otimes k^*-1}\otimes(\mathbf{w}^*)^{\perp\mathbf{w}}\rangle)^2].$$

Let $f_{\mathbf{w}^*}(\mathbf{x}):=\langle\mathbf{He}_{k^*}(\mathbf{x}),\mathbf{w}^{\otimes k^*-1}\otimes(\mathbf{w}^*)^{\perp\mathbf{w}}\rangle$. Note that $\mathbf{E}_{\mathbf{x}\sim\mathcal{N}_d}[f_{\mathbf{w}^*}(\mathbf{x})^2]=\|\mathbf{w}^{\otimes k^*-1}\otimes(\mathbf{w}^*)^{\perp\mathbf{w}}\|_F^2=\|(\mathbf{w}^*)^{\perp\mathbf{w}}\|_2^2$. Since $\mathbf{w}^*,\mathbf{w}\in\mathbb{S}^{d-1}$, we have $\|(\mathbf{w}^*)^{\perp\mathbf{w}}\|_2=\sin\theta$. Thus, by Fact D.9,

$$\|f_{\mathbf{w}^*}^2(\mathbf{x})\|_{L^p}=\left(\mathop{\mathbf{E}}_{\mathbf{x}\sim\mathcal{N}_d}[(f_{\mathbf{w}^*}(\mathbf{x}))^{2p}]\right)^{2/(2p)}\leq\left((2p-1)^{k^*/2}\mathop{\mathbf{E}}_{\mathbf{x}\sim\mathcal{N}_d}[(f_{\mathbf{w}^*}(\mathbf{x}))^2]\right)^2\leq(2p)^{k^*}\sin^4\theta.$$

Hence, applying Fact D.10 with $A=y^2$, $B=f_{\mathbf{w}^*}^2(\mathbf{x})$, $\sigma_B=\sin^4\theta$, and $C=k^*$, we obtain

$$\mathop{\mathbf{E}}_{(\mathbf{x},y)\sim\mathcal{D}}[(y\langle\mathbf{He}_{k^*}(\mathbf{x}),\mathbf{w}^{\otimes k^*-1}\otimes(\mathbf{w}^*)^{\perp\mathbf{w}}\rangle)^2]\lesssim\sin^4\theta e^{k^*}\log^{k^*}(B_4/\epsilon).$$

Therefore, it holds that

$$\mathbf{Pr}[|(\widehat{\mathbf{g}}(\mathbf{w})-\mathbf{g}(\mathbf{w}))\cdot\mathbf{w}^*|\geq t]\lesssim\frac{(k^*c_{k^*})^2 e^{k^*}\log^{k^*}(B_4/\epsilon)\sin^4(\theta)}{nt^2},$$

which implies that with probability at least $1-\delta$ it holds

$$(\widehat{\mathbf{g}}(\mathbf{w})-\mathbf{g}(\mathbf{w}))\cdot\mathbf{w}^*\lesssim\sqrt{\frac{(k^*c_{k^*})^2 e^{k^*}\log^{k^*}(B_4/\epsilon)\sin^4(\theta)}{n\delta}}.$$

$\square$

We proceed to the main theorem of this paper. It shows that using at most $\tilde{\Theta}(d^{\lceil k/2\rceil}+d/\epsilon)$ samples, Algorithm 2 (with initialization subroutineAlgorithm 1) generates a vector $\widehat{\mathbf{w}}$ such that $\mathcal{L}_2^\sigma(\widehat{\mathbf{w}})=O(\mathrm{OPT})+\epsilon$ within $O(\log(1/\epsilon))$ iterations.

### E.3 Proof of Main Theorem

**Theorem E.6.** *Suppose that Assumption 1 holds. Choose the batch size of Algorithm 4 to be $n=\Theta(C_{k^*}de^{k^*}\log^{k^*+1}(B_4/\epsilon)/(\epsilon\delta))$, and choose the step size $\eta=9/(40ek^*c_{k^*})$. Then, after $T=O(\log(C_{k^*}/\epsilon))$ iterations, with probability at least $1-\delta$, Algorithm 4 generates a parameter $\mathbf{w}^T$ that satisfies $\mathcal{L}_2^\sigma(\mathbf{w}^T)=O(C_{k^*}\mathrm{OPT})+\epsilon$. The total number of samples required for Algorithm 4 is*

$$N=\Theta\left((k^*/c_{k^*})^2 e^{k^*}\log^{k^*}(B_4/\epsilon)d^{\lceil k^*/2\rceil}+\left(e^{k^*}\log^{k^*+2}(B_4/\epsilon)C_{k^*}\right)\frac{d}{\epsilon\delta}\right).$$

*Proof.* Suppose first that $\mathrm{OPT}\geq(c_{k^*}/(64k^*))^2$, i.e., OPT is of constant value. Then by Claim E.7 we know that for any unit vector $\widehat{\mathbf{w}}$ (e.g., $\widehat{\mathbf{w}}=\mathbf{e}_1$) it holds

$$\mathcal{L}_2^\sigma(\widehat{\mathbf{w}})\leq 2\mathrm{OPT}+4\left(\sum_{k\geq k^*}kc_k^2(1-(\mathbf{w}\cdot\mathbf{w}^*))\right)\leq 2\mathrm{OPT}+4C_{k^*}=O(\mathrm{OPT}).$$

Hence $\widehat{\mathbf{w}}$ is an approximate solution of the agnostic learning problem.

Now suppose $\text{OPT} \leq (c_{k^*}/(64k^*))^2$, then the assumption in Proposition D.2 is satisfied and Algorithm 3 can be applied. Consider the distance between $\mathbf{w}^t$ and $\mathbf{w}^*$ after each update of Algorithm 4. By the non-expansive property of projection operators, we have

$$
\begin{aligned}
\|\mathbf{w}^{t+1} - \mathbf{w}^*\|_2^2 &= \|\text{proj}_{\mathbb{B}_d}(\mathbf{w}^t - \eta \widehat{\mathbf{g}}(\mathbf{w}^t)) - \mathbf{w}^*\|_2^2 \\
&\leq \|\mathbf{w}^t - \eta \widehat{\mathbf{g}}(\mathbf{w}^t) - \mathbf{w}^*\|_2^2 \\
&= \|\mathbf{w}^t - \mathbf{w}^*\|_2^2 + 2\eta \widehat{\mathbf{g}}(\mathbf{w}^t) \cdot (\mathbf{w}^* - \mathbf{w}^t) + \eta^2 \|\widehat{\mathbf{g}}(\mathbf{w}^t)\|_2^2.
\end{aligned} \tag{47}
$$

Let us denote the angle between $\mathbf{w}^t$ and $\mathbf{w}^*$ by $\theta_t$. Furthermore, let us assume for now that $\theta_t$ satisfies $\sin\theta_t \geq 4e\sqrt{\text{OPT}} + \sqrt{\epsilon}$, hence the condition for Lemma E.4 is satisfied. Note by definition $\widehat{\mathbf{g}}(\mathbf{w}^t) \perp \mathbf{w}^t$, hence using Lemma E.4 and Lemma E.5 we have that with probability at least $1 - \delta$, the inner product term in Equation (47) is bounded above by:

$$
\begin{aligned}
\widehat{\mathbf{g}}(\mathbf{w}^t) \cdot (\mathbf{w}^* - \mathbf{w}^t) &= \widehat{\mathbf{g}}(\mathbf{w}^t) \cdot \mathbf{w}^* = (\widehat{\mathbf{g}}(\mathbf{w}^t) - \mathbf{g}(\mathbf{w}^t)) \cdot \mathbf{w}^* + \mathbf{g}(\mathbf{w}^t) \cdot \mathbf{w}^* \\
&\leq \frac{C_1 k^* c_{k^*} e^{k^*/2} \log^{k^*/2}(B_4/\epsilon)}{\sqrt{n\delta}} \sin^2(\theta_t) - \frac{1}{2}\|\mathbf{g}^*(\mathbf{w}^*)\|_2 \sin\theta_t,
\end{aligned} \tag{48}
$$

where $C_1$ is a sufficiently large absolute constant. On the other hand, the squared norm term $\|\widehat{\mathbf{g}}(\mathbf{w}^t)\|_2^2$ from Equation (47) can be bounded above using Lemma E.5 and Corollary E.3:

$$
\begin{aligned}
\|\widehat{\mathbf{g}}(\mathbf{w}^t)\|_2^2 &= \|(\widehat{\mathbf{g}}(\mathbf{w}^t) - \mathbf{g}(\mathbf{w}^t)) + \mathbf{g}(\mathbf{w}^t)\|_2^2 \\
&\leq 2\|\widehat{\mathbf{g}}(\mathbf{w}^t) - \mathbf{g}(\mathbf{w}^t)\|_2^2 + 2\|\mathbf{g}(\mathbf{w}^t)\|_2^2 \\
&\leq \frac{C_2 d(k^* c_{k^*})^2 e^{k^*} \log^{k^*}(B_4/\epsilon)}{n\delta} + (k^* c_{k^*})^2 \text{OPT} + \|\mathbf{g}^*(\mathbf{w})\|_2^2,
\end{aligned} \tag{49}
$$

for a sufficiently large absolute constant $C_2$. Plugging Equation (48) and Equation (49) back into Equation (47), and denoting $\kappa := \max\{C_1, C_2\} k^* c_{k^*} e^{k^*/2} \log^{k^*/2}(B_4/\epsilon)$, we get that with probability at least $1 - \delta$,

$$
\begin{aligned}
\|\mathbf{w}^{t+1} - \mathbf{w}^*\|_2^2 \leq {}& \|\mathbf{w}^t - \mathbf{w}^*\|_2^2 + \frac{\eta\kappa}{\sqrt{n\delta}} \sin^2\theta_t - \frac{\eta}{2}\|\mathbf{g}^*(\mathbf{w}^t)\|_2 \sin\theta_t \\
& + \eta^2 \left( \frac{d\kappa^2}{n\delta} + (k^* c_{k^*})^2 \text{OPT} + \|\mathbf{g}^*(\mathbf{w}^t)\|_2^2 \right).
\end{aligned}
$$

Let us assume first that $\theta_t \leq \theta_{t-1} \leq \cdots \leq \theta_0$ and $\theta_t$ satisfies $\sin\theta_t \geq 4e\sqrt{\text{OPT}} + \sqrt{\epsilon}$. We will argue that in this case $\theta_{t+1} \leq \theta_t$ (in fact, that it contracts by a constant factor). Then, by an inductive argument, we immediately know that the assumption is valid and that $\theta_t$ is a decreasing sequence (as long as $\sin\theta_t \geq 4e\sqrt{\text{OPT}} + \epsilon$). To prove $\theta_{t+1} \leq \theta_t$, recall that in Claim E.1 it was shown that

$$
\|\mathbf{g}^*(\mathbf{w}^t)\|_2 = 2k^* c_{k^*} (\mathbf{w}^t \cdot \mathbf{w}^*)\|(\mathbf{w}^*)^{\perp \mathbf{w}^t}\|_2 = 2k^* c_{k^*} (\mathbf{w}^t \cdot \mathbf{w}^*)^{k^*-1} \sin\theta_t.
$$

Recall that $\mathbf{w}^0$ is the initial parameter vector that satisfies $\mathbf{w}^0 \cdot \mathbf{w}^* \geq 1 - 1/k^*$. By the inductive hypothesis it holds $\theta_t \leq \theta_0$, hence $\mathbf{w}^t \cdot \mathbf{w}^* \geq \mathbf{w}^0 \cdot \mathbf{w}^* \geq 1 - 1/k^*$. Furthermore, as we have $(1 - 1/t)^{t-1} \geq 1/e$ for $t \geq 1$, it holds $1 \geq (\mathbf{w}^t \cdot \mathbf{w}^*)^{k^*-1} \geq 1/e$. Therefore, we further obtain

$$
(2k^* c_{k^*}/e) \sin\theta_t \leq \|\mathbf{g}^*(\mathbf{w}^t)\|_2 \leq 2k^* c_{k^*} \sin\theta_t.
$$

Now choosing $n \gtrsim d\kappa/((k^* c_{k^*})^2 \epsilon\delta)$, and recalling that we have assumed $\sin\theta_t \geq 4e\sqrt{\text{OPT}} + \sqrt{\epsilon}$ by the induction hypothesis, we can further bound $\|\mathbf{w}^{t+1} - \mathbf{w}^*\|_2^2$ above as:

$$
\begin{aligned}
\|\mathbf{w}^{t+1} - \mathbf{w}^*\|_2^2 \leq {}& \|\mathbf{w}^t - \mathbf{w}^*\|_2^2 + \eta((\epsilon/d)^{1/2} - (4k^* c_{k^*}/e)) \sin^2\theta_t \\
& + \eta^2 (k^* c_{k^*})^2 (\epsilon + \text{OPT} + 4\sin^2\theta_t) \\
\leq {}& \|\mathbf{w}^t - \mathbf{w}^*\|_2^2 - (3k^* c_{k^*}/e)\eta \sin^2\theta_t + 5\eta^2 (k^* c_{k^*})^2 \sin^2\theta_t.
\end{aligned} \tag{50}
$$

Observe that since $\theta_t \leq \theta_0$ and by assumption $\mathbf{w}^0 \cdot \mathbf{w}^* = \cos\theta_0 \geq 1 - \min\{1/k^*, 1/2\} \geq 1/2$, we have $\cos(\theta_t/2) \geq \sqrt{3}/2$ and thus it further holds that $(\sqrt{3}/2)(2\sin(\theta_t/2)) \leq \sin\theta_t \leq 2\sin(\theta_t/2)$. Since $\|\mathbf{w}^t - \mathbf{w}^*\|_2 = 2\sin(\theta_t/2)$ follows from $\mathbf{w}^t, \mathbf{w}^* \in \mathbb{S}^{d-1}$, we finally obtain that, with probability at least $1 - \delta$,

$$
\|\mathbf{w}^{t+1} - \mathbf{w}^*\|_2^2 \leq (1 - (9k^* c_{k^*}/(4e))\eta + 5(k^* c_{k^*})^2 \eta^2)\|\mathbf{w}^t - \mathbf{w}^*\|_2^2.
$$

Choosing $\eta = 9/(40ek^*c_{k^*})$ yields (with probability at least $1 - \delta$):

$$
\begin{aligned}
4\sin^2(\theta_{t+1}/2) &= \|\mathbf{w}^{t+1} - \mathbf{w}^*\|_2^2 \\
&\leq (1 - (81/(320e^2)))\|\mathbf{w}^{t+1} - \mathbf{w}^*\|_2^2 \\
&= (1 - (81/(320e^2)))(4\sin^2(\theta_t/2)).
\end{aligned}
\tag{51}
$$

This shows that $\theta_{t+1} \leq \theta_t$, hence completing the inductive argument. Furthermore, Equation (51) implies that after at most $T = O(\log(1/\epsilon))$ iterations it must hold that $\sin\theta_T \leq 4e\sqrt{\text{OPT}} + \sqrt{\epsilon}$, therefore, we can end the algorithm after at most $O(\log(1/\epsilon))$ iterations. Though the contraction Equation (51) only holds when $\sin\theta_T \geq 4e\sqrt{\text{OPT}} + \sqrt{\epsilon}$, we can further show that if after some iteration $t^*$ we have $\sin\theta_{t^*} \leq 4e\sqrt{\text{OPT}} + \sqrt{\epsilon}$, then $\sin\theta_{t^*+1}$ is still of order $\sqrt{\text{OPT}} + \sqrt{\epsilon}$. Concretely, if there exists some step $t^* \leq T$ such that $\sin(\theta_{t^*}) \leq 4e\sqrt{\text{OPT}} + \sqrt{\epsilon}$, then at step $t^* + 1$ it must hold (by Equation (50)):

$$
\sin(\theta_{t^*+1}) \leq \sqrt{1 + 8\eta^2(k^*c_{k^*})^2}\sin(\theta_{t^*}) \leq 3\sin\theta_{t^*} \leq 3(4e\sqrt{\text{OPT}} + \sqrt{\epsilon}).
$$

In other words, for all steps $t^* \leq t \leq T$, it holds that $\sin\theta_t \leq 30(\sqrt{\text{OPT}} + \sqrt{\epsilon})$. Thus, in summary, choosing $T = O(\log(1/\epsilon))$, we get that with probability at least $1 - \delta T$, $\sin\theta_T \lesssim \sqrt{\text{OPT}} + \sqrt{\epsilon}$, and applying Claim E.7 we get:

$$
\mathcal{L}_2^\sigma(\mathbf{w}^T) = O\left(\left(\sum_{k \geq k^*} kc_k^2\right)(\text{OPT} + \epsilon)\right) = O(C_{k^*}\text{OPT}) + \epsilon',
$$

where we set $\epsilon' = \epsilon/C_{k^*} \leq \epsilon/(\sum_{k \geq k^*} kc_k^2)$, and used Assumption 1$(iii)$ that $\sum_{k \geq k^*} kc_k^2 \leq C_{k^*}$.

Thus, choosing $\delta' = \delta T$, where $T = O(\log(C_{k^*}/\epsilon'))$, Algorithm 4 outputs a parameter $\mathbf{w}^T$ such that with probability at least $1 - \delta'$, $\mathcal{L}_2^\sigma(\mathbf{w}^T) = O(C_{k^*}\text{OPT}) + \epsilon'$, with batch size

$$
n = \Theta\left(\frac{dC_{k^*}e^{k^*}\log^{k^*+1}(B_4/\epsilon')}{\epsilon'\delta'}\right).
$$

In summary, the total number of samples required for Algorithm 4 is

$$
N = \Theta\left(\frac{(k^*)^2 e^{k^*}\log^{k^*}(B_4/\epsilon')d^{\lceil k^*/2\rceil}}{c_{k^*}^2} + \frac{C_{k^*}de^{k^*}\log^{k^*+2}(B_4/\epsilon')}{\epsilon'\delta'}\right).
$$

$\square$

The final claim shows that if $\sin(\theta(\mathbf{w}, \mathbf{w}^*)) \lesssim \sqrt{\text{OPT}} + \sqrt{\epsilon}$, then $\mathcal{L}_2^\sigma(\mathbf{w}) \lesssim C_{k^*}(\text{OPT} + \epsilon)$.

**Claim E.7.** *Let $\mathbf{w} \in \mathbb{S}^d$ and denote the angle between $\mathbf{w}$ and $\mathbf{w}^*$ by $\theta$. If $\theta$ satisfies $\sin\theta \leq C(\sqrt{\text{OPT}} + \sqrt{\epsilon})$ for some absolute constant $C$, then we have*

$$
\mathcal{L}_2^\sigma(\mathbf{w}) \lesssim \left(\sum_{k \geq k^*} kc_k^2\right)(\text{OPT} + \epsilon).
$$

*Proof.* Since $\mathbf{w} \cdot \mathbf{w}^* = \cos \theta \geq 1 - \sin^2 \theta$, according to Claim C.4 the $L_2^2$ loss $\mathcal{L}_2^\sigma(\mathbf{w})$ can be upper bounded by

$$
\begin{aligned}
\mathcal{L}_2^\sigma(\mathbf{w}) &\leq 2\text{OPT} + 4\left(1 - \sum_{k \geq k^*} c_k^2 (\mathbf{w} \cdot \mathbf{w}^*)^k\right) \\
&= 2\text{OPT} + 4\left(\sum_{k \geq k^*} c_k^2 (1 - (\mathbf{w} \cdot \mathbf{w}^*)^k)\right) \\
&= 2\text{OPT} + 4\left(\sum_{k \geq k^*} c_k^2 (1 - (\mathbf{w} \cdot \mathbf{w}^*))(1 + (\mathbf{w} \cdot \mathbf{w}^*) + \cdots + (\mathbf{w} \cdot \mathbf{w}^*)^{k-1})\right) \\
&\leq 2\text{OPT} + 4\left(\sum_{k \geq k^*} k c_k^2 (1 - (\mathbf{w} \cdot \mathbf{w}^*))\right) \\
&\leq 2\text{OPT} + 4 \sum_{k \geq k^*} k c_k^2 \sin^2 \theta \\
&\leq 2\text{OPT} + 4\left(\sum_{k \geq k^*} k c_k^2\right) C^2(\text{OPT} + \epsilon) \lesssim \left(\sum_{k \geq k^*} k c_k^2\right)(\text{OPT} + \epsilon).
\end{aligned}
$$

$\square$

