# OpenReview forum: "Sample and Computationally Efficient Robust Learning of Gaussian Single-Index Models"
_NeurIPS.cc/2024/Conference — NeurIPS 2024 poster_

### Official Review · Reviewer_z8KF · 2024-07-07

**Soundness:** 3
**Presentation:** 4
**Contribution:** 3
**Rating:** 7
**Confidence:** 3

**Summary:**

This paper studies efficient parameter estimation from generalized linear models with adversarial noise (a.k.a. agnostic learning), assuming a known link function.
The goal is to find a parameter vector that best explains the data.
The authors show that $d^{\lceil k^*/2 \rceil}$ samples suffice for this problem with $k^*$ being the information exponent.
The algorithm is given by certain projected gradient descent (w.r.t. loss truncated to degree $k^*$ Hermite polynomials) initialized with PCA for tensor unfolding.
The sample complexity matches a known CSQ lower bound $d^{k^*/2}$ for even $k^*$ in the realizable setting (i.e., random noise).

**Strengths:**

NA

**Weaknesses:**

NA

**Questions:**

Major comments:

1. The work [DPVLB24] is compared in B.3, but I'm not sure it's comparable.
That paper obtained the conjecturally optimal sample complexity $d^{\bar{k}^*/2}$ with $\bar{k}^*$ being the generative exponent which can be much smaller than $k^*$.
The way I read [DPVLB24] is that if preprocessing is allowed, $d^{\bar{k}^*/2}$ can be achieved.
Otherwise, we're stuck at $d^{{k}^*/2}$.
In the current paper, no preprocessing is used.
Could the authors comment on the possibility of reducing the sample complexity in the agnostic setting with preprocessing?
Or the difficulty of applying a suitable preprocessing that brings down the effective information exponent?
Note that the optimal preprocessing in [DPVLB24] relies on the link function.
With additional work, they can make it work for misspecified models (a.k.a. unknown link function).
But I'm not sure if this is possible with adversarial noise.

1. I would be cautious to claim that $d^{\lceil k^*/2 \rceil}$ **nearly** matches the CSQ lower bound $d^{k^* / 2}$ since there's a nontrivial gap in the exponent for odd $k^*$.
This is inconsistent with the standard definition of "nearly" in say TCS.

1. Regarding the ceiling in the exponent, I suspect that this is a proof artifact since for tensor PCA, the ceiling in [RM14] can be removed with (much) more careful analysis https://arxiv.org/abs/2110.10210 .

Minor comments:

1. I didn't check all the cited papers (in the intro and appendix B) but would like to invite the authors to caution the following terminology discrepancy.
In classical stats, learning single-index models means learning the unknown link function.
OTOH, for known link function with unknown parameter vector, the model is referred to as generalized linear models.
There're also works consider learning the parameter vector from GLMs with unknown link function (as a nuisance parameter).
However, it seems in recent years, people (especially deep learning theorists) have been interchangeably using SIM and GLM to refer to the above problems.
Please make sure such discrepancies are taken into account when comparing prior works with the present result.

1. Line 63-66, I'm not familiar with the line of cited work on hardness, why is $d^{O(1/\epsilon)}$ regarded as hard?
This seems to me a legitimate polynomial running time.
Could the authors say a few more words about under which computational model (low-deg, SQ, SoS, AMP, etc) the problem is hard, and hard in what sense?

1. Line 72, what does $\sim$ mean?

1. Line 77, the condition $c_{k^*} = \Omega(1)$ doesn't make sense to me.
I'm not sure what's the asymptotic in $\Omega(\cdot)$.
I suppose $\sigma$ is a fixed function independent of $n,d$ (otherwise it again doesn't seem to make sense), so $c_{k^*}$ is either zero or strictly nonzero (note also that it can be negative).

1. Line 103, could the authors also specify the dependence of running time on $1/\epsilon$?

1. Paragraph after Thm 1.2: there's a huge difference between CSQ and SQ. Are there more formal reasons (e.g., SQ lower bound) to believe that agnostic learning GLM is still hard under SQ (beyond the fact that known SQ algorithms for realizable learning fail)?

1. Line 195, "small constant $\epsilon_0$" is mildly misleading -- $1/2$ doesn't seem to be a small constant...

1. End of line 200, there's a redundant 2 in the exponent.

1. Line 223, is $l$ defined to be $\lceil k/2 \rceil$ or $\lfloor k/2 \rfloor$?

1. Line 2 of Algorithm 2, the subroutine Initialization is undefined. I assume it refers to PCA for tensor unfolding.

1. Line 557 is somewhat confusing, do the authors mean that **for any** positive constant $c$, partial trace cannot achieve $\mathbf{w}^0 \cdot \mathbf{w}^* \ge c$?

1. Line 572, I'm not sure why $\mathbf{w}^*$ is desired to be the unique top eigenvector.
Do the authors mean that $\mathbf{w}^*$ is an outlier top eigenvector (meaning there's an eigengap between top two eigenvalues)?

1. Line 581-583, I think the exact information theoretic threshold is known for tensor PCA: https://arxiv.org/abs/1812.03403

1. Line 590, I think [HSS15] studies SoS instead of tensor unfolding (correct me otherwise).
For tensor unfolding, the correct reference is https://arxiv.org/abs/2110.10210 which confirms the $d^{(k-2)/4}$ conjecture of [RM14].
I'm not sure of the relevance of the bound $d^{k/4}$.

1. Line 591, I don't understand where the $O(d)$ bound for unfolding comes from.
The suboptimal analysis of [RM14] already gives $d^{1/2}$.
And again, the above reference shows that $d^{1/4}$ suffices for tensor unfolding (which is sharp for unfolding).
I'm not sure the relevance of $d^{3/4}$ here.

---

> ### Author Rebuttal · Authors · 2024-08-07
>
> We thank the reviewer for the time and effort in reviewing our paper, as well as the positive evaluation of our work. We respond to the reviewer's comments below.
>
> #### Response to Questions
>
> ##### Question 1:
> (Q1) We agree that care is needed when comparing to [DPVLB24]. As the reviewer remarks, it is unclear whether their preprocessing ideas can be transferred to the agnostic model, for the following reasons.
>
> First, the noise model considered in [DPVLB24] **does not cover the agnostic noise** setting. In particular, [DPVLB24] requires the label $Y$ to be independent from $w\cdot x$ for any $w\perp w^*$, which excludes even some standard classes of structured semi-random noise such as the Massart noise. Thus, it is **unclear** whether the SQ lower bound obtained in [DPVLB24] is **tight** for the **agnostic** setting.
>
> Second, it is not clear how to apply the label transformation process $T(y) = E[he_{\bar{k^*}}(w^*\cdot x) | Y = y]$ in the **agnostic** setting. Since the distribution of $Y$ in **unknown**, we cannot compute this conditional expectation. Further, how to even approximate this function $T(y) = E[he_{\bar{k^*}}(w^*\cdot x) | Y = y]$ is unclear when the labels take real values.
>
> The difficulty of calculating or approximating this conditional expectation in the agnostic setting also poses challenges in transferring this measure of hardness to our setting. In particular, it is not clear how to calculate the generative component $\bar{k^*}$ uniformly over **all possible noise distributions**.
>
> Finally, we note that other data transform processes also **fail** in the **agnostic** setting. For example, the SQ algorithm proposed in [CM20] that applies an implicit data transformation process via a thresholding procedure, **provably fails in our setting**, as we have remarked in the introduction. That said, we do not claim that further reducing our sample complexity via other more sophisticated techniques is impossible, and we leave it as an interesting future direction.
>
> We further refer to our response to reviewer 9olo, weakness 2 and question 1, for additional  comparison with [DPVLB24].
>
> (Q2&Q3) Thank you for pointing this out; we will revise appropriately. We also thank the reviewer for the provided reference, which we will look into for possibly closing the gap mentioned by the reviewer.
>
> #### Minor comments:
> (C1) We thank the reviewer for pointing out the possible ambiguity in the terminology. We chose to be consistent with the terminology used in the existing line of work addressing the same activations, from [CM20] to [DPVLB24]. However, we will add a remark to disambiguate.
>
> (C2) We are interested in efficient algorithms with sample complexity and runtime that are polynomial in both $d$ and $1/\epsilon$, as $\epsilon$ is usually chosen as a small error tolerance. It was showin in [GGK20,DKZ20,DKPZ21] that for any SQ algorithm, agnostically learning ReLU (one of the most basic activations) to error $\mathrm{OPT}+\epsilon$ requires at least $d^{\Omega(\mathrm{poly}(1/\epsilon))}$ queries, which is not polynomial in $1/\epsilon$. Thus, this line of prior work rules out the possibility of having an SQ algorithm that achieves $\mathrm{OPT} +\epsilon$ error with $\mathrm{poly}(d, 1/\epsilon)$ complexity under our setting. Moreover, the same hardness result ($d^{\Omega(\mathrm{poly}(1/\epsilon))}$ complexity lower bound) holds for low-degree polynomial tests (using the near-equivalence between SQ and low-degree tests shown in [BBHLT21]) and via reduction-based computational hardness (cryptographic hardness) [DKR23].
>
> (C3) In line 72, we use $\sim$ to denote that the function on both sides is equal in the Hermite orthonormal basis under Gaussian measure. Note that $\sigma(z)$ might not be equal to $\sum_{k\geq 0} c_k he_k(z)$ pointwise due to the Gibbs phenomenon of Fourier expansions. We will clarify.
>
> (C4) Thank you for catching this typo, it should be $|c_{k^*}| = \Omega(1)$, meaning that it is a universal constant, independent of other problem parameters like $d$ and $1/\epsilon$.
>
> (C5) It takes roughly $d^{k^*}$ time to read one $k^*$-Chow tensor, and thus it takes $\tilde{O}(d^{3\lceil k^*/2\rceil} + d^{k^*}/\epsilon)$ time to read all $k^*$-Chow tensors required for estimating the unfolding matrix. Since we only need to estimate the top singular vector to constant error, this can be done with a standard matrix SVD algorithm (like power iteration) in time no higher than reading the aforementioned tensors, up to constants. The optimization subroutine takes time $O(nT) = O(d\log(1/\epsilon))$. Thus, in summary, the total runtime of our algorithm would be $\tilde{O}(d^{3\lceil k^*/2\rceil} + d^{k^*}/\epsilon)$.
>
> (C6) If one is interested in error  $\mathrm{OPT} +\epsilon$, such SQ lower bounds are known (as explained above). As elaborated in our response to reviewer 9olo, the SQ complexity of agnostically learning SIMs to error $O(\mathrm{OPT}) +\epsilon$ is not well-understood. Specifically, it is unclear if the SQ complexity is the same or significantly higher than that of the realizable setting. Fully characterizing the SQ complexity of the agnostic problem remains an interesting open problem. The main point is that our algorithm is the best known algorithm for this problem to date and almost matches the lower bound for a natural restricted class of algorithms (CSQ algorithms).
>
> (C9&10) In line 223, $l$ is defined as $\lfloor k/2 \rfloor$; in line 2 of algorithm 2, the initialization subroutine is the tensor PCA algorithm, i.e., algorithm 1 –– we will clarify this in the description of algorithm 2.

---

> > ### Comment · Reviewer_z8KF · 2024-08-11
> >
> > I thank the authors for their detailed response.
> > The comments regarding preprocessing appear convincing to me and my overall evaluation remains unchanged.
> >
> > I have one more quick clarification question that's somewhat related to (C2), (C5).
> > Should I think of $\epsilon$ as a constant or depending on $d$ (potentially arbitrarily)?

---

> ### Author Response · Authors · 2024-08-07
>
> (C11) Our intention was to explan that even when $\mathrm{OPT}$ is very small, i.e, when $\mathrm{OPT}\gtrsim d^{-(k^* - 2)/2}$, we cannot use the partial trace algorithm to find a vector $w^0$ such that the inner product between $w^0$ and $w^*$ is greater than some positive absolute constant. Note that with a random initialization, one easily gets a vector $w^0$ such that $w^0\cdot w^* \approx 1/\sqrt{d}$ with high probability; however, this trivial alignment is not sufficient for our optimization subroutine to work.
>
> (C12) We want to point out that there are no significant eigengaps between the eigenvalue of $w^*$ and the eigenvalues of other eigenvectors that are orthogonal to $w^*$; hence, we cannot pick out the target vector $w^*$ from the eigenvectors of the partial-trace matrix efficiently.
>
> (C14&15) We apologize for the confusion in the prose of lines 587-594. We emphasize that in Appendix B.4 the definition of the signal strength $\tau$ is different from the definition of the signal strength $\beta$ in [RM14]. In particular, since [RM14] takes a normalization step, it holds that $\tau \approx \sqrt{d}\beta$, as remarked in footnote 1 in [HSS15]. Hence, since [RM14] requires $\beta\gtrsim \sqrt{d}$ for tensor-unfolding, it translates to $\tau\gtrsim d$ in our setting. We have added a footnote to remark on the relation between the definition of $\beta$ in [RM14] and the definition of $\tau$ in our setting, and we have modified lines 587-594 to clarify this.
>
> Reference:
>
> [BBHLT21]Statistical Query Algorithms and Low-Degree Tests Are Almost Equivalent, Matthew Brennan, Guy Bresler, Samuel B. Hopkins, Jerry Li, Tselil Schramm COLT 2021

---

### Official Review · Reviewer_wBuC · 2024-07-09

**Soundness:** 3
**Presentation:** 3
**Contribution:** 3
**Rating:** 7
**Confidence:** 4

**Summary:**

This paper studies the problem of _agnostic learning_ of single-index models, which consists of trying to learn a distribution $\mathcal D$ on $\mathbb R^d \times \mathbb R$ with an estimator of the form $y = \sigma(\langle w, x \rangle)$. Since the problem is not necessarily well-posed, the goal is to reach a performance of at most $C \cdot OPT + \epsilon$, where $OPT$ is the best possible performance.

The authors present an algorithm that achieves this bound with a sample complexity of $d^{\lceil k^\star/2 \rceil} + d/\epsilon$, where $k^\star$ is the _information exponent_ of the function $\sigma$. This algorithm proceeds in two steps: the first obtains an informed initialization of the optimal vector $w^\star$ through a tensor unfolding method, which is then passed through an SGD algorithm to obtain the final estimator.

**Strengths:**

I found the topic of the paper very interesting; I was mostly familiar with the realizable setting, and the adversarial version provides interesting insights and challenges. I particularly appreciated Appendix B, which compares in-depth the present work with previous papers on both the realizable and agnostic settings.

The results encompass a wide class of link functions, and make no assumption on the data distribution at all (apart from Gaussian marginals). The arguments to handle such a diverse class of problems are conceptually interesting, and (as far as I checked -- see `Weaknesses`) correct.

**Weaknesses:**

Apart from a few qualms with the presentation (see the minor remarks), my main issue with the paper is simply its length. With the short review time and high review count, it is strictly impossible to certify that a proof which consists of 20-25 pages of dense math, with many computation-heavy steps, is correct. I did check that the proof outline and the main steps seem correct, but it is my opinion that such a paper is incompatible with the Neurips review format, and should be submitted to a journal instead. My recommendation for acceptance, based on the overall quality of the paper, should be taken with such a caveat.

Minor remarks:
- the vector $w^\star$ is never rigorously defined; for the sake of exposition, the link between solving Problem 1.1 and obtaining good alignment with $w^\star$ should be emphasized.
- In Assumption 1, the wording is confusing: since $\sigma$ is fixed, what is the $c_k^\star = \Omega(1)$ referring to ? My best guess is that the results hold uniformly over a class of link functions where $c_k^\star$ is bounded from below, and $C_k^\star$ and $B_4$ from above, but this should be made apparent if this is the case.
- there should be a $w^t$ in the RHS of eq. (16).
- You mention in the introduction that achieving $OPT+\epsilon$ performance is likely to be hard, but it's difficult to parse why the proof wouldn't extend in this case: this seems hidden in the recursion of Theorem 3.5. Highlighting this barrier would be a nice addition to the proof.

**Questions:**

- In papers on the realizable setting (e.g. [DNGL23, DPVLB24]), the difficulty of the problem is not measured in terms of the link function $\sigma$, but the distribution of $(x, y)$ (assuming that $\sigma$ is ``nice''). Can such a measure be defined in your setting, instead of considering the worst-case scenario on $(x, y)$ ?
- On a related note, in the aforementioned setting, a mismatch between $\sigma$ and the ``true'' link function can lead to a value of $OPT$ of order $\Omega(1)$, but some algorithms manage to achieve almost perfect alignment with the vector $w^\star$. In your case, a large value of $OPT$ yields vacuous bounds on both the performance of your algorithm and its alignment with $w^\star$; is there an explanation of this phenomenon ?

---

> ### Author Rebuttal · Authors · 2024-08-07
>
> We thank the reviewer for the time and effort in reviewing our paper and the positive assessment. Below, we provide specific responses to the points and questions raised by the reviewer.
>
>
>
> >(Question 1): Can such a measure be defined in your setting, instead of considering the worst-case scenario on (x,y)?
>
> The prior work [DPVLB24] considered the generative exponent $\bar{k}^*$ of the label $y$, which is defined as the first non-zero term $E_y[(E_x[he_k(w^*\cdot x)|y])^2]$. It is not clear how to leverage such a measure for the agnostic setting for a number of reasons, including the fact that the "noise" in our setting is **unknown** to the algorithm and that the labels may take real values (in which case it is unclear how to estimate the conditional expectation stated above).
>
>
> > (Question 2): a mismatch between 𝜎 and the ``true'' link function can lead to a value of 𝑂𝑃𝑇 of order Ω(1), but some algorithms manage to achieve almost perfect alignment with the vector $w^*$. In your case, a large value of 𝑂𝑃𝑇 yields vacuous bounds on both the performance of your algorithm and its alignment with $w^*$; is there an explanation of this phenomenon ?
>
> This is indeed the case and it is a consequence of the **agnostic** model. Specifically, in the presence of agnostic noise, it is not possible to achieve perfect recovery for the target vector $w^*$ (i.e., find a $w$ such that $w\cdot w^*\geq 1 - \epsilon$ for any desired accuracy $\epsilon>0$) -- even information-theoretically. Specifically, one can construct examples where two different weight vectors both achieve the optimal error while being far from each other. This is a fairly standard fact in agnostic learning and holds even for simple activations like ReLU. In the **realizable** setting, the reason that one can recover the target vector $w^*$ almost perfectly is that the distribution of the label $y$ is **known**, and hence it is possible to manipulate the labels to gain more information about $w^*$.
>
>
>
> >(Minor Remark 1) the vector $w^*$ is never rigorously defined
>
>
> We thank the reviewer for pointing this out. We clarify that $w^*\in\mathrm{argmin}_{w\in\mathbb{S}^{d-1}} \mathcal{L}_2^{\sigma}(w)$; i.e., $w^*$ is defined as a vector that achieves the minimum $L_2^2$ loss. We have added this definition to the main body.
>
> > (Minor Remark 2) My best guess is that the results hold uniformly over a class of link functions where $c_{k^*}$ is bounded from below, and $C_{k^*}$ and $B_4$ from above
>
> The reviewer’s understanding of assumption 1 is correct. We further clarify that the activation $\sigma$ is fixed, but it needs to satisfy assumption 1 so that our algorithm achieves $C\cdot \mathrm{OPT}+\epsilon$ error, where $C$ is an absolute constant, using the sample complexity and runtime as claimed in Theorem 1.2. The constant $C$ in the error depends on the parameter $C_{k^*}$, therefore, if $C_{k^*}$ is not an absolute constant, then our algorithm does not achieve constant factor approximate error. The parameters $c_{k^*}$ and $B_4$ appear in the sample complexity. This implies that if $c_{k^*}$ and $B_4$ are not independent of $d$, the sample complexity and runtime might not be of order $O(d^{\lceil k^*/2 \rceil} + d/\epsilon)$.
>
> >(Minor Remark 3): You mention in the introduction that achieving 𝑂𝑃𝑇+𝜖 performance is likely to be hard, but it's difficult to parse why the proof wouldn't extend in this case: this seems hidden in the recursion of Theorem 3.5. Highlighting this barrier would be a nice addition to the proof.
>
> Thank you for the suggestion; we'll incorporate it.
>
> First, we note (as stated in the paper) that there exist both SQ lower bounds and reduction-based hardness results implying that achieving error $\mathrm{OPT}+\epsilon$ requires $d^{\mathrm{poly}(1/\epsilon)}$ time, even for a ReLU activation. To see why a constant factor approximation is inherent in our algorithmic approach, we refer to the Technical Overview section (lines 176-185). The main technical reason is that the sharpness structural does not hold on the whole sphere –– it is only valid on the sphere excluding a spherical cap centered at the target vector $w^*$. Specifically, by Lemma 3.3, we have sharpness on the subset $\mathbb{S}^{d-1} \setminus \mathcal{S}$, where $\mathcal{S} = \lbrace w: || w || = 1, \sin(\angle(w, w^*)) \leq 4e\sqrt{\mathrm{OPT}} \rbrace$. This restriction of sharpness is due to the strong agnostic noise. Since every point in the spherical cap $\mathcal{S}$ is a $O(C_{k^*}\mathrm{OPT})+\epsilon$ accurate solution (Claim E.7),  we can terminate the algorithm after entering this spherical cap if we are only looking for constant factor approximate solutions. However, since sharpness no longer holds once the algorithm’s iterates enter this spherical cap $\mathcal{S}$, we lose the information about the direction in which we should update $w$; hence, we cannot continue decreasing the error to $\mathrm{OPT} + \epsilon$.

---

### Official Review · Reviewer_9oLo · 2024-07-14

**Soundness:** 3
**Presentation:** 2
**Contribution:** 2
**Rating:** 4
**Confidence:** 3

**Summary:**

The paper provides a polynomial-time algorithm reaching the optimal CSQ sample complexity (upto sub-leading factors) for a general class of single-index models under the setup of adversarial noise. Similar to existing works, the algorithm utilizes the empirical estimate of the k_th Hermite tensor of the target to obtain an initialization with a non-trivial overlap (weak-recovery) by estimating the top singular vector of an unfolded tensor. Subsequently, the algorithm utilizes mini-batch SGD on the sphere starting from non-trivial overlap to reach the optimal error upto a constant factor.

**Strengths:**

The paper makes novel technical contributions towards understanding the sample-complexity of learning under the adversarial noise. Understanding the differences between the adversarial and realizable/random noise settings is crucial towards justifying applicability of CSQ and SQ lower-bounds in typical maching learning setups. The paper adequetly describes the technical challenges in matrix-concentration under adversarial noise as well as why the partial trace algorithm in Damian et al. (2024) cannot be directly applied under the adversarial noise setting (Appendix B.3).

**Weaknesses:**

* The central ideas of obtaining weak-recovery by estimating the Hermite tensor and subsequently running minibatch SGD is largely similar to existing works (Biroli et al. (2020), Damian et al. (2023), Damian et al. (2024), Chen and Meka (2020)). Therefore, the major contribution of the work are adapting the ideas and analysis in these papers to adversarial noise, which has some technical but limited conceptual novelty. Even under adversarial noise, prior works have establishing similar guarantees for smaller classes of link functions. It is unclear whether the setup of  adversarial noise is relevant for gradient based training in machine learning models. Discussion of the motivation behind studying the adversarial noise  and the novel conceptual contributions/implications of the work will improve the paper.

* In the realizable and random-noise case, recent works Dandi et al. (2024) and Damian et al. (2024) show the learnability of multi/single index models under reduced SQ complexity through reuse of data (implicitly transforming the labels) and explicit transformation of labels respectively. In light of these works, it has become apparent to the community that the SQ class is more suitable towards describing the limitations of gradient-based learning rather CSQ. Therefore, it is important for the paper to justify the choice of the CSQ class and discuss the effect of data reuse, label transformation. The paper presently only discusses the limitation of some SQ algorithms such as the one in Damian et al. (2024) under adversarial noise. This alone doesn’t justify the relevance of CSQ lower-bounds under the possibility of label transformation in the adversarial noise case.

Missing references of closely related works:
- Biroli, Giulio, Chiara Cammarota, and Federico Ricci-Tersenghi. "How to iron out rough landscapes and get optimal performances: averaged gradient descent and its application to tensor PCA." Journal of Physics A: Mathematical and Theoretical 53.17 (2020): 174003.
- Dandi, Y., Krzakala, F., Loureiro, B., Pesce, L., & Stephan, L. (2023). How two-layer neural networks learn, one (giant) step at a time. arXiv preprint arXiv:2305.18270.
- Dandi, Y., Troiani, E., Arnaboldi, L., Pesce, L., Zdeborova, L., & Krzakala, F. The Benefits of Reusing Batches for Gradient Descent in Two-Layer Networks: Breaking the Curse of Information and Leap Exponents. In Forty-first International Conference on Machine Learning.

**Questions:**

* Is the proposed algorithm expected to reach the SQ lower-bounds upon transformation of the labels?

* Intuitively, why does unfolding fare better than the partial trace estimator under adversarial noise?

* Can the memory and runtime requirements for Algorithm 1 be optimized simlar to the partial trace algorithm (see Remark 4.2 in Damian et al. 2024)?

* The discussion in lines 158-161 regarding the Gaussianity of the noise term in prior works isn’t accurate since Damian et al. 2024 also consider arbitrary labels leading to general noise.

* Why using the tensor PCA for a star while we know there are more efficient starting point for such problems, e.g. (https://arxiv.org/abs/1708.05932, arXiv:1811.04420,https://arxiv.org/abs/2012.04524) ?

*  Correction: The references list "G. B. Arous" instead of "G. Ben Arous" (Ben Arous is the surname, not a second name). This should be corrected to accurately reflect the author's name.

**Limitations:**

The work precisely describes the theoretical assumptions. One major limitation of the work is the absence of a discussion of label transformations/batch reuse (see weaknesses above). The work is primarily of a theoretical nature and has no potential negative societal impact.

---

> ### Author Rebuttal · Authors · 2024-08-07
>
> **General Response to Reviewer 9oLo**
>
> We thank the reviewer for the provided feedback. Before responding to specific questions and comments, we address two main points, which we believe are the main sources of the reviewer's somewhat negative view of our work. We hope that upon clarifying the context, the reviewer would consider reevaluating our work.
>
> First, our work is motivated by the line of work on **agnostic learning** of single-index models under structured marginal distributions and broad classes of activations, as discussed in the top-level response. Agnostic learning, introduced in [Hau92, KSS94], is a well-established model meant to capture **realistic** learning settings, where we do not assume that the labels perfectly follow a model from the class (where "perfect" also accounts for, e.g., zero-mean noise), and the goal is to be competitive with the best-fit model from the class. The line of work on agnostic learning has a long history in the learning community, with papers regularly published at top ML theory venues such as COLT, NeurIPS, and ICML over the past three decades.
>
> The aforementioned line of work on agnostically learning SIMs (see the top-level response) elaborates on the  difficulties of polynomial sample and time learning, and develops the first **efficient** and **constant-factor** **agnostic** algorithms for **monotone** activations. While these algorithms rely on first-order methods, they do not rely on “vanilla SGD on the square loss” and their analyses are fairly sophisticated.
>
> Second, it is important to note here that our focus was **not on the CSQ model** itself or on the optimality of our sample complexity for **all SQ algorithms**. It is a plausible conjecture that the sample complexity of our algorithm can be improved (by an appropriate label transformation preprocessing or some other method) and this remains an interesting open question for future work. As we explain below, the applicability of existing such approaches is **unclear**. Importantly, our main result is **the first** constant-factor agnostic learner with polynomial sample and time complexity addressing the broad class of activations defined via Hermite polynomials. It is also **the most sample and computationally efficient algorithm** for this task known to date — not only within the class of first order/SGD-type algorithms, but **in general**.
>
> ---
> Below we address specific comments and questions raised by the reviewer.
>
> #### Weakness 1
> We respectfully disagree with the reviewer’s points. As is explained in the submission and we reiterate below, there is a vast difference between the **realizable/random noise** setting and the challenging **agnostic** setting that we study (both in terms of algorithms and analysis).
>
> Conceptually, it is important to recall that our goal is to obtain error $O(\mathrm{OPT})+\epsilon$, which is the best possible error achievable in polynomial time (please see our general response). None of the aforementioned works achieves such a guarantee, even for very special cases, e.g., for a ReLU activation. (That is, the algorithms themselves in these prior works do not suffice; not merely the analyses.)
>
> At the technical-level, our approach also differs significantly from these prior works. Specifically, our algorithm has two main components:  an initialization subroutine and an optimization algorithm. Both are new and require novel analysis. To obtain a non-trivial weak recovery in the initialization step, we carried out a **fine-grained** analysis of a $k$-tensor-PCA algorithm in the presence of agnostic noise. The optimization algorithm is different from these prior works as well and its analysis hinges upon a critical **structural result** for the $L_2^2$ loss, which we term ‘(alignment) sharpness’ (Lemma 3.3). In more detail, we show that the Riemannian gradient $\mathbf{g}$ of the $L_2^2$ loss of the *truncated activation* contains a **strong signal** in the direction of $w^*$: $\mathbf{g} \cdot w^* \leq -\mu\sin^2(\theta)$, where $\mu$ is an absolute constant. Intuitively, this structural result conveys that the gradient vector $\mathbf{g}$ can *pull* the algorithm iterates towards $O(\mathrm{OPT})+\epsilon$ solutions. **None** of the prior works listed by the reviewer established such a structural result, which is **key** to obtaining a constant factor approximation. We refer to Section 1.2 (Technical Overview, line 176-192) for a more detailed discussion.
>
> >It is unclear whether the setup of adversarial noise is relevant for gradient based training in machine learning models [...]
>
> As noted in our general response, the agnostic model is **fundamental** in learning theory and has been **extensively studied** in the context of learning SIMs with monotone activations. From a practical perspective, the realizable or random label noise settings are often unrealistic, as they posit the existence of a model that perfectly fits the data (possibly on average).
>
> #### Weakness 2
>
> As noted in our general response, the focus of our work is **not on the CSQ vs SQ** distinction. We provide **the first polynomial-time algorithm** for our problem that achieves near-optimal error of $O(\mathrm{OPT})+\epsilon$; a goal **not achieved by these prior works** (with any polynomial sample complexity). Importantly, when one talks about "SQ complexity" in the agnostic model, the accuracy achieved by the algorithm is **critical**. Please see below as a response to the reviewer's relevant question.

---

> ### Author Response · Authors · 2024-08-07
>
> #### Response to Questions
>
> (Q1) It is important to note that the SQ complexity of the learning problem in the **agnostic** setting is **not necessarily the same** as the SQ complexity in the **realizable or random noise** cases. Specifically, the SQ complexity in the agnostic setting **depends on the desired accuracy**. For example, for accuracy $\mathrm{OPT}+\epsilon$, the SQ complexity of agnostically learning a ReLU (corresponding to $k^{\ast}=1$) under Gaussian marginals is known to be $d^{\mathrm{poly}(1/\epsilon)}$ -- i.e., **exponential** in $1/\epsilon$ [GGK20,DKZ20,DKPZ21]. In contrast, in the realizable/random noise setting, the SQ complexity is $\mathrm{poly}(d/\epsilon)$, as long as $k^{\ast}$ is bounded by an absolute constant. While an SQ *lower bound* for the realizable setting also applies to the agnostic setting, it is **unclear if a matching SQ upper bound exists**. In particular, it remains **an open problem** what the SQ complexity of our agnostic problem is for error $O(\mathrm{OPT})+\epsilon$.
>
> At a more technical level, it is not clear whether the approaches of [DPVLB24, DTA+24] can be leveraged in the agnostic setting to achieve a constant factor approximation. Regarding [DPVLB24], as explained in our submission, Section B.3 line 536-547, the generative exponent defined there is **specific to the joint distribution** $P(x,y)$ (in their notation), where the labels $y$ are corrupted by **structured and known** noise and this is **significantly weaker** than the agnostic model. Notably, they require the noise $\xi$ to be independent of $w\cdot x$ for any $w \perp w^*,$ which **excludes** even, for instance, Massart noise. Regarding the result in [DTA+24], it is unclear that it suffices **even for weak recovery** under the **agnostic** setting. The reason is that, as shown in [LOSW24], the sample-reuse method implements **monomial transformation** on the labels $y$, but in the **agnostic** setting the labels are **not guaranteed to have bounded higher moments**.
>
> (Q2) At a high level, applying the partial trace operator to a tensor can be viewed as **smoothing the tensor PCA objective** (see [ADGM17]). While in the **realizable** setting, smoothing the objective **helps** the optimization algorithm escape the local minima near the equator, in the **agnostic** setting, smoothing the landscape could also **bury the signal** that is already very weak. In particular, the partial trace operator sums up many entries of the noise tensor. Due to the agnostic nature of the noise, this unfortunately further **corrupts** the labels and makes it harder to discover the signal of the target vector $w^*$ from the tensor.
>
> (Q3) We think this might be possible using the special structure of the Hermite tensors. However, this is beyond the scope of our work, and we leave it as a future direction.
>
> (Q4) First, the label noise handled by Damian et al. 2024 is **not arbitrary**; please see the second paragraph in our response to the first question.
>
> Additionally, we would like to clarify that here we are referring to the traditional tensor PCA methods (provided in [RM14]), which rely on the Gaussianity of the noise tensor. We emphasize that the analysis and guarantees of traditional tensor PCA methods, like the tensor unfolding or partial trace, **cannot be directly applied to our setting**, and a fine-grained analysis is required. To avoid confusion, we have modified the phrasing in line 158-161.
>
> (Q5) We respectfully disagree with the reviewer’s comment. First, we are not sure what  ‘such problems’ in the reviewer’s question refers to. The articles the reviewer listed provide matrix spectrum methods for phase retrieval problems, which address **one specific link function** with **information exponent** $k^* = 2$. In our work, we require algorithms that can **weakly recover** the signal for **much more general link functions** with **any constant information exponent** $k^*$ from a high dimensional tensor. Furthermore, the methods the reviewer points to are **not** qualitatively more efficient compared to our initialization method. Specifically, they require $n\propto d$ samples **asymptotically**. This is the same as the sample complexity of our initialization algorithm, which requires $n = O(d)$ samples **non-asymptotically** for phase retrieval problems, as $k^* = 2$ in this case. In fact, when $k^* =2$, our initialization algorithm is simply a Chow-matrix PCA algorithm that can be carried out efficiently using power iteration.
>
> References:
>
> [DTA+24] Y. Dandi, E. Troiani, L. Arnaboldi, L. Pesce, L. Zdeborova, and F.
> Krzakala. The benefits of reusing batches for gradient descent in two-layer networks: Breaking the curse of information and leap exponents. In Forty-first International Conference on Machine Learning, 2024.
>
> [LOSW24] J. D. Lee, K. Oko, T.i Suzuki, D. Wu. Neural network learns low-dimensional polynomials with SGD near the information-theoretic limit. https://arxiv.org/abs/2406.01581.

---

> > ### Comment · Reviewer_9oLo · 2024-08-11
> > **Your answer Q5 is not correct**
> >
> > With respect to Q5, your statements are uncorrected.
> >
> > The papers I mentioned solve ALL problems with generative (and not information, as you claim) exponent up to two. This means that virtually any link function (except made-up, carefully fine-tuned ones) are covered. See, for instance, the discussion on page 5 in the first paragraph of https://arxiv.org/pdf/2403.05529: "In fact, [MM18, BKM+19, MLKZ20] give a necessary and sufficient condition on P that enables such T to lower the information exponent to 2."
> >
> > I invite the authors to check theorem 1 and theorem 2 in [MM18] https://arxiv.org/abs/1708.05932, which are for "generic sensing models," (this is also the case of [BKM+19, MLKZ20], phase retrieval is just an a particular application). The authors can also check https://arxiv.org/abs/2012.04524, section 1.2 Main results, eq(5) for optimal transformation valid for almost any link function.
> >
> > The ONLY functions not covered in these works are those with generative (again, not information) exponent 3 and higher (see again https://arxiv.org/pdf/2403.05529), a class that essentially consists of fine-tuned functions, corresponding to unnatural, made-up problems, without much application for generic single-index exponents.
> >
> > To quote your answer, all these papers do study  "algorithms that can weakly recover the signal for much more general link functions with any constant information exponent $k^*$" (in fact they do so for arbitrary k^* with just O(d) samples and O(dlogd) iterations) and I thus invite the authors to check these papers carefully.

---

> > > ### Author Response · Authors · 2024-08-12
> > >
> > > We reiterate that our work provides guarantees in the *agnostic model*, where the labels $y$ can be arbitrarily corrupted. Consequently, a function of the form $f(x)= E[y|x]$  is  *unknown* to the learner and cannot even be estimated efficiently (as the $x$-marginal follows a standard normal, which makes it impossible to sample the same point twice).
> > > The works cited by the reviewer give necessary conditions to lower the information exponent down to $2$, assuming that the distribution of $y$ is *known* to the learner. We invite the reviewer to check the statement of Theorem 2 in [MM18], and in particular how the function $T(y)$ is defined. In summary, the algorithms given in these works do not succeed in our corruption model.
> > >
> > > Moreover, even assuming the distribution of $y$ is known to the learner (in which case the above theorem would be applicable), there exist function/distribution pairs where the information exponent=generating exponent>2 (see for example [MM18] and Figure 2 in [DPVLB24]). For such instances, the aforementioned works do not achieve the stated results. In summary, even for the easier setting where the learner knows the distribution of $y$ a priori, the above works do not in general achieve improved sample complexity for our setting.
> > >
> > > The reviewer argues that “virtually any link function (except made-up, carefully fine-tuned ones)” has a generative exponent up to $2$. We respond in two parts. First, our work is theoretical, establishing algorithmic results under a clearly defined set of assumptions. If we strengthen these assumptions (e.g., assume a generative exponent at most $2$), potentially more efficient results are possible (but even that remains open in the agnostic model). We emphasize however that this is formally a special case of our setting. Second, from the practicality perspective, one can construct natural examples where information exponent=generating exponent>2.  In particular, this holds if the distribution of $y$  is supported in ${\pm 1}$, in which case the two exponents are identical (because in this case SQ and CSQ are known to be equivalent). Note that the distribution of $y$ can be Boolean-valued even if the link function is real-valued (e.g., a sigmoid).

---

### Author Rebuttal · Authors · 2024-08-07

**Top-Level Response**

We thank the reviewers for their time and effort invested in evaluating our paper. We are encouraged by reviewers finding our results **technically novel** (**9olo**), **conceptually interesting** (**wBuC**), and rating the **presentation** of our work as **excellent** (**z8KF**).

Below, we restate our motivation and main contributions, along the way hoping to clarify the context of our work. Specific comments from the reviews are addressed in individual responses. We look forward to the opportunity to respond to further questions and engage in a discussion with the reviewers.

#### Motivation & Context

The main motivation of our work is to develop polynomial-sample and time algorithms for **robustly** learning SIMs when the activation function is **not necessarily monotone**. "Robust" here refers to the well-established **agnostic learning** setting, where the labels do not necessarily correspond to any model from the class (i.e., there is no "perfect" model), and the goal is to be competitive with the *best-fit* model. Prior work [DGK+20, DKTZ22, ATV23, WZDD23, GGKS23, ZWDD24] has developed such efficient algorithms, albeit **restricted to a subclass of monotone functions**.  Obtaining similar results for more general activations, namely for monotone and Lipschitz functions (a **subset** of the activations we handle) was explicitly stated as an open problem in [ZWDD24].

A **major goal** in the agnostic setting is to obtain as high accuracy as possible in polynomial time. The information-theoretically optimal error is $\mathrm{OPT}+\epsilon$ where $\mathrm{OPT}$ is defined as the minimum mean square loss that is attainable by any function in the target class. However, obtaining such an error guarantee requires SQ complexity $d^{\mathrm{poly}(1/\epsilon)}$. The **best error we can hope for** with SQ complexity $\mathrm{poly}(d/\epsilon)$ is $C*\mathrm{OPT}+\epsilon$, where $C>1$ is a **universal constant** independent of the problem dimension. Obtaining such an error guarantee in polynomial time is **highly non-trivial**. Standard SGD-based algorithms and/or their analyses **inherently fail** to achieve this error even for the basic case of a ReLU activation, as discussed, for example, in the introduction  of [WZDD23]. Specifically, even with a tight analysis, the best possible parameter $C$ for such methods would scale polynomially either with the **dimension** or the **diameter** of the space; and the dependence on $\mathrm{OPT}$ would **not necessarily** be **linear** (e.g., scaling with $\mathrm{OPT}^{1/2}$).

The class of activations we consider is the same as considered in the prior work of [DNGL23], also studied in [DPVLB24] (and other works), generalizing the monotone activations in the line of work mentioned above. We emphasize that the algorithms appearing in  [DNGL23, DPVLB24] do **not** achieve the desired $O(\mathrm{OPT})+\epsilon$ error guarantee in the agnostic model, even restricted to the case of a ReLU activation (which corresponds to the information exponent of $k^*=1$).

#### Main Contribution

Our main contribution is **the first polynomial-time algorithm** that achieves the $O(\mathrm{OPT}) + \epsilon$  error guarantee for any activation in the aforementioned broad class, with sample complexity scaling with $d^{\lceil k^*/2\rceil},$ where $k^*$ is the information exponent characterizing activations in the class. For small constant values of $k^*$, this yields **the first polynomial sample and time agnostic learner** that goes well beyond the monotone case. Prior work does not achieve this error guarantee **even when restricted to** $k^*=1$.

---

### Decision · Program_Chairs · 2024-09-25

**Decision:**

Accept (poster)

**Comment:**

This paper make progress on a problem in machine learning theory of significant recent interest: efficiently learning single index models. The paper gives a computationally efficient algorithms with near optimal sample complexity that works even in the challenging agnostic (adversarial noise) setting. Prior work mainly focuses on the realizable setting. While there are areas for the result to be tightened or improved, it seems clear from the discussion that this work will open the doors to future results, and the very knowledgeable reviewers were all positive on the work in the end. We recommend acceptance.